# Quantile Constrained Reinforcement Learning: A Reinforcement Learning Framework Constraining Outage Probability

**Whiyoung Jung, Myungsik Cho, Jongeui Park, Youngchul Sung**[*]
School of Electrical Engineering, KAIST
Daejeon 34141, Republic of Korea
{wy.jung, ms.cho, jongeui.park, ycsung}@kaist.ac.kr

## Abstract

Constrained reinforcement learning (RL) is an area of RL whose objective is to find an optimal policy that maximizes expected cumulative return while satisfying a given constraint. Most of the previous constrained RL works consider expected cumulative sum cost as the constraint. However, optimization with this constraint cannot guarantee a target probability of outage event that the cumulative sum cost exceeds a given threshold. This paper proposes a framework, named Quantile Constrained RL (QCRL), to constrain the quantile of the distribution of the cumulative sum cost that is a necessary and sufficient condition to satisfy the outage constraint. This is the first work that tackles the issue of applying the policy gradient theorem to the quantile and provides theoretical results for approximating the gradient of the quantile. Based on the derived theoretical results and the technique of the Lagrange multiplier, we construct a constrained RL algorithm named Quantile Constrained Policy Optimization (QCPO). We use distributional RL with the Large Deviation Principle (LDP) to estimate quantiles and tail probability of the cumulative sum cost for the implementation of QCPO. The implemented algorithm satisfies the outage probability constraint after the training period.

## 1 Introduction

Reinforcement learning (RL) has been developed in the direction of finding an optimal policy that maximizes expected cumulative return for a given environment. Thus, most of the works in RL consider only rewards given by the environment to optimize the policy. However, many real-world control problems impose constraints on the behavior of a policy. Constrained RL is an area of RL whose objective is to find an optimal policy that maximizes expected cumulative return while satisfying a certain constraint on the cumulative cost. A conventional constrained RL problem (ExpCP) can be written as

$$\begin{array}{ll} \text{Maximize} & V^\pi(s_0) := \mathbb{E}_\pi \left[ \sum_{t=0}^\infty \gamma^t r(s_t, a_t) \right] \\ \text{subject to} & C^\pi(s_0) := \mathbb{E}_\pi \left[ \sum_{t=0}^\infty \gamma^t c(s_t, a_t) \right] \leq d_{th}, \end{array} \quad \text{(ExpCP)}$$

where the cost constraint is that the expectation of the sum of costs is less than or equal to a threshold parameter $d_{th}$. Note that the threshold $d_{th}$ is set on the average (i.e., expectation) of the cumulative sum cost to avoid undesired high-cost events in this formulation. If we do not want any event causing a positive cost, $d_{th}$ should be set as a sufficiently small value, i.e., $d_{th} \approx 0$. On the other hand, if we can afford events with low costs, we can set $d_{th}$ properly as we desire. Most of the previous constrained RL works solved the problem (ExpCP) [1, 12, 14, 24, 25, 28, 29] partly because the

---

[*]Corresponding author

36th Conference on Neural Information Processing Systems (NeurIPS 2022).

constraint on the expectation of the cumulative sum cost in (ExpCP) is well fit with the objective given by the expectation of the sum reward, and this makes the problem amenable. However, solving the problem (ExpCP) may have an undesirable outcome for real environments that typically need a constrained behavior on the event that the cost exceeds the threshold $d_{th}$. For example, in the case of an autonomous driving car, what we want for our sure safety is to control and limit the probability of accident itself. In the case of a telecommunication system, what we want to control is the probability of packet loss through the communication system. These probabilities are called 'outage probability' in general. Thus, in many real-world systems, the system requires a constraint on the outage probability, i.e., the probability of critical or unsafe events. In this case, a constraint on the expectation of critical events, as in (ExpCP), cannot guarantee the desired target probability of critical events. To illustrate this, let us consider the following example.

Consider a simple two-path environment, as shown in Fig. 1a. The objective of the environment is for an agent to reach the goal by driving a car. The environment gives a reward when the agent reaches the goal, and gives costs until the agent reaches the goal. There exist two paths to reach the goal, and each path has a different cost distribution. Fig. 1b shows the distribution of the cumulative sum cost (curve) and its mean

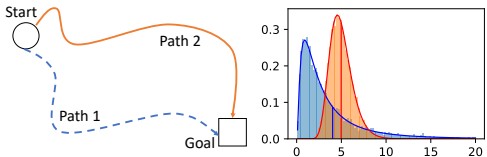

(a) Two-path environment    (b) Distribution of cost

Figure 1: A simple example of environment.

(vertical line) for each path: blue for path 1 and red for path 2. As we can see in Fig. 1b, the distribution for path 1 has a lower average than that for path 2, but has a longer right tail than that for path 2. If we use the expectation of cost as a constraint, then following path 1 is a better choice, but in this case, the probability of a high cumulative sum cost (e.g. > 10.0 in Fig. 1b) is higher than following path 2. If the threshold of 10 represents a catastrophic event, we should constrain the probability of events exceeding 10 to a small value. Thus, solving the problem with the expectation constraint (ExpCP) does not necessarily have precise control over the target outage probability. When the event that the cumulative sum cost exceeds $d_{th}$ is a critical unsafe event, this means that such critical event can occur in high probability even if we solve the constrained problem (ExpCP). Therefore, in this paper, we aim to solve the following constrained RL problem with an outage probability constraint:

Maximize    $V^\pi(s_0) = \mathbb{E}_\pi \left[ \sum_{t=0}^\infty \gamma^t r(s_t, a_t) \right]$
Subject to    $\Pr \left[ \sum_{t=0}^\infty \gamma^t c(S_t, A_t) > d_{th} \right] \leq \epsilon_0$          (ProbCP)
   for $S_0 = s_0, A_t \sim \pi(\cdot|S_t), S_{t+1} \sim M(\cdot|S_t, A_t)$.

Our approach to this problem is first to convert the outage probability constraint in (ProbCP) into a quantile constraint $q_{1-\epsilon_0}^\pi(s_0) := \inf \left\{ x \mid \Pr \left( \sum_{t=0}^\infty \gamma^t c(S_t, A_t) \leq x \right) \geq 1 - \epsilon_0 \right\} \leq d_{th}$ which is equivalent to the outage probability constraint (See Fig. 2), and then to solve the optimization:

$$\min_{\lambda \geq 0} \max_\pi \left\{ V^\pi(s_0) - \lambda \left( q_{1-\epsilon_0}^\pi(s_0) - d_{th} \right) \right\}, \tag{1}$$

where $\lambda$ is the Lagrange multiplier, based on policy gradient with the parameterized policy. However, we note that the policy gradient theorem, which is the most basic theorem for on-policy RL, cannot be applied directly to compute the gradient of (1) with respect to (w.r.t.) the policy parameter due to the quantile term $q_{1-\epsilon_0}^\pi(s_0)$. Therefore, we derive theoretical results for approximating the gradient of the quantile (in Section 3). Then, based on the derived theoretical results and the technique of the Lagrange multi-

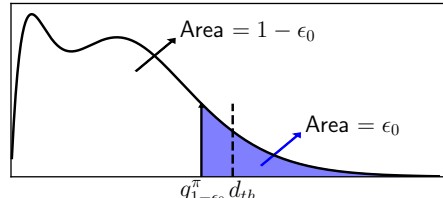

Figure 2: Equivalence between the outage probability constraint and the quantile constraint

plier, we construct our algorithm named Quantile Constrained Policy Optimization (QCPO) to solve the outage probability constrained RL problem (in Section 4). Here, we use distributional RL with the Large Deviations Principle (LDP) to estimate quantiles and tail probability of the cumulative sum cost for implementation of QCPO. The implemented algorithm satisfies the outage probability constraint after the training period. To the best of our knowledge, this is the first work that tackles the issue of applying the policy gradient theorem to the quantile and obtains an (approximate) policy gradient for the quantile, and this is one of the main contributions of this paper, together with the QCPO algorithm.

## 2  Background and Related Works

**Constrained RL**  A constrained Markov decision process (CMDP) is defined as a tuple $\langle \mathcal{S}, \mathcal{A}, r, c, M, \gamma \rangle$, where $\mathcal{S}$ is the state space, $\mathcal{A}$ is the action space, $r : \mathcal{S} \times \mathcal{A} \to \mathbb{R}$ is the reward function, $c : \mathcal{S} \times \mathcal{A} \to \mathbb{R}_{\geq 0}$ is the cost function, $M : \mathcal{S} \times \mathcal{A} \times \mathcal{S} \to [0, 1]$ is the state transition probability, and $\gamma$ is the discount factor.

Constrained RL or Safe RL is an area of RL whose objective is to find an optimal policy for a given CMDP that maximizes the expected return $\mathbb{E}_\pi \left[ \sum_{t=0}^\infty \gamma^t r(s_t, a_t) \right]$ while satisfying a constraint on the cumulative sum cost $\sum_{t=0}^\infty \gamma^t c(s_t, a_t)$. If one wants to constrain the average cumulative sum cost of the policy $\pi$, then constraint $C^\pi(s_0) := \mathbb{E}_\pi \left[ \sum_{t=0}^\infty \gamma^t c(s_t, a_t) \right] \leq d_{th}$ can be considered. On the other hand, if one wants to constrain the outage behavior of the policy $\pi$, then constraint $\Pr\left( X^\pi(s_0) > d_{th} \right) \leq \epsilon_0$ should be considered, where $X^\pi(s_0)$ is a random variable defined as $X^\pi(s_0) := \sum_{t=0}^\infty \gamma^t c(S_t, A_t)$ with $S_0 = s_0$, $A_t \sim \pi(\cdot | S_t)$, $S_{t+1} \sim M(\cdot | S_t, A_t)$ for $t = 0, 1, \cdots$.

Most of the previous constrained RL works considered a constraint on the expectation of the cumulative sum cost: $C^\pi(s_0) \leq d_{th}$ [1, 12, 14, 24, 25, 28, 29]. In order to solve the constrained optimization problem with this expectation-based constraint, researchers considered the Lagrangian multiplier method [1, 21], Lyapunov-based methods [6, 7], projection-based methods [28], safety-layer methods [10].

Table 1: Comparison of constrained RL with probabilistic constraints

| Papers | Algorithm Type | Distribution Modeling | Theory | Deep RL |
|---|---|---|---|---|
| Risk-Const. RL [5] | On-policy (Trajectory-based) | No (need only trajectory samples) | Yes | No |
| WCSAC [27] | Off-policy | Gaussian (on all range of distribution) | No | Yes |
| QCPO (This paper) | On-policy (State-based) | LDP with Weibull (only on tail) | Yes | Yes |

*Quantile (i.e., Value at Risk, VaR)* and *Conditional Value at Risk (CVaR)* are two well-known techniques to manage undesirable events in the domain of finance[18]. In the context of RL, the definitions of the quantile and the CVaR for the distribution of the cumulative sum cost for a given $\pi$ are given by $q_u^\pi(s_0) := \inf\{x \mid \Pr(X^\pi(s_0) \leq x) \geq u\}$ and $\text{CVaR}_u^\pi(s_0) := \mathbb{E}_\pi \left[ X^\pi(s_0) \mid X^\pi(s_0) \geq q_u^\pi(s_0) \right]$, respectively.

The CVaR was previously used in RL to constrain undesirable events and the problem with a CVaR constraint is explicitly formulated as

$$\begin{aligned} \text{Maximize} \quad & \mathbb{E}_\pi \left[ \sum_{t=0}^\infty \gamma^t r(s_t, a_t) \right] \\ \text{Subject to} \quad & \text{CVaR}_{1-\epsilon_0}^\pi(s_0) \leq d_{th}. \end{aligned} \qquad \text{(CVaR-CP)}$$

Note that the $(1 - \epsilon_0)$-CVaR denoted as $\text{CVaR}_{1-\epsilon_0}^\pi(s)$ is always greater than or equal to the $(1 - \epsilon_0)$-quantile denoted as $q_{1-\epsilon_0}^\pi(s)$ for all $s \in \mathcal{S}$ because of the definition of the CVaR. Therefore, satisfying the CVaR constraint $\text{CVaR}_{1-\epsilon_0}^\pi(s_0) \leq d_{th}$ in (CVaR-CP) is a sufficient condition for satisfying the probabilistic constraint $\Pr\left( X^\pi(s_0) > d_{th} \right) \leq \epsilon_0$ in (ProbCP), and hence (CVaR-CP) is a stricter problem than (ProbCP). Therefore, algorithms proposed to solve (CVaR-CP) can be used for solving (ProbCP), and this should satisfy the probabilistic constraint in theory.

Chow et al. [5] proposed a trajectory-based CVaR method and provided convergence for their method. They used trajectory-based policy gradient to their Lagrangian, and it is simple to compute. Like most trajectory-based RL algorithms, however, it suffers from sample inefficiency since it collects a number of trajectories and updates its parameter once. Recently, Yang et al. [27] proposed an off-policy algorithm to solve (CVaR-CP). They only estimated the mean and variance of the cost distribution using a technique in distributional RL and computed $\text{CVaR}_u^\pi(s)^2$ as the CVaR of the Gaussian distribution of the estimated mean and variance. However, the distribution of $X^\pi(s)$ is not Gaussian in general, and the Gaussian approximation has limited capability to capture the decay rate of the tail probability because a Gaussian probability density function (PDF) has the form of $\exp(-\beta x^2)$ with fixed rate function $x^2$. Therefore, this algorithm can yield a poor estimation

---

[2]They actually consider the CVaR $\text{CVaR}_u^\pi(s, a)$ of the cumulative sum cost $X^\pi(s, a)$ for a given $(s, a)$ pair.

of the CVaR of the tail, especially for small tail probability, and cannot guarantee to satisfy the CVaR constraint (see Section 5). Furthermore, note that the CVaR and the quantile are two different measures for undesirable events, and the choice between the two depends on what we desire. For example, an insurance company prefers the CVaR of undesirable events to determine an insurance premium. On the other hand, a company developing an autonomous driving car system needs the quantile of undesirable events to guarantee the accident probability for safety. Thus, in the context of safe learning, our work focuses directly on *the constraint on the quantile*, which is an equivalent (i.e., necessary and sufficient) constraint to the outage probability constraint in (ProbCP). To the best of our knowledge, this is the first work that provides a state-based policy gradient for the quantile and required theoretical results regarding the quantile-constrained RL problem. Moreover, our implementation approximates the tail distribution of $X^\pi(s)$ with a Weibull distribution (a particular case of generalized Gamma distribution), which is general enough to capture various rates of decay of the tail probability. Table 1 summarizes the previous constrained RL methods and this paper.

**Large Deviation Principle (LDP)** Large deviation principle (LDP) [11] is a technique for estimating the limiting behavior of a sequence of distributions, especially on the tail. A simple example is the empirical mean $\bar{X}_n = \frac{1}{n}\sum_{k=1}^{n} X_k$ of i.i.d. random variables $X_i$. We say that a sequence $\{\bar{X}_n\}$ satisfies LDP if the sequence of its log probability distribution $\frac{1}{n}\log \Pr\left(\bar{X}_n \in \Gamma\right)$ satisfies the following condition $\frac{1}{n}\log \Pr\left(\bar{X}_n \in \Gamma\right) \overset{n\to\infty}{\longrightarrow} -\inf_{x\in\Gamma} I(x)$ for some function $I(x)$. The function $I(x)$ satisfying such limiting behavior is called the rate function of $\bar{X}_n$. The rate function $I(x)$ is also related to the cumulative distribution function (CDF) $F_{\bar{X}_n}(x)$ since $1 - F_{\bar{X}_n}(x_0) = \Pr\left(\bar{X}_n \in [x_0, \infty)\right) \approx \exp\left(-n\inf_{x\in[x_0,\infty)} I(x)\right)$ for some $x_0 > \mathbb{E}[X]$ and sufficiently large $n$. LDP can be applied to finite-state Markov chains, and there exists a rate function for a given Markov chain [11]. In this paper, we consider the tail probability of the distribution of the cumulative sum cost $X^\pi(s_0) = \sum_{t=0}^{\infty} \gamma^t c(s_t, a_t)$. Finding its analytic rate function is hard. Therefore, we instead approximate the rate function directly as $I_{X^\pi(s)}(x) \approx (x/\beta(s))^{\alpha(s)}$ with learnable parameters $\alpha(s)$ and $\beta(s)$, which results in a Weibull distribution: $1 - F_{X^\pi(s)}(x) = \exp\left\{-(x/\beta(s))^{\alpha(s)}\right\}$. We use this distribution to approximate the tail probability of $p_{X^\pi(s)}(x)$ of $X^\pi(s)$.

## 3 Quantile Constrained RL

In this section, we explain an equivalent form of (ProbCP) that we use to learn an optimal constrained policy under the outage probability constraint and then explain the difficulty of applying the policy gradient theorem to optimize the Lagrangian of the equivalent problem. Finally, we provide theoretical results that circumvent this difficulty in Section 3.2.

### 3.1 Motivation: Problem of Applying Policy Gradient Theorem to Quantile

Solving (ProbCP) with a direct approach is too hard in making a loss function for $\pi$ based on the outage probability. Thus, we convert the probability constrained problem to an equivalent form of a quantile constrained problem:

$$\begin{aligned} \text{Maximize} \quad & \mathbb{E}_\pi\left[\sum_{t=0}^{\infty} \gamma^t r(s_t, a_t)\right] \\ \text{Subject to} \quad & q_{1-\epsilon_0}^\pi(s_0) \leq d_{th}, \end{aligned} \quad\quad \text{(QuantCP)}$$

where $q_u^\pi(s) = \inf\{x \mid \Pr(X^\pi(s) \leq x) \geq u\}$ is the $u$-quantile of the random variable $X^\pi(s)$ of the cumulative sum cost: $X^\pi(s) = \sum_{t=0}^{\infty} \gamma^t c(S_t, A_t)$ with $S_0 = s$, $A_t \sim \pi(\cdot|S_t)$, $S_{t+1} \sim M(\cdot|S_t, A_t)$, $t = 0, 1, 2, \cdots$. Note that $q_{1-\epsilon_0}^\pi(s_0) \leq d_{th}$ is equivalent to $\Pr\left[X^\pi(s_0) > d_{th}\right] \leq \epsilon_0$ due to the definition of the quantile. We propose a direct approach to solve the equivalent problem (QuantCP) instead of (ProbCP). Although we have an equivalent form of (ProbCP), it is still difficult to solve (QuantCP). We explain what makes solving the problem (QuantCP) still hard below.

In the case of (ExpCP), the Lagrange-based optimization of (ExpCP) is given by $\min_{\lambda\geq 0}\max_\pi L_{exp}(\pi, \lambda) := V^\pi(s_0) - \lambda\left(C^\pi(s_0) - d_{th}\right)$, where $V^\pi(s_0) = \mathbb{E}_\pi\left[\sum_{t=0}^{\infty} \gamma^t r(s_t, a_t)\right]$ and $C^\pi(s_0) = \mathbb{E}_\pi\left[\sum_{t=0}^{\infty} \gamma^t c(s_t, a_t)\right]$. Then, due to the form of $C^\pi(s_0)$, the policy gradient theorem [22] can directly be applied, and the gradient of the Lagrangian w.r.t. $\pi$ is given by the expectation form:

$$\nabla_\pi L_{exp}(\pi, \lambda) = \sum_s \rho^\pi(s) \sum_a \nabla\pi(a|s)\left\{A_r^\pi(s, a) - \lambda A_c^\pi(s, a)\right\}, \quad\quad (2)$$

where $\rho^\pi(s) := \sum_{t=0}^\infty \gamma^t \Pr(S_t = s | s_0, \pi)$, $A_r^\pi(s, a) := r(s, a) + \gamma \mathbb{E}_{s' \sim M} [V^\pi(s')] - V^\pi(s)$, and $A_c^\pi(s, a) := c(s, a) + \gamma \mathbb{E}_{s' \sim M} [C^\pi(s')] - C^\pi(s)$. However, the gradient of the Lagrangian of the problem (QuantCP)

$$\min_{\lambda \geq 0} \max_\pi L_{quant}(\pi, \lambda) := V^\pi(s_0) - \lambda \left( q_{1-\epsilon_0}^\pi(s_0) - d_{th} \right) \tag{3}$$

w.r.t. the policy $\pi$ cannot be expressed as an expectation form:

$$\nabla_\pi L_{quant}(\pi, \lambda) \neq \mathbb{E}_\pi \left[ \nabla \log \pi(a|s) \left\{ A_r^\pi(s, a) - \lambda \bar{A}_{1-\epsilon_0}^\pi(s, a) \right\} \right] \tag{4}$$

where $\bar{A}_u^\pi(s, a) := c(s, a) + \gamma \mathbb{E}_{s' \sim M} [q_u^\pi(s')] - q_u^\pi(s)$. This is because the $u$-quantile $q_u^\pi(s)$ is not the expectation of the cumulative sum cost. However, if the $u$-quantile $q_u^\pi(s)$ can be written as

$$q_u^\pi(s_0) = \mathbb{E}_\pi \left[ \sum_{t=0}^\infty \gamma^t \left\{ c(s_t, a_t) + \tilde{c}_u(s_t, a_t) \right\} \right] \tag{5}$$

for some function $\tilde{c}_u(s, a)$ that is independent of the policy $\pi$, we can apply the policy gradient theorem by defining the advantage function for the quantile term:

$$A_u^\pi(s, a) := c(s, a) + \tilde{c}_u(s, a) + \gamma \mathbb{E}_{s' \sim M} [q_u^\pi(s')] - q_u^\pi(s). \tag{6}$$

This fact motivates us to search for such $\tilde{c}_u(s, a)$. For this, under mild assumptions, we first show the existence of a policy-dependent additional cost $\tilde{c}_u^\pi(s, a)$ and then show that the additional cost can be approximated by a cost $\tilde{c}_u^{\pi'}(s, a)$ for some fixed $\pi'$ independent of $\pi$ except the requirement $\max_s \mathrm{KL}(\pi'(\cdot|s) \| \pi(\cdot|s)) \leq \delta$.

### 3.2 Theoretical Results

We here provide theoretical results showing the existence of an additional cost $\tilde{c}_u^\pi(s, a)$ and showing that this can be approximated as another cost $\tilde{c}_u^{\pi'}(s, a)$ for a base policy $\pi'$ independent of $\pi$, only requiring $\max_s \mathrm{KL} \left( \pi'(\cdot|s) \| \pi(\cdot|s) \right) \leq \delta$ for some $\delta > 0$. These theoretical results make the quantile constrained policy optimization tractable by enabling application of the policy gradient theorem. For the theoretical results, we assume that the CDF $F_{X^\pi(s)}(x)$ is strictly increasing on $[0, \infty)$, and it is continuously differentiable for all $s \in \mathcal{S}$. The proofs of the theoretical results are in Appendix B.

We begin with deriving the temporal-difference (TD) relation between the $u$-quantiles of $X^\pi(s)$ at $s_t$ and $s_{t+1}$. Theorem 1 states the TD relation for the $u$-quantile under the following assumptions of boundness of quantile difference and smoothness of CDF of $X^\pi(s)$.

**Assumption 1** (Boundness of quantile difference). *For a given policy $\pi$, the following two quantities are bounded*

$$|c(s, a) + \gamma q_u^\pi(s') - q_u^\pi(s)| \leq \gamma R \tag{7}$$

$$\left| q_u^\pi(s) - F_{X^\pi(s)}^{-1} \left( F_{X^\pi(s')} \left( \frac{q_u^\pi(s) - c(s, a)}{\gamma} \right) \right) \right| \leq R \tag{8}$$

*for all $(s, a, s') \in \mathcal{S} \times \mathcal{A} \times \mathcal{S}$ such that $\pi(a|s) \cdot M(s'|s, a) > 0$.*

**Assumption 2** (Smoothness of CDF of $X^\pi(s)$). *For each state $s$, the average slope of $F_{X^\pi(s)}(x)$ between $q_u^\pi(s)$ and $y \in [q_u^\pi(s) - R, q_u^\pi(s) + R]$ is bounded by*

$$\frac{1}{1 + \epsilon} \cdot p_{X^\pi(s)} (q_u^\pi(s)) \leq \frac{F_{X^\pi(s)} (q_u^\pi(s)) - F_{X^\pi(s)} (y)}{q_u^\pi(s) - y} \leq \frac{1}{1 - \epsilon} \cdot p_{X^\pi(s)} (q_u^\pi(s)) \tag{9}$$

*for small $0 < \epsilon < \frac{1}{2}$.*

**Theorem 1.** *Under Assumptions 1 and 2, the $u$-quantile of the random variable $X^\pi(s_t)$ satisfies the following temporal-difference (TD) relation. For some constant $R$ and small $\epsilon > 0$,*

$$\left| \mathbb{E}_\pi \left[ \mu_u^\pi (s_t, a_t, s_{t+1}) \left\{ c(s_t, a_t) + \gamma q_u^\pi(s_{t+1}) - q_u^\pi(s_t) \right\} \right] \right| \leq \frac{\epsilon}{1 - \epsilon} R, \tag{10}$$

*where $\mu_u^\pi (s, a, s') = p_{X^\pi(s')} \left( \frac{q_u^\pi(s) - c(s, a)}{\gamma} \right) \big/ \gamma p_{X^\pi(s)} (q_u^\pi(s))$. Here, the expectation is for the action $a_t \sim \pi(\cdot|s_t)$ and the next state $s_{t+1} \sim M(\cdot|s_t, a_t)$. ($s_t$ is given.)*

Note that for the expectation of the cumulative sum cost $C^\pi(s)$ considered in (ExpCP), the expectation of TD under the policy $\pi$ follows

$$\mathbb{E}_\pi \left[ c(s_t, a_t) + \gamma C^\pi(s_{t+1}) - C^\pi(s_t) \right] = 0 \tag{11}$$

by the Bellman equation. The TD relation (11) for expectation has a similar form to that for the $u$-quantile (10), but the difference is that (10) is the weighted expectation of the TD ($c(s_t, a_t) + \gamma q_u^\pi(s_{t+1}) - q_u^\pi(s_t)$). The numerator $p_{X^\pi(s_{t+1})}\left(\frac{q_u^\pi(s_t) - c(s_t, a_t)}{\gamma}\right)$ of the weight $\mu_u^\pi(s_t, a_t, s_{t+1})$ in (10) involves two quantities: 1) a target quantile $\frac{q_u^\pi(s_t) - c(s_t, a_t)}{\gamma}$ and 2) the PDF of the sum of costs $X^\pi(s_{t+1}) = \sum_{k=0}^\infty \gamma^k c(s_{t+k+1}, a_{t+k+1})$ starting from state $s_{t+1}$. Here, the value $\frac{q_u^\pi(s_t) - c(s_t, a_t)}{\gamma}$ is the target value of the sum of costs $\sum_{k=0}^\infty \gamma^k c(s_{t+k+1}, a_{t+k+1})$ from the next state $s_{t+1}$ such that the sum of costs $\sum_{k=0}^\infty \gamma^k c(s_{t+k}, a_{t+k})$ for a given pair $(s_t, a_t)$ at $t$ is the $u$-quantile $q_u^\pi(s_t)$. Thus, the numerator $p_{X^\pi(s_{t+1})}\left(\frac{q_u^\pi(s_t) - c(s_t, a_t)}{\gamma}\right)$ of the weight $\mu_u^\pi(s_t, a_t, s_{t+1})$ in (10) is the probability of the event that the cumulative sum cost starting from $(s_t, a_t)$ becomes the $u$-quantile $q_u^\pi(s_t)$ at $s_t$ from the perspective of the next state $s_{t+1}$. Based on Theorem 1, we obtain the following corollary:

**Corollary 1.** *Under Assumptions 1 and 2, the $u$-quantile $q_u^\pi(s_t)$ of the random variable $X^\pi(s_t)$ is bounded as*

$$\left| q_u^\pi(s_t) - \mathbb{E}_\pi \left[ \mu_u^\pi(s_t, a_t, s_{t+1}) \left\{ c(s_t, a_t) + \gamma q_u^\pi(s_{t+1}) \right\} \right] \right| \le \frac{\epsilon}{1 - \epsilon} R. \tag{12}$$

*Proof:* Note that the term $q_u^\pi(s_t)$ can go outside the expectation in (10) since the expectation is over $(a_t, s_{t+1})$. From eq. (26) in Appendix A.1, the expectation of the numerator of $\mu_u^\pi(s_t, a_t, s_{t+1})$ is the same as the the denominator of the weight, i.e., $\mathbb{E}_\pi \left[ p_{X^\pi(s_{t+1})}\left(\frac{q_u^\pi(s_t) - c(s_t, a_t)}{\gamma}\right) \right] = \gamma p_{X^\pi(s_t)}(q_u^\pi(s_t))$, and this leads to $\mathbb{E}_\pi[\mu_u^\pi(s_t, a_t, s_{t+1})] = 1$. So, we have the claim. $\square$

As seen in Corollary 1, the $u$-quantile at $s_t$ can be approximated as a weighted expectation of $c(s_t, a_t) + \gamma q_u^\pi(s_{t+1})$, and the weight is proportional to $p_{X^\pi(s_{t+1})}\left(\frac{q_u^\pi(s_t) - c(s_t, a_t)}{\gamma}\right)$. This means that the more probable is the pair $(s_t, a_t, s_{t+1})$ to achieve $q_u^\pi(s_t)$, the higher weight is multiplied to $c(s_t, a_t) + \gamma q_u^\pi(s_{t+1})$ for approximating $q_u^\pi(s_t)$. Furthermore, if we assume that the transition dynamics of CMDP are deterministic, i.e., $s_{t+1} = h(s_t, a_t)$ as in many real-world control problem, we can approximate the $u$-quantile $q_u^\pi(s_0)$ at $s_0$ as the expectation of the sum of costs under a distorted policy $\tilde{\pi}_u$, as stated in the following lemma:

**Lemma 1.** *Suppose that the state transition dynamics are deterministic, i.e., $s_{t+1} = h(s_t, a_t)$. Then, under Assumptions 1 and 2, the $u$-quantile $q_u^\pi(s_0)$ of the random variable $X^\pi(s_0)$ is expressed as*

$$\left| q_u^\pi(s_0) - \mathbb{E}_{\tilde{\pi}_u} \left[ \sum_{t=0}^\infty \gamma^t c(s_t, a_t) \right] \right| \le \frac{\epsilon R}{(1 - \epsilon)(1 - \gamma)}, \tag{13}$$

*where $\tilde{\pi}_u(a|s) = \pi(a|s) \cdot \mu_u^\pi(s, a, h(s, a)) \propto \pi(a|s) \cdot p_{X^\pi(h(s,a))}\left(\frac{q_u^\pi(s) - c(s, a)}{\gamma}\right)$.*

Now, plugging (13) into the quantile term in the Lagrangian (3) of the problem (QuantCP), we may apply the policy gradient theorem based on the chain rule since the $u$-quantile $q_u^\pi(s_0)$ is expressed as the expectation of the sum of costs. However, the gradient of $\tilde{\pi}_u$ w.r.t. $\pi$ for chain rule is too complicated due to the $\mu_u^\pi$ term in Lemma 1. Thus, we find another expectation form of $q_u^\pi(s_0)$ using an additional cost function $\tilde{c}_u^\pi(s, a)$, as stated in the following theorem:

**Theorem 2.** *Under deterministic dynamics $s_{t+1} = h(s_t, a_t)$ and Assumptions 1 and 2, $q_u^\pi(s)$ can be expressed as*

$$\left| q_u^\pi(s_0) - \mathbb{E}_\pi \left[ \sum_{t=0}^\infty \gamma^t \left\{ c(s_t, a_t) + \tilde{c}_u^\pi(s_t, a_t) \right\} \right] \right| \le \frac{\epsilon R}{(1 - \epsilon)(1 - \gamma)}, \tag{14}$$

*where $\tilde{c}_u^\pi(s, a) = (\mu_u^\pi(s, a, h(s, a)) - 1) \cdot [c(s, a) + \gamma q_u^\pi(h(s, a))]$.*

Note that the additional cost $\tilde{c}_u^\pi(s_t, a_t)$ in Theorem 2 is a policy-dependent cost function. Under an additional mild assumption, we can find an upper bound of (14) which replaces the policy-dependent cost function $\tilde{c}_u^\pi(s, a)$ with another cost $\tilde{c}_u^{\pi'}(s, a)$ for some fixed $\pi'$ independent of $\pi$, only requiring $\max_s \text{KL}\left(\pi'(\cdot|s) \| \pi(\cdot|s)\right) \le \delta$ for some $\delta > 0$. The additional assumption is as follows:

**Assumption 3** (Lipschitz continuity of $\tilde{c}_u^\pi(s,a)$ over $\pi$). *For any given fixed $u \in (0,1)$ and any policies $\pi$ and $\pi'$, there exists a coefficient $C_u$ such that*

$$\left| \tilde{c}_u^{\pi'}(s,a) - \tilde{c}_u^\pi(s,a) \right| \le C_u \cdot \max_{s'} KL\left(\pi'(\cdot|s') \,\|\, \pi(\cdot|s')\right), \quad \forall s \in \mathcal{S}, a \in \mathcal{A}. \tag{15}$$

Basically, Assumption 3 is that the function $\tilde{c}_u^\pi$ as a function of $\pi$ is continuous, which is expected to be satisfied if there is no abrupt change in the associated distributions. With Assumption 3 and Theorem 2, we obtain an expression for the quantile $q_u^\pi(s_0)$ as a form of desired expected sum:

**Theorem 3.** *Under deterministic dynamics $s_{t+1} = h(s_t, a_t)$ and Assumptions 1, 2, and 3, the $u$-quantile $q_u^\pi(s_0)$ is expressed as the expectation of the sum of actual cost and a $\pi$-independent additional cost $\tilde{c}_u^{\pi'}(s,a)$ for $\pi'$ satisfying $\max_s KL(\pi'(\cdot|s) \,\|\, \pi(\cdot|s)) \le \delta$:*

$$\left| q_u^\pi(s_0) - \mathbb{E}_\pi \left[ \sum_{t=0}^\infty \gamma^t \left\{ c(s_t, a_t) + \tilde{c}_u^{\pi'}(s_t, a_t) \right\} \right] \right| \le \frac{\epsilon R}{(1-\epsilon)(1-\gamma)} + \frac{C_u}{1-\gamma}\delta. \tag{16}$$

By Theorem 3, we can approximate the $u$-quantile $q_u^\pi(s_0)$ as the expectation of the sum of costs plus $\pi$-independent additional costs $\tilde{c}_u^{\pi'}(s,a)$ for a base policy $\pi'$, and this approximation is tighter when the distance between the current policy $\pi$ and the base policy $\pi'$ is smaller. Theorem 3 can be interpreted the other way around. As in the case of PPO [20], if we first simply set $\pi'$ as the policy before the update, denoted as $\pi_{old}$, then the updated $\pi$ is near from the base policy $\pi_{old} = \pi'$, and we can compute the corresponding KL distance between $\pi_{old}$ (= $\pi'$) and $\pi$. Then, still, the inequality (16) holds for $\delta = \max_s KL\left(\pi_{old}(\cdot|s)\|\pi(\cdot|s)\right)$. Now, this result enables us to solve the quantile constrained problem (QuantCP) by applying the policy gradient theorem.

## 4 Quantile Constrained Policy Optimization

Using the theoretical results in Section 3, we now construct an algorithm named quantile constrained policy optimization (QCPO) to solve (QuantCP) based on an on-policy RL algorithm: PPO [20]. The QCPO is a direct method to constrain the outage probability and consists of three parts: 1) estimation of the $u$-quantile $q_u^\pi(s)$ of $X^\pi(s)$ for a given policy $\pi$, 2) estimation of PDF of $X^\pi(s)$ to compute the additional cost $\tilde{c}_u^{\pi'}(s,a)$, and 3) updating method of the Lagrange multiplier to control the outage probability. We first explain the overall structure of QCPO and the base loss function for the policy, which has a similar form to that in [19]. Then, we provide a condition for policy improvement for the proposed method. The implementation of the proposed algorithm is based on the implementation of [21], and details of the implementation, including the network structure, the loss functions, the Lagrangian multiplier update method, and the hyper-parameters, are in Appendix E. The implementation code of QCPO is available at github.com/wyjung0625/QCPO.

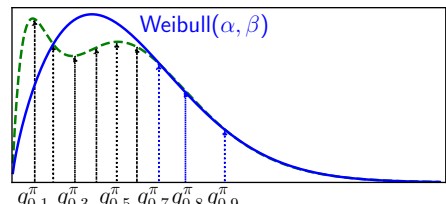

Figure 3: Illustration of the quantile approximation and tail-probability approximation. Green-dash curve is the unknown PDF $p_{X^\pi(s)}(x)$ and up-arrow points are the estimated $u$-quantiles $q_u^\pi(s)$. We approximate the PDF $p_{X^\pi(s)}(x)$ on the right tail by a Weibull distribution (blue curve) and this is approximated using the 4 rightmost quantile points (blue arrow).

### 4.1 Overall Structure of QCPO

The agent of QCPO uses function approximators for the policy $\pi$, the value function $V^\pi(s) = \mathbb{E}_\pi \left[ \sum_{t=0}^\infty \gamma^t r(s_t, a_t) \right]$ and the quantile function $q^\pi(s) = \left[ q_{u_1}^\pi(s), q_{u_2}^\pi(s), \dots, q_{u_{n_q}}^\pi(s) \right]$ of $X^\pi(s) = \sum_{t=0}^\infty \gamma^t c(s_t, a_t)$ for the policy $\pi$. These functions are parameterized by deep neural networks with parameters $\theta$, $\phi$, and $\psi$, respectively. We denote $\theta_{old}$, $\phi_{old}$, and $\psi_{old}$ as their old parameters. Note that the quantile function $q_\psi(s)$ outputs $n_q$ values, and the $i$-th value represents an estimate of the $u_i$-quantile $q_{u_i}^\pi(s)$ of $X^\pi(s)$ for fixed target CDF values $\left[u_1, u_2, \dots, u_{n_q}\right]$ with $u_1 < u_2 < \cdots < u_{n_q}$.

In addition to these parameterized functions, we need another function that approximates the PDF of $X^\pi(s)$ on the right tail. (Here, the right-tail probability most matters since the target outage probability is typically small.)

As aforementioned in Section 2, we know that $X^\pi(s)$ follows LDP with a rate function $I_{X^\pi(s)}(x)$. However, finding the rate function $I_{X^\pi(s)}(x)$ in an analytic approach is hard. Therefore, in QCPO, the agent approximates the rate function of the form of $I_{X^\pi(s)}(x) \approx (x/\beta(s))^{\alpha(s)}$ and learns the state-dependent parameters $\alpha(s)$ and $\beta(s)$ by using the quantiles on the right tail approximated by its quantile function for the right tail: $q^\pi_{u_{n_q-k+1}}(s), \ldots, q^\pi_{u_{n_q}}(s)$. This approximation of the rate function results in approximation probability on the right tail as a Weibull distribution, whose tail distribution is $1 - F_{X^\pi(s)}(x) = e^{-(x/\beta(s))^{\alpha(s)}}$. In order to obtain the state-dependent parameters $\alpha(s)$ and $\beta(s)$ for right-tail distribution approximation, we again parameterize them by neural networks with parameters $\xi$ and $\zeta$, respectively. Fig. 3 shows both the quantile approximation and right-tail approximation of QCPO. Note that our approach actually learns the rate function governing the tail-probability decay rate, whereas the previous Gaussian approximation [27] on the PDF of $X^\pi(s)$ fixes the rate function as quadratic $x^2$, which is not the correct rate function in general.

The overall procedure of QCPO is as follows: 1) estimate the value function $V^\pi(s)$ for return and estimate the quantile function $q^\pi_u(s)$, $u \in [u_1, u_2, \ldots, u_{n_q}]$ for the cumulative sum cost, 2) approximate tail distribution $p_{X^\pi(s)}(x)$ on the right tail using a Weibull distribution with parameters $\alpha(s)$, $\beta(s)$, 3) compute the additional cost for the base policy $\pi'$, $\tilde{c}^{\pi'}_{1-\epsilon_0}(s, a)$ for the quantile advantage $A^\pi_{1-\epsilon_0}(s, a) := c(s, a) + \tilde{c}^{\pi'}_{1-\epsilon_0}(s, a) + \gamma q^\pi_{1-\epsilon_0}(s') - q^\pi_{1-\epsilon_0}(s)$ , 4) take policy gradient using the sum of the value advantage and the quantile advantage $A_r(s, a) - \lambda A^\pi_{1-\epsilon_0}(s, a)$, 5) update the Lagrange multiplier $\lambda$. Since QCPO is based on PPO [20], the loss functions for the policy and the value function are similar to those of PPO [20]. Please see Appendix E.2 and E.3 for detail.

## 4.2 Policy Loss Function and Policy Improvement Condition

Let us consider the policy loss function of QCPO to solve (QuantCP). The basic loss function of QCPO for a given Lagrange multiplier $\lambda$ is given by

$$L^{\pi_{old}}(\pi_\theta) - \tilde{C}_1 \max_s \text{KL}(\pi_{old}(\cdot|s) \| \pi_\theta(\cdot|s)) \tag{17}$$

where

$$L^{\pi_{old}}(\pi_\theta) = \left(V^{\pi_{old}}(s_0) - \lambda q^{\pi_{old}}_{1-\epsilon_0}(s_0)\right) + \mathbb{E}_{s\sim\rho^{\pi_{old}}, a\sim\pi_\theta}\left[A^{\pi_{old}}_r(s_t, a) - \lambda A^{\pi_{old}}_{1-\epsilon_0}(s_t, a)\right] \tag{18}$$

$$A^{\pi_{old}}_r(s, a) = r(s, a) + \gamma\mathbb{E}_{s'\sim M(\cdot|s,a)}\left[V^{\pi_{old}}(s')\right] - V^{\pi_{old}}(s) \tag{19}$$

$$A^{\pi_{old}}_{1-\epsilon_0}(s, a) = c(s, a) + \tilde{c}^{\pi_{old}}_{1-\epsilon_0}(s, a) + \gamma\mathbb{E}_{s'\sim M(\cdot|s,a)}\left[q^{\pi_{old}}_{1-\epsilon_0}(s')\right] - q^{\pi_{old}}_{1-\epsilon_0}(s), \tag{20}$$

and $\pi_{old} := \pi_{\theta_{old}}$ is the policy that collects the most recent batch of samples, $\rho^{\pi_{old}}(s) := \sum_{t=0}^\infty \gamma^t \Pr(S_t = s|s_0, \pi_{old})$ is the stationary state distribution under $\pi_{old}$, $\tilde{C}_1$ is a constant (see Appendix B.5), Now, we consider the relationship between the actually-desired maximization objective $L_{quant}(\pi, \lambda) = V^\pi(s_0) - \lambda\left(q^\pi_{1-\epsilon_0}(s_0) - d_{th}\right)$ in (3) and the practical QCPO objective $L^{\pi_{old}}(\theta) - \tilde{C}_1 \max_s \text{KL}(\pi_{old}(\cdot|s) \| \pi_\theta(\cdot|s))$ in (17). The relationship between the two is given by the following theorem.

**Theorem 4.** *Let $\pi_{new} := \pi_{\theta_{new}}$ be the solution of the problem of maximizing*

$$L^{\pi_{old}}(\pi_\theta) - \tilde{C}_1 \max_s KL(\pi_{old}(\cdot|s) \| \pi_\theta(\cdot|s)) \tag{21}$$

*for some constant $\tilde{C}_1 > 0$. Then, under deterministic dynamics $s_{t+1} = h(s_t, a_t)$ and Assumptions 1, 2, and 3, the following inequality holds:*

$$L_{quant}(\pi_{new}, \lambda) - L_{quant}(\pi_{old}, \lambda) \tag{22}$$

$$\geq L^{\pi_{old}}(\pi_{new}) - L^{\pi_{old}}(\pi_{old}) - \tilde{C}_1 KL_{max}(\pi_{old}\|\pi_{new}) - \underbrace{\tilde{C}_2\frac{\epsilon}{1-\epsilon}}_{\text{approximation loss}} \tag{23}$$

*for a given Lagrange multiplier $\lambda > 0$, some constant $\tilde{C}_2$ and small $\epsilon > 0$.*

Note that the term $\tilde{C}_2\frac{\epsilon}{1-\epsilon}$ in (23) is due to our approximation of the quantile as an expected sum to apply policy gradient. Therefore, by Theorem 4, when the improvement $L^{\pi_{old}}(\pi_{new}) - L^{\pi_{old}}(\pi_{old}) - \tilde{C}_1\text{KL}_{max}(\pi_{old}\|\pi_{new})$ $(> 0)$ by the policy update from the QCPO loss function is large enough to compensate for the approximation loss, the desired quantity will also be improved by our policy update. That is, the Lagrangian for the quantile constrained problem for $\pi_{new}$ will be higher than that for $\pi_{old}$.

## 5  Experiments

### 5.1  Environments

We examined the performance of the proposed QCPO and compared it to that of WCSAC,

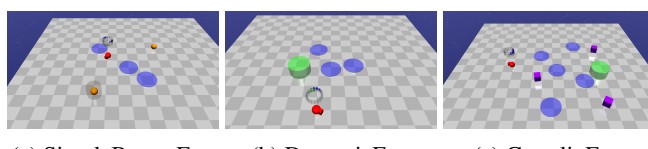

(a) SimpleButtonEnv    (b) DynamicEnv    (c) GremlinEnv

Figure 4: The considered environments

which uses the CVaR constraint. The environments we considered are SimpleButtonEnv, DynamicEnv [27], and GremlinEnv, which are based on Safety Gym [17], MuJoCo [23], and OpenAI Gym [4]. The environments can be considered as simplified versions of a real environment of an automatic serving robot and are illustrated in Fig. 4. The goal of these environments is for a robot (red sphere) to reach a goal (orange sphere wrapped by a grey translucent pillar, or green pillar) while avoiding the non-goal button (orange sphere), hazards (blue circle) or moving gremlins (purple box). Once the robot reaches the current goal, the environments generate the next goal deterministically (SimpleButtonEnv) or randomly (DynamicEnv, GremlinEnv), so the task complexity increases in the order of SimpleButtonEnv, DynamicEnv, and GremlinEnv. When the robot performs an action at time $t$, it receives a reward $\{\|p_{t+1} - p_{\text{goal}}\|_2 - \|p_t - p_{\text{goal}}\|_2\} + 1_{\text{goal reached}}$, where $p_t$ is the position $(x, y)$ of the robot at time $t$ and $p_{\text{goal}}$ is the current goal position at time $t$. It also receives a cost $+1$ if the robot touches a non-goal object (the non-goal button, a hazard, or a gremlin) and $0$ otherwise. Thus, for the robot, it receives a higher return when the robot touches more goals in maximum timesteps $T = 1000$, and a higher sum of costs when the robot touches one of the other objects more often. A more detailed explanation of the environments is in Appendix C.

### 5.2  Empirical Results

We compared the performance of the proposed algorithm (QCPO) with that of PPO with the Lagrangian multiplier method (PPO_Lag)[3] for (ExpCP) and that of WCSAC [27][4] for (CVaR-CP) which is a stricter problem than (ProbCP). We set the threshold $d_{th} = 15$ in (ExpCP), (CVaR-CP), and (QuantCP) and the target outage probability $\epsilon_0 = 0.1, 0.2$ in (CVaR-CP) and (QuantCP).

Fig. 5 shows the results of the considered algorithms on SimpleButtonEnv, DynamicEnv, and GremlinEnv. All experiments were done with 10 different random seeds, and the real line and the shaded area represent the average and average $\pm$ standard deviation, respectively. PPO with the Lagrangian multiplier method for (ExpCP) (green) keeps the average of the sum cost around the threshold $d_{th} = 15$ well (please see the graph in Appendix D.3), and its outage probability becomes around 0.35 as we can observe in Fig. 5d, 5e, and 5f. As aforementioned, the CVaR approach (WCSAC) should satisfy a sufficient condition for satisfying the outage probability constraint in (ProbCP). It is seen that WCSAC ($\epsilon_0 = 0.2$ (purple), $\epsilon_0 = 0.1$ (red)) achieves a lower or similar outage probability to the threshold $\epsilon_0$ in Fig. 5d, but the algorithm does not satisfy the outage probability constraint exactly in Fig. 5e and 5f. This means that the Gaussian distribution approximation of the distribution of $X^\pi(s)$ has limited capability to capture the decay rate of the tail probability. On the other hand, the proposed QCPO ($\epsilon_0 = 0.2$ (blue), $\epsilon_0 = 0.1$ (orange)) maintains the outage probability around the desired target outage probability very well, as shown in Fig. 5d, 5e, and 5f.

Now consider the average return of these algorithms. In constrained RL, in general, if an algorithm is allowed to have a higher sum of costs, then it has a higher return. Thus, as seen in Fig. 5d, 5e, and 5f, PPO_Lag induces the highest outage probability, so it has the highest average return, as shown in Fig. 5a, 5b, and 5c. The direct comparison between WCSAC and QCPO is less meaningful in

---

[3]We used the implementation code in `https://github.com/astooke/rlpyt/tree/master/rlpyt/projects/safe`, (MIT License)

[4]We used the github code that the authors of the paper uploaded: `https://github.com/AlgTUDelft/WCSAC`, (MIT License)

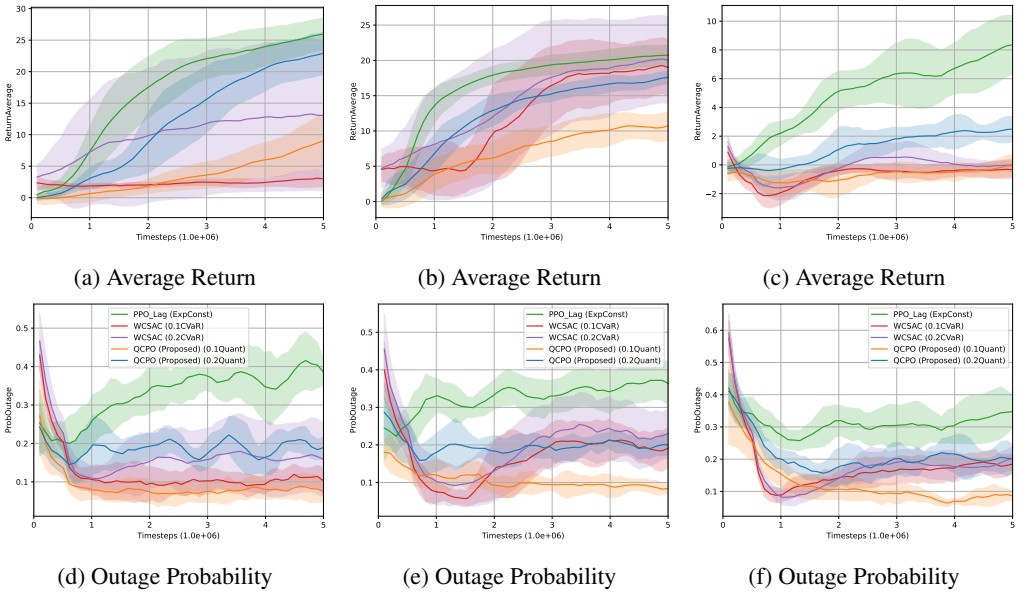

Figure 5: (left) SimpleButtonEnv, (middle) DynamicEnv, and (right) GremlinEnv: (upper row) average return and (lower row) outage probability of the most current 100 episodes.

DynamicEnv and GremlinEnv, since WCSAC does not satisfy the outage probability constraint, but it is fair in SimpleButtonEnv because both algorithms satisfy the outage probability constraint. As seen in Fig. 5a, QCPO achieves a higher average return than WCSAC for the same target probability constraint $\epsilon_0 = 0.1, 0.2$. This is because QCPO satisfies the target outage probability exactly, i.e., uses the given cost budget fully for a higher return. We provided more results in Appendix D.

## 6  Conclusion

We have proposed the framework of quantile-constrained RL to constrain the outage probability by adopting a constraint on the quantile, which is equivalent to the outage probability constraint. We have investigated issues in applying the policy gradient theorem to the Lagrangian of the quantile-constrained RL problem and have converted the quantile into an additive form of costs so that the application of the policy gradient theorem is feasible. Based on our derivation, we have constructed the QCPO algorithm, which uses distributional RL techniques to learn the $u$-quantile of the cumulative sum cost $X^\pi(s)$, and Weibull distribution to approximate the tail distribution of $X^\pi(s)$. We also proved the policy improvement condition for QCPO and showed that there exists an approximation loss due to our approximation of the quantile. Empirical results show that QCPO constrains the outage probability well as the desired target value. The meaning of such exact satisfaction of the outage probability is two-fold: First, the constraint on the outage probability is satisfied to control the probability of unsafe events, and second, the exact satisfaction of the cost constraint enables us to exploit the cost budget fully and obtain a higher return. Empirical results demonstrated the effectiveness of the proposed scheme.

## Acknowledgments and Disclosure of Funding

This work was supported by Institute of Information & Communications Technology Planning & Evaluation (IITP) grant funded by the Korea government (MSIT) (No.2022-0-00469, Development of Core Technologies for Task-oriented Reinforcement Learning for Commercialization of Autonomous Drones, 50%) and by Institute of Information & Communications Technology Planning & Evaluation (IITP) grant funded by the Korea government (MSIT) (No.2022-0-00124, Development of Artificial Intelligence Technology for Self-Improving Competency-Aware Learning Capabilities, 50%)

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
