# A  More Backgrounds

## A.1  Distributional RL

Distributional RL [2, 3, 8] is an area of RL that considers the distribution of the cumulative return $Z^\pi(s,a) = \sum_{t=0}^\infty \gamma^t r(S_t, A_t)$ for $S_0 = s$, $A_0 = a$, $S_{t+1} \sim M(\cdot|S_t, A_t)$, $A_{t+1} \sim \pi(\cdot|S_t)$, $t = 0, 1, \cdots$, instead of the expectation of the cumulative return $Q^\pi(s,a) = \mathbb{E}_\pi[Z^\pi(s,a)] = \mathbb{E}_\pi\left[\sum_{t=0}^\infty \gamma^t r(s_t, a_t)\right]$ to optimize a policy $\pi$. In distributional RL, the distribution of the cumulative return $Z^\pi(s,a)$ is computed by the distributional Bellman equation [3], defined as

$$Z^\pi(s,a) \overset{D}{=} r(s,a) + Z^\pi(S', A') \tag{24}$$

for $S' \sim M(\cdot|s,a)$, $A' \sim \pi(\cdot|S')$, where $\overset{D}{=}$ means that the random variable in the left-hand side (LHS) has the same distribution to that in the right-hand side (RHS). So, the following holds[15]:

$$F_{Z^\pi(s,a)}(z) = \mathbb{E}_{s' \sim M, a' \sim \pi}\left[F_{r(s,a)+\gamma Z^\pi(s',a')}(z)\right]$$

$$= \mathbb{E}_{s' \sim M, a' \sim \pi}\left[F_{Z^\pi(s',a')}\left(\frac{z - r(s,a)}{\gamma}\right)\right] \tag{25}$$

$$p_{Z^\pi(s,a)}(z) = \frac{1}{\gamma}\mathbb{E}_{s' \sim M, a' \sim \pi}\left[p_{Z^\pi(s',a')}\left(\frac{z - r(s,a)}{\gamma}\right)\right], \tag{26}$$

where $F_X(x)$ and $P_X(x)$ denote the cumulative distribution function (CDF) and PDF of a random variable $X$, respectively, and (26) is obtained by taking derivative of (25). To train the distribution of the cumulative return $Z^\pi(s,a)$, the $p$-Wasserstein distance $W_p(X, Y)$ is typically used, which can be written explicitly as

$$W_p(X, Y) = \left(\int_0^1 \left|F_X^{-1}(u) - F_Y^{-1}(u)\right|^p\right)^{1/p} \tag{27}$$

for $p < \infty$, where $F_X^{-1}(u) = \inf\{x \mid F_X(x) \geq u\} =: Q_X(u)$ is the quantile function (inverse CDF) of the random variable $X$. Dabney et al. [8, 9], Mavrin et al. [16], Kuznetsov et al. [13], Yang et al. [26] used quantile regression to learn the quantile of the cumulative return $Z^\pi(s,a)$. The quantile regression loss is given by $L_{quant,u}(q) = \mathbb{E}_X[l_{quant,u}(X - q)]$, where

$$l_{quant,u}(x) = \left(u - 1_{\{x<0\}}\right) \cdot x. \tag{28}$$

To smooth the gradient, they used the quantile Huber loss function $L_{Huber,u}(q) = \mathbb{E}_X[l_{Huber,u}(X - q)]$ for a given $\kappa > 0$, where

$$l_{Huber,u}(x) = \left|u - 1_{\{x<0\}}\right| \frac{L_\kappa(x)}{\kappa}, \tag{29}$$

$$L_\kappa(x) = \begin{cases} \frac{1}{2}x^2, & \text{if } |x| \leq \kappa \\ \kappa\left(|x| - \frac{1}{2}\kappa\right), & \text{otherwise.} \end{cases}$$

In this paper, we estimate the quantiles of the cumulative sum cost using the quantile loss, and use them to solve the constrained optimization problem (QuantCP).

$$\begin{aligned} \text{Maximize} \quad & \mathbb{E}_\pi\left[\sum_{t=0}^\infty \gamma^t r(s_t, a_t)\right] \\ \text{Subject to} \quad & q_{1-\epsilon_0}^\pi(s_0) \leq d_{th}, \end{aligned} \tag{QuantCP}$$

## A.2  Large Deviation Principle (LDP)

Large deviation principle (LDP) [11] is a technique for estimating the limiting behavior of a sequence of distributions. A simple example is the empirical mean $\bar{X}_n = \frac{1}{n}\sum_{k=1}^n X_k$ of i.i.d. random variables $X_i$. We say that a sequence $\{\bar{X}_n\}$ satisfies LDP if the sequence of its log probability distribution $\frac{1}{n}\log\Pr\left(\bar{X}_n \in \Gamma\right)$ satisfies the following condition $\frac{1}{n}\log\Pr\left(\bar{X}_n \in \Gamma\right) \overset{n\to\infty}{\longrightarrow} -\inf_{x\in\Gamma} I(x)$ for some function $I(x)$. The function $I(x)$ satisfying such limiting behavior is called the rate function of $\bar{X}_n$. The rate function $I(x)$ is also related to the cumulative distribution function $F_{\bar{X}_n}(x)$ since $1 - F_{\bar{X}_n}(x_0) = \Pr\left(\bar{X}_n \in [x_0, \infty)\right) \approx \exp\left(-n\inf_{x\in[x_0,\infty)} I(x)\right)$ for some $x_0 > \mathbb{E}[X]$ and sufficiently large $n$.

LDP can be applied to finite state Markov chains [11]. Let $Y_k \in \mathcal{Y} = \{y^1, \dots y^m\}$ be random variables that follows the Markov property: $\Pr(Y_1 = y_1, \dots, Y_n = y_n) = p_0(y_1) \prod_{i=1}^{n} M(y_{i+1}|y_i)$. Then, the sequence of empirical means $Z_n := \frac{1}{n} \sum_{k=0}^{n} X_k$, where $X_k = f(Y_k)$ for some function $f : \mathcal{Y} \to \mathbb{R}^d$, satisfies LDP and the rate function is given by $I(z) = \sup_{\lambda \in \mathbb{R}^d} \{\langle \lambda, z \rangle - \log \rho(\Pi_\lambda)\}$, where $\rho(\Pi)$ is the Perron-Frobenius eigenvalue of a given matrix $\Pi$, and $\Pi_\lambda$ is the matrix whose $(i,j)$-th element is $M(y^j|y^i) \exp \langle \lambda, f(y^j) \rangle$.

In this paper, we consider the tail probability of the distribution of the cumulative sum cost $X^\pi(s_0) = \sum_{t=0}^{\infty} \gamma^t c(s_t, a_t)$. Finding its analytic rate function is hard. Therefore, we instead approximate the rate function directly as $I_{X^\pi(s)}(x) \approx (x/\beta(s))^{\alpha(s)}$ with learnable parameters $\alpha(s)$ and $\beta(s)$, which results in a Weibull distribution: $1 - F_{X^\pi(s)}(x) = \exp\{-(x/\beta(s))^{\alpha(s)}\}$. We use this distribution to approximate the tail probability of $p_{X^\pi(s)}(x)$ of $X^\pi(s)$.

## A.3 The Considered Constrained Problems

In this subsection, we list the problems for constrained RL. The first constrained problem is a common problem used in many previous constrained RL papers.

$$\begin{aligned} \text{Maximize} \quad & V^\pi(s_0) := \mathbb{E}_\pi\left[\sum_{t=0}^{\infty} \gamma^t r(s_t, a_t)\right] \\ \text{subject to} \quad & C^\pi(s_0) := \mathbb{E}_\pi\left[\sum_{t=0}^{\infty} \gamma^t c(s_t, a_t)\right] \le d_{th}, \end{aligned} \quad \text{(ExpCP)}$$

In (ExpCP), the cost constraint is that the expectation of the sum of costs is less than or equal to a threshold parameter $d_{th}$. Note that the threshold $d_{th}$ is set on the average (i.e., expectation) of the cumulative sum cost to avoid undesired high-cost events in this formulation. However, solving the problem (ExpCP) may have undesirable outcomes for real environments that typically need constrained behavior on the event that the cost exceeds the threshold $d_{th}$.

There are two well-known techniques, called Value at Risk (VaR, or Quantile) and Conditional Value at Risk (CVaR), to manage undesirable events in the domain of finance[18]. In the context of RL, the definitions of the quantile and the CVaR for the distribution of the cumulative sum cost for a given $\pi$ are given by $q_u^\pi(s_0) := \inf\{x \mid \Pr(X^\pi(s_0) \le x) \ge u\}$ and $\text{CVaR}_u^\pi(s_0) := \mathbb{E}_\pi[X^\pi(s_0) \mid X^\pi(s_0) \ge q_u^\pi(s_0)]$, respectively. Note that the CVaR and the quantile are two different measures for undesirable events, and the choice between the two depends on what we desire. For example, an insurance company prefers the CVaR of undesirable events to determine an insurance premium. On the other hand, a company developing an autonomous driving car system needs the quantile of undesirable events to guarantee the accident probability for safety.

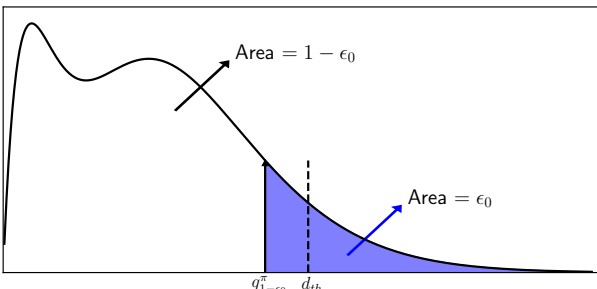

Figure 6: Equivalence between the outage probability constraint and the quantile constraint

The CVaR constrained problem to constrain undesirable events was previously used in RL [5, 27], and the problem is explicitly formulated as

$$\begin{aligned} \text{Maximize} \quad & \mathbb{E}_\pi\left[\sum_{t=0}^{\infty} \gamma^t r(s_t, a_t)\right] \\ \text{Subject to} \quad & \text{CVaR}_{1-\epsilon_0}^\pi(s_0) \le d_{th}, \end{aligned} \quad \text{(CVaR-CP)}$$

In this paper, we focus on constraining the probability of undesirable events that the cost exceeds the threshold $d_{th}$. Thus we can consider a constrained problem with a probabilistic constraint as follows:

$$\begin{aligned} \text{Maximize} \quad & V^\pi(s_0) = \mathbb{E}_\pi\left[\sum_{t=0}^{\infty} \gamma^t r(s_t, a_t)\right] \\ \text{Subject to} \quad & \Pr\left[\sum_{t=0}^{\infty} \gamma^t c(S_t, A_t) > d_{th}\right] \le \epsilon_0 \\ & \text{for } S_0 = s_0, A_t \sim \pi(\cdot|S_t), S_{t+1} \sim M(\cdot|S_t, A_t). \end{aligned} \quad \text{(ProbCP)}$$

Our approach to this problem is first to convert the outage probability constraint in (ProbCP) into a quantile constraint $q_{1-\epsilon_0}^{\pi}(s_0) \leq d_{th}$, which is equivalent to the original probabilistic constraint (See Fig. 6), and then to solve the equivalent optimization:

$$
\begin{aligned}
&\text{Maximize} &&\mathbb{E}_\pi\left[\sum_{t=0}^{\infty} \gamma^t r(s_t, a_t)\right] \\
&\text{Subject to} &&q_{1-\epsilon_0}^{\pi}(s_0) \leq d_{th},
\end{aligned}
\tag{QuantCP}
$$

Note that the $(1 - \epsilon_0)$-quantile denoted as $q_{1-\epsilon_0}^{\pi}(s)$ is always less or equal to than the $(1 - \epsilon_0)$-CVaR denoted as $\text{CVaR}_{1-\epsilon_0}^{\pi}(s)$ for all $s \in \mathcal{S}$ because of the definition of the CVaR. Therefore, satisfying the CVaR constraint is a sufficient condition for satisfying the probabilistic constraint, and hence this problem is a stricter problem than (ProbCP) or (QuantCP). Therefore, the algorithms proposed to solve (CVaR-CP) can be used for solving (ProbCP), and this should satisfy the probabilistic constraint in theory.

# B  Proofs

In the following proofs, we used text color so that readers can follow the proof easily.

## B.1  Proof of Theorem 1

**Assumption 1** (Boundness of quantile difference). *For a given policy $\pi$, the following two quantities are bounded*

$$|c(s,a) + \gamma q_u^\pi(s') - q_u^\pi(s)| \le \gamma R \tag{30}$$

$$\left| q_u^\pi(s) - F_{X^\pi(s)}^{-1}\left( F_{X^\pi(s')}\left( \frac{q_u^\pi(s) - c(s,a)}{\gamma} \right) \right) \right| \le R \tag{31}$$

*for all $(s, a, s') \in \mathcal{S} \times \mathcal{A} \times \mathcal{S}$ such that $\pi(a|s) \cdot M(s'|s,a) > 0$.*

Note that for finite MDPs, which are assumed for many RL proofs, this assumption definitely holds with a finite cost function.

**Assumption 2** (Smoothness of CDF of $X^\pi(s)$). *For each state $s$, the average slope of $F_{X^\pi(s)}(x)$ between $q_u^\pi(s)$ and $y \in [q_u^\pi(s) - R, q_u^\pi(s) + R]$ is bounded by*

$$\frac{1}{1+\epsilon} \cdot p_{X^\pi(s)}(q_u^\pi(s)) \le \frac{F_{X^\pi(s)}(q_u^\pi(s)) - F_{X^\pi(s)}(y)}{q_u^\pi(s) - y} \le \frac{1}{1-\epsilon} \cdot p_{X^\pi(s)}(q_u^\pi(s)) \tag{32}$$

*for small $0 < \epsilon < \frac{1}{2}$.*

This assumption holds when discrete masses are not present in the PDF and the CDF is continuous.

**Theorem 1.** *Under Assumptions 1 and 2, the $u$-quantile of the random variable $X^\pi(s_t)$ satisfies the following temporal-difference(TD) relation. For some constant $R$ and small $\epsilon > 0$,*

$$\left| \mathbb{E}_\pi \left[ \frac{p_{X^\pi(s_{t+1})}\left( \frac{q_u^\pi(s_t) - c(s_t, a_t)}{\gamma} \right)}{\gamma p_{X^\pi(s_t)}(q_u^\pi(s_t))} \{c(s_t, a_t) + \gamma q_u^\pi(s_{t+1}) - q_u^\pi(s_t)\} \right] \right| \le \frac{\epsilon}{1-\epsilon} R, \tag{33}$$

*Here, the expectation is for the action $a_t \sim \pi(\cdot|s_t)$ and the next state $s_{t+1} \sim M(\cdot|s_t, a_t)$. ($s_t$ is given.)*

*Proof.* Note that from (25),

$$F_{X^\pi(s)}(x) = \mathbb{E}_\pi \left[ F_{X^\pi(s')}\left( \frac{x - c(s,a)}{\gamma} \right) \right], \tag{34}$$

for all $x$. If $x = q_u^\pi(s)$, then this becomes

$$u = F_{X^\pi(s)}(q_u^\pi(s)) = \mathbb{E}_\pi \left[ F_{X^\pi(s')}\left( \frac{q_u^\pi(s) - c(s,a)}{\gamma} \right) \right]. \tag{35}$$

Using (35), we can obtain

$$\mathbb{E}_\pi \left[ \frac{p_{X^\pi(s_{t+1})}\left( \frac{q_u^\pi(s_t) - c(s_t, a_t)}{\gamma} \right)}{\gamma p_{X^\pi(s_t)}(q_u^\pi(s_t))} \{c(s_t, a_t) + \gamma q_u^\pi(s_{t+1}) - q_u^\pi(s_t)\} \right] \tag{36}$$

$$= \frac{\mathbb{E}_\pi \left[ p_{X^\pi(s_{t+1})}\left( \frac{q_u^\pi(s_t) - c(s_t, a_t)}{\gamma} \right) \{c(s_t, a_t) + \gamma q_u^\pi(s_{t+1}) - q_u^\pi(s_t)\} \right]}{\gamma p_{X^\pi(s_t)}(q_u^\pi(s_t))} \tag{37}$$

$$= \frac{1}{\gamma p_{X^\pi(s_t)}(q_u^\pi(s_t))} \times \mathbb{E}_\pi \left[ \gamma \underbrace{\left( F_{X^\pi(s_{t+1})}\left( \frac{q_u^\pi(s_t) - c(s_t, a_t)}{\gamma} \right) - u \right)}_{=0 \text{ by (35)}} \right.$$

$$\left. + p_{X^\pi(s_{t+1})}\left( \frac{q_u^\pi(s_t) - c(s_t, a_t)}{\gamma} \right) \{c(s_t, a_t) + \gamma q_u^\pi(s_{t+1}) - q_u^\pi(s_t)\} \right] \tag{38}$$

$$= \mathbb{E}_\pi \left[ \frac{\left( F_{X^\pi(s_{t+1})}\left( \frac{q_u^\pi(s_t)-c(s_t,a_t)}{\gamma} \right) - u \right) + p_{X^\pi(s_{t+1})}\left( \frac{q_u^\pi(s_t)-c(s_t,a_t)}{\gamma} \right) \left\{ q_u^\pi(s_{t+1}) - \frac{q_u^\pi(s_t)-c(s_t,a_t)}{\gamma} \right\}}{p_{X^\pi(s_t)}\left( q_u^\pi(s_t) \right)} \right] \tag{39}$$

$$= \mathbb{E}_\pi \left[ \frac{\left( u - F_{X^\pi(s_{t+1})}\left( \frac{q_u^\pi(s_t)-c(s_t,a_t)}{\gamma} \right) \right)}{p_{X^\pi(s_t)}\left( q_u^\pi(s_t) \right)} \right.$$
$$\left. \times \frac{\left( F_{X^\pi(s_{t+1})}\left( \frac{q_u^\pi(s_t)-c(s_t,a_t)}{\gamma} \right) - u \right) + p_{X^\pi(s_{t+1})}\left( \frac{q_u^\pi(s_t)-c(s_t,a_t)}{\gamma} \right) \left\{ q_u^\pi(s_{t+1}) - \frac{q_u^\pi(s_t)-c(s_t,a_t)}{\gamma} \right\}}{\left( u - F_{X^\pi(s_{t+1})}\left( \frac{q_u^\pi(s_t)-c(s_t,a_t)}{\gamma} \right) \right)} \right] \tag{40}$$

$$= \mathbb{E}_\pi \left[ \frac{\left( u - F_{X^\pi(s_{t+1})}\left( \frac{q_u^\pi(s_t)-c(s_t,a_t)}{\gamma} \right) \right)}{p_{X^\pi(s_t)}\left( q_u^\pi(s_t) \right)} \right.$$
$$\left. \times \left\{ \frac{p_{X^\pi(s_{t+1})}\left( \frac{q_u^\pi(s_t)-c(s_t,a_t)}{\gamma} \right) \left\{ q_u^\pi(s_{t+1}) - \frac{q_u^\pi(s_t)-c(s_t,a_t)}{\gamma} \right\}}{\left( u - F_{X^\pi(s_{t+1})}\left( \frac{q_u^\pi(s_t)-c(s_t,a_t)}{\gamma} \right) \right)} - 1 \right\} \right] \tag{41}$$

Then, by Cauchy-Schwarz inequality, we can obtain a bound such that

$$\mathbb{E}_\pi \left[ \frac{p_{X^\pi(s_{t+1})}\left( \frac{q_u^\pi(s_t)-c(s_t,a_t)}{\gamma} \right)}{\gamma p_{X^\pi(s_t)}\left( q_u^\pi(s_t) \right)} \left\{ c(s_t,a_t) + \gamma q_u^\pi(s_{t+1}) - q_u^\pi(s_t) \right\} \right]^2 \tag{42}$$

$$= \mathbb{E}_\pi \left[ \frac{\left( u - F_{X^\pi(s_{t+1})}\left( \frac{q_u^\pi(s_t)-c(s_t,a_t)}{\gamma} \right) \right)}{p_{X^\pi(s_t)}\left( q_u^\pi(s_t) \right)} \right.$$
$$\left. \times \left\{ \frac{p_{X^\pi(s_{t+1})}\left( \frac{q_u^\pi(s_t)-c(s_t,a_t)}{\gamma} \right) \left\{ q_u^\pi(s_{t+1}) - \frac{q_u^\pi(s_t)-c(s_t,a_t)}{\gamma} \right\}}{\left( u - F_{X^\pi(s_{t+1})}\left( \frac{q_u^\pi(s_t)-c(s_t,a_t)}{\gamma} \right) \right)} - 1 \right\} \right]^2 \tag{43}$$

$$\leq \underbrace{\mathbb{E}_\pi \left[ \left( \frac{u - F_{X^\pi(s_{t+1})}\left( \frac{q_u^\pi(s_t)-c(s_t,a_t)}{\gamma} \right)}{p_{X^\pi(s_t)}\left( q_u^\pi(s_t) \right)} \right)^2 \right]}_{(a)}$$

$$\times \underbrace{\mathbb{E}_\pi \left[ \left( 1 - \frac{p_{X^\pi(s_{t+1})}\left( \frac{q_u^\pi(s_t)-c(s_t,a_t)}{\gamma} \right) \cdot \left\{ q_u^\pi(s_{t+1}) - \frac{q_u^\pi(s_t)-c(s_t,a_t)}{\gamma} \right\}}{u - F_{X^\pi(s_{t+1})}\left( \frac{q_u^\pi(s_t)-c(s_t,a_t)}{\gamma} \right)} \right)^2 \right]}_{(b)} \tag{44}$$

Now we find upper bounds of (a) and (b).

- First, consider an upper bound of (a).

$$\frac{u - F_{X^\pi(s_{t+1})}\left( \frac{q_u^\pi(s_t)-c(s_t,a_t)}{\gamma} \right)}{p_{X^\pi(s_t)}\left( q_u^\pi(s_t) \right)} \tag{45}$$

$$= \frac{u - F_{X^\pi(s_t)}\left( F_{X^\pi(s_t)}^{-1}\left( F_{X^\pi(s_{t+1})}\left( \frac{q_u^\pi(s_t)-c(s_t,a_t)}{\gamma} \right) \right) \right)}{p_{X^\pi(s_t)}\left( q_u^\pi(s_t) \right)} \tag{46}$$

$$= \frac{u - F_{X^\pi(s_t)}\left( \bar{q}_u^\pi(s_t, a_t, s_{t+1}) \right)}{p_{X^\pi(s_t)}\left( q_u^\pi(s_t) \right)} \tag{47}$$

$$= \frac{F_{X^\pi(s_t)}\left(q_u^\pi(s_t)\right) - F_{X^\pi(s_t)}\left(\bar{q}_u^\pi(s_t, a_t, s_{t+1})\right)}{p_{X^\pi(s_t)}\left(q_u^\pi(s_t)\right)} \tag{48}$$

$$= \frac{\frac{F_{X^\pi(s_t)}(q_u^\pi(s_t)) - F_{X^\pi(s_t)}(\bar{q}_u^\pi(s_t, a_t, s_{t+1}))}{q_u^\pi(s_t) - \bar{q}_u^\pi(s_t, a_t, s_{t+1})}}{p_{X^\pi(s_t)}\left(q_u^\pi(s_t)\right)} \cdot \left\{q_u^\pi(s_t) - \bar{q}_u^\pi(s_t, a_t, s_{t+1})\right\} \tag{49}$$

where $\bar{q}_u^\pi(s_t, a_t, s_{t+1}) = F_{X^\pi(s_t)}^{-1}\left(F_{X^\pi(s_{t+1})}\left(\frac{q_u^\pi(s_t) - c(s_t, a_t)}{\gamma}\right)\right)$. Note that

$$\frac{F_{X^\pi(s_t)}\left(q_u^\pi(s_t)\right) - F_{X^\pi(s_t)}\left(\bar{q}_u^\pi(s_t, a_t, s_{t+1})\right)}{q_u^\pi(s_t) - \bar{q}_u^\pi(s_t, a_t, s_{t+1})}$$

is the average slope of $F_{X^\pi(s_t)}(x)$ between $q_u^\pi(s_t)$ and $\bar{q}_u^\pi(s_t, a_t, s_{t+1})$, and $p_{X^\pi(s_t)}\left(q_u^\pi(s_t)\right)$ is the slope of $F_{X^\pi(s_t)}(x)$ at $x = q_u^\pi(s_t)$. Therefore by Assumption 1 and 2, we can obtain an upper bound of (a) as follows:

$$\mathbb{E}_\pi\left[\left(\frac{u - F_{X^\pi(s_{t+1})}\left(\frac{q_u^\pi(s_t) - c(s_t, a_t)}{\gamma}\right)}{p_{X^\pi(s_t)}\left(q_u^\pi(s_t)\right)}\right)^2\right] \tag{50}$$

$$= \mathbb{E}_\pi\left[\left(\frac{\frac{F_{X^\pi(s_t)}(q_u^\pi(s_t)) - F_{X^\pi(s_t)}(\bar{q}_u^\pi(s_t, a_t, s_{t+1}))}{q_u^\pi(s_t) - \bar{q}_u^\pi(s_t, a_t, s_{t+1})}}{p_{X^\pi(s_t)}\left(q_u^\pi(s_t)\right)} \cdot \left\{q_u^\pi(s_t) - \bar{q}_u^\pi(s_t, a_t, s_{t+1})\right\}\right)^2\right] \tag{51}$$

$$= \mathbb{E}_\pi\Bigg[\Bigg(\frac{\frac{F_{X^\pi(s_t)}(q_u^\pi(s_t)) - F_{X^\pi(s_t)}(\bar{q}_u^\pi(s_t, a_t, s_{t+1}))}{q_u^\pi(s_t) - \bar{q}_u^\pi(s_t, a_t, s_{t+1})}}{p_{X^\pi(s_t)}\left(q_u^\pi(s_t)\right)}$$

$$\times \left\{q_u^\pi(s_t) - F_{X^\pi(s_t)}^{-1}\left(F_{X^\pi(s_{t+1})}\left(\frac{q_u^\pi(s_t) - c(s_t, a_t)}{\gamma}\right)\right)\right\}\Bigg)^2\Bigg] \tag{52}$$

$$\leq \frac{1}{(1-\epsilon)^2}R^2 \tag{53}$$

- Next, we consider an upper bound of (b). By Assumption 1 and 2,

$$\mathbb{E}_\pi\left[\left(1 - \frac{p_{X^\pi(s_{t+1})}\left(\frac{q_u^\pi(s_t) - c(s_t, a_t)}{\gamma}\right) \cdot \left\{q_u^\pi(s_{t+1}) - \frac{q_u^\pi(s_t) - c(s_t, a_t)}{\gamma}\right\}}{u - F_{X^\pi(s_{t+1})}\left(\frac{q_u^\pi(s_t) - c(s_t, a_t)}{\gamma}\right)}\right)^2\right] \tag{54}$$

$$= \mathbb{E}_\pi\left[\left(1 - \frac{p_{X^\pi(s_{t+1})}\left(\frac{q_u^\pi(s_t) - c(s_t, a_t)}{\gamma}\right)}{\frac{u - F_{X^\pi(s_{t+1})}\left(\frac{q_u^\pi(s_t) - c(s_t, a_t)}{\gamma}\right)}{\left\{q_u^\pi(s_{t+1}) - \frac{q_u^\pi(s_t) - c(s_t, a_t)}{\gamma}\right\}}}\right)^2\right] \tag{55}$$

$$\leq \epsilon^2 \tag{56}$$

Therefore by combining two upper bounds, we can conclude the theorem.

$$\left|\mathbb{E}_\pi\left[\frac{p_{X^\pi(s_{t+1})}\left(\frac{q_u^\pi(s_t) - c(s_t, a_t)}{\gamma}\right)}{\gamma p_{X^\pi(s_t)}\left(q_u^\pi(s_t)\right)}\left\{c(s_t, a_t) + \gamma q_u^\pi(s_{t+1}) - q_u^\pi(s_t)\right\}\right]\right| \tag{57}$$

$$\leq \left(\underbrace{\mathbb{E}_\pi\left[\left(\frac{u - F_{X^\pi(s_{t+1})}\left(\frac{q_u^\pi(s_t) - c(s_t, a_t)}{\gamma}\right)}{p_{X^\pi(s_t)}\left(q_u^\pi(s_t)\right)}\right)^2\right]}_{(a)}\right)^{\frac{1}{2}} \tag{58}$$

$$\times \left( \underbrace{\mathbb{E}_\pi \left[ \left( 1 - \frac{p_{X^\pi(s_{t+1})}\left(\frac{q_u^\pi(s_t)-c(s_t,a_t)}{\gamma}\right) \cdot \left\{ q_u^\pi(s_{t+1}) - \frac{q_u^\pi(s_t)-c(s_t,a_t)}{\gamma} \right\}}{u - F_{X^\pi(s_{t+1})}\left(\frac{q_u^\pi(s_t)-c(s_t,a_t)}{\gamma}\right)} \right)^2 \right]}_{(b)} \right)^{\frac{1}{2}} \tag{59}$$

$$\leq \frac{\epsilon}{1-\epsilon} R \tag{60}$$

$\square$

**Corollary 1.** *Under Assumptions 1 and 2, the $u$-quantile $q_u^\pi(s_t)$ of the random variable $X^\pi(s_t)$ is bounded as*

$$\left| q_u^\pi(s_t) - \mathbb{E}_\pi \left[ \mu_u^\pi(s_t, a_t, s_{t+1}) \left\{ c(s_t, a_t) + \gamma q_u^\pi(s_{t+1}) \right\} \right] \right| \leq \frac{\epsilon}{1-\epsilon} R. \tag{61}$$

*where*

$$\mu_u^\pi(s_t, a_t, s_{t+1}) := \frac{p_{X^\pi(s_{t+1})}\left(\frac{q_u^\pi(s_t)-c(s_t,a_t)}{\gamma}\right)}{\gamma p_{X^\pi(s_t)}(q_u^\pi(s_t))} \tag{62}$$

*Proof.* Note that the term $q_u^\pi(s_t)$ can go outside the expectation in (33) since the expectation is over $(a_t, s_{t+1})$. From eq. (26) in Appendix A.1, the expectation of the numerator of $\mu_u^\pi(s_t, a_t, s_{t+1})$ is the same as the the denominator of the weight, i.e., $\mathbb{E}_\pi\left[ p_{X^\pi(s_{t+1})}\left(\frac{q_u^\pi(s_t)-c(s_t,a_t)}{\gamma}\right) \right] = \gamma p_{X^\pi(s_t)}(q_u^\pi(s_t))$ and this leads to $\mathbb{E}_\pi[\mu_u^\pi(s_t, a_t, s_{t+1})] = 1$. So, we have the claim. $\square$

### B.2 Proof of Lemma 1

**Lemma 1.** *Suppose that the state transition dynamics are deterministic, i.e., $s_{t+1} = h(s_t, a_t)$. Then, under Assumptions 1 and 2, the $u$-quantile $q_u^\pi(s_0)$ of the random variable $X^\pi(s_0)$ is expressed as*

$$\left| q_u^\pi(s_0) - \mathbb{E}_{\tilde{\pi}_u}\left[ \sum_{t=0}^\infty \gamma^t c(s_t, a_t) \right] \right| \leq \frac{\epsilon R}{(1-\epsilon)(1-\gamma)}, \tag{63}$$

*where*

$$\tilde{\pi}_u(a|s) = \pi(a|s) \cdot \frac{p_{X^\pi(h(s,a))}\left(\frac{q_u^\pi(s)-c(s,a)}{\gamma}\right)}{\gamma p_{X^\pi(s)}(q_u^\pi(s))} \propto \pi(a|s) \cdot p_{X^\pi(h(s,a))}\left(\frac{q_u^\pi(s)-c(s,a)}{\gamma}\right). \tag{64}$$

*Proof.* Remind that $\mu_u^\pi(s, a, s')$ is defined in (62) as

$$\mu_u^\pi(s, a, s') := \frac{p_{X^\pi(s')}\left(\frac{q_u^\pi(s)-c(s,a)}{\gamma}\right)}{\gamma p_{X^\pi(s)}(q_u^\pi(s))}$$

Consider $\mathbb{E}_\pi\left[ \mu_u^\pi(s, a, s') \left\{ c(s, a) + \gamma q_u^\pi(s') \right\} \right]$.

$$\mathbb{E}_\pi\left[ \mu_u^\pi(s, a, s') \left\{ c(s, a) + \gamma q_u^\pi(s') \right\} \right] \tag{65}$$

$$= \sum_a \sum_{s'} \pi(a|s) \cdot M(s'|s, a) \cdot \mu_u^\pi(s, a, s') \left\{ c(s, a) + \gamma q_u^\pi(s') \right\} \tag{66}$$

$$= \sum_a \sum_{s'} \tilde{\pi}_u(a|s) \cdot \tilde{M}_u(s'|s, a) \left\{ c(s, a) + \gamma q_u^\pi(s') \right\} \tag{67}$$

for some distorted policy $\tilde{\pi}_u$ and some distorted state transition dynamics $\tilde{M}_u$. This is because

$$\mathbb{E}_\pi\left[ \mu_u^\pi(s, a, s') \right] = \sum_a \sum_{s'} \pi(a|s) \cdot M(s'|s, a) \cdot \frac{p_{X^\pi(s')}\left(\frac{q_u^\pi(s)-c(s,a)}{\gamma}\right)}{\gamma p_{X^\pi(s)}(q_u^\pi(s))} = 1. \tag{68}$$

where the last equation holds from (26). Now, under the assumption that the state transition dynamics is deterministic $s' = h(s,a)$, i.e., $p(s'|s,a) = \delta_{h(s,a)}(s')$, the distorted transition dynamics are the same as the original transition dynamics and the only difference is the distorted policy:

$$\tilde{M}_u(s'|s,a) = \frac{\pi(a|s)M(s'|a,s)\mu_u^\pi(s,a,s')}{\sum_{\tilde{s}} \pi(a|s)M(\tilde{s}|a,s)\mu_u^\pi(s,a,\tilde{s})} \tag{69}$$

$$= \frac{\delta_{h(s,a)}(s')\pi(a|s)\mu_u^\pi(s,a,s')}{\sum_{\tilde{s}} \delta_{h(s,a)}(\tilde{s})\pi(a|s)\mu_u^\pi(s,a,\tilde{s})} \tag{70}$$

$$= \frac{\delta_{h(s,a)}(s')\mu_u^\pi(s,a,h(s,a))}{\mu_u^\pi(s,a,h(s,a))} \tag{71}$$

$$= \delta_{h(s,a)}(s') \tag{72}$$

$$= M(s'|s,a) \tag{73}$$

$$\tilde{\pi}_u(a|s) = \pi(a|s)\frac{\sum_{s'} M(s'|a,s)\mu_u^\pi(s,a,s')}{\sum_{\tilde{a},s'} \pi(\tilde{a}|s)M(s'|\tilde{a},s)\mu_u^\pi(s,\tilde{a},s')} \tag{74}$$

$$= \pi(a|s)\frac{\mu_u^\pi(s,a,h(s,a))}{\sum_{\tilde{a}} \pi(\tilde{a}|s)\mu_u^\pi(s,\tilde{a},h(s,\tilde{a}))} \tag{75}$$

$$\overset{(a)}{=} \pi(a|s)\mu_u^\pi(s,a,h(s,a)) \tag{76}$$

$$= \pi(a|s)\frac{p_{X^\pi(h(s,a))}\left(\frac{q_u^\pi(s)-c(s,a)}{\gamma}\right)}{\gamma p_{X^\pi(s)}(q_u^\pi(s))} \tag{77}$$

$$\propto \pi(a|s) \cdot p_{X^\pi(h(s,a))}\left(\frac{q_u^\pi(s)-c(s,a)}{\gamma}\right). \tag{78}$$

Here the equality (a) holds from (68). Thus, from Corollary 1, we can obtain the following approximation:

$$\mathbb{E}_{a_t \sim \tilde{\pi}_u}[c(s_t,a_t) + \gamma q_u^\pi(s_{t+1})] - \frac{\epsilon R}{1-\epsilon} \le q_u^\pi(s_t) \le \mathbb{E}_{a_t \sim \tilde{\pi}_u}[c(s_t,a_t) + \gamma q_u^\pi(s_{t+1})] + \frac{\epsilon R}{1-\epsilon}. \tag{79}$$

Therefore, we obtain

$$q_u^\pi(s_0) \le \mathbb{E}_{a_0 \sim \tilde{\pi}_u}[c(s_0,a_0) + \gamma q_u^\pi(s_1)] + \frac{\epsilon R}{1-\epsilon} \tag{80}$$

$$\le \mathbb{E}_{a_0,a_1 \sim \tilde{\pi}_u}[c(s_0,a_0) + \gamma c(s_1,a_1) + \gamma^2 q_u^\pi(s_2)] + \frac{\epsilon R}{1-\epsilon}(1+\gamma) \tag{81}$$

$$\le \cdots \tag{82}$$

$$\le \mathbb{E}_{\tilde{\pi}_u}\left[\sum_{t=0}^\infty \gamma^t c(s_t,a_t)\right] + \frac{\epsilon R}{(1-\epsilon)(1-\gamma)}, \tag{83}$$

$$q_u^\pi(s_0) \ge \mathbb{E}_{a_0 \sim \tilde{\pi}_u}[c(s_0,a_0) + \gamma q_u^\pi(s_1)] - \frac{\epsilon R}{1-\epsilon} \tag{84}$$

$$\ge \mathbb{E}_{a_0,a_1 \sim \tilde{\pi}_u}[c(s_0,a_0) + \gamma c(s_1,a_1) + \gamma^2 q_u^\pi(s_2)] - \frac{\epsilon R}{1-\epsilon}(1+\gamma) \tag{85}$$

$$\ge \cdots \tag{86}$$

$$\ge \mathbb{E}_{\tilde{\pi}_u}\left[\sum_{t=0}^\infty \gamma^t c(s_t,a_t)\right] - \frac{\epsilon R}{(1-\epsilon)(1-\gamma)}. \tag{87}$$

$\square$

## B.3 Proof of Theorem 2

**Theorem 2.** *Under deterministic dynamics $s_{t+1} = h(s_t,a_t)$ and Assumptions 1 and 2, $q_u^\pi(s)$ can be expressed as*

$$\left| q_u^\pi(s_0) - \mathbb{E}_\pi\left[\sum_{t=0}^\infty \gamma^t \{c(s_t,a_t) + \tilde{c}_u^\pi(s_t,a_t)\}\right] \right| \le \frac{\epsilon R}{(1-\epsilon)(1-\gamma)}, \tag{88}$$

*where*

$$\tilde{c}_u^\pi(s, a) = \left( \frac{p_{X^\pi(s')} \left( \frac{q_u^\pi(s) - c(s,a)}{\gamma} \right)}{\gamma p_{X^\pi(s)} \left( q_u^\pi(s) \right)} - 1 \right) \cdot [c(s, a) + \gamma q_u^\pi(h(s, a))].$$

*Proof.* From (79), we have

$$q_u^\pi(s_t) \leq \mathbb{E}_{a_t \sim \tilde{\pi}_u} \left[ c(s_t, a_t) + \gamma q_u^\pi(s_{t+1}) \right] + \frac{\epsilon R}{1 - \epsilon} \tag{89}$$

$$q_u^\pi(s_t) \geq \mathbb{E}_{a_t \sim \tilde{\pi}_u} \left[ c(s_t, a_t) + \gamma q_u^\pi(s_{t+1}) \right] - \frac{\epsilon R}{1 - \epsilon} \tag{90}$$

The expectation $\mathbb{E}_{a_t \sim \tilde{\pi}_u} \left[ c(s_t, a_t) + \gamma q_u^\pi(s_{t+1}) \right]$ can be rewritten as

$$\mathbb{E}_{a_t \sim \tilde{\pi}_u} \left[ c(s_t, a_t) + \gamma q_u^\pi(s_{t+1}) \right] = \mathbb{E}_{a_t \sim \pi} \left[ \frac{\tilde{\pi}_u(a_t|s_t)}{\pi(a_t|s_t)} \cdot \{ c(s_t, a_t) + \gamma q_u^\pi(h(s_t, a_t)) \} \right] \tag{91}$$

$$= \mathbb{E}_{a_t \sim \pi} \left[ c(s_t, a_t) + \tilde{c}_u^\pi(s_t, a_t) + \gamma q_u^\pi(h(s_t, a_t)) \right] \tag{92}$$

$$= \mathbb{E}_{a_t \sim \pi} \left[ c(s_t, a_t) + \tilde{c}_u^\pi(s_t, a_t) + \gamma q_u^\pi(s_{t+1}) \right] \tag{93}$$

where

$$\tilde{c}_u^\pi(s, a) := \left( \frac{\tilde{\pi}_u(a|s)}{\pi(a|s)} - 1 \right) \cdot \{ c(s, a) + \gamma q_u^\pi(h(s, a)) \} \tag{94}$$

$$= \left( \frac{p_{X^\pi(h(s,a))} \left( \frac{q_u^\pi(s) - c(s,a)}{\gamma} \right)}{\gamma p_{X^\pi(s)} \left( q_u^\pi(s) \right)} - 1 \right) \cdot \{ c(s, a) + \gamma q_u^\pi(h(s, a)) \}. \tag{95}$$

Then, using (93), we obtain

$$q_u^\pi(s_0) \leq \mathbb{E}_{a_0 \sim \tilde{\pi}_u} \left[ c(s_0, a_0) + \gamma q_u^\pi(s_1) \right] + \frac{\epsilon R}{1 - \epsilon} \tag{96}$$

$$= \mathbb{E}_{a_0 \sim \pi} \left[ c(s_0, a_0) + \tilde{c}_u^\pi(s_0, a_0) + \gamma q_u^\pi(s_1) \right] + \frac{\epsilon R}{1 - \epsilon} \tag{97}$$

$$\leq \mathbb{E}_{a_0, a_1 \sim \pi} \left[ \{ c(s_0, a_0) + \tilde{c}_u^\pi(s_0, a_0) \} + \gamma \{ c(s_1, a_1) + \tilde{c}_u^\pi(s_1, a_1) \} + \gamma^2 q_u^\pi(s_2) \right] \tag{98}$$

$$+ \frac{\epsilon R}{1 - \epsilon} (1 + \gamma) \tag{99}$$

$$\leq \cdots \tag{100}$$

$$\leq \mathbb{E}_\pi \left[ \sum_{t=0}^\infty \gamma^t \{ c(s_t, a_t) + \tilde{c}_u^\pi(s_t, a_t) \} \right] + \frac{\epsilon R}{(1 - \epsilon)(1 - \gamma)} \tag{101}$$

$$q_u^\pi(s_0) \geq \mathbb{E}_{a_0 \sim \tilde{\pi}_u} \left[ c(s_0, a_0) + \gamma q_u^\pi(s_1) \right] - \frac{\epsilon R}{1 - \epsilon} \tag{102}$$

$$= \mathbb{E}_{a_0 \sim \pi} \left[ c(s_0, a_0) + \tilde{c}_u^\pi(s_0, a_0) + \gamma q_u^\pi(s_1) \right] - \frac{\epsilon R}{1 - \epsilon} \tag{103}$$

$$\geq \mathbb{E}_{a_0, a_1 \sim \pi} \left[ \{ c(s_0, a_0) + \tilde{c}_u^\pi(s_0, a_0) \} + \gamma \{ c(s_1, a_1) + \tilde{c}_u^\pi(s_1, a_1) \} + \gamma^2 q_u^\pi(s_2) \right] \tag{104}$$

$$- \frac{\epsilon R}{1 - \epsilon} (1 + \gamma) \tag{105}$$

$$\geq \cdots \tag{106}$$

$$\geq \mathbb{E}_\pi \left[ \sum_{t=0}^\infty \gamma^t \{ c(s_t, a_t) + \tilde{c}_u^\pi(s_t, a_t) \} \right] - \frac{\epsilon R}{(1 - \epsilon)(1 - \gamma)} \tag{107}$$

$$\tag{108}$$

$$\square$$

## B.4 Proof of Theorem 3

**Assumption 3** (Lipschitz continuity of $\tilde{c}_u^\pi(s,a)$ over $\pi$). *For any given fixed $u \in (0,1)$ and any policies $\pi$ and $\pi'$, there exists a coefficient $C_u$ such that*

$$\left| \tilde{c}_u^{\pi'}(s,a) - \tilde{c}_u^\pi(s,a) \right| \leq C_u \cdot \max_{s'} KL\left( \pi'(\cdot|s') \,\|\, \pi(\cdot|s') \right) \tag{109}$$

*for all $s \in \mathcal{S}$, $a \in \mathcal{A}$.*

Basically, Assumption 3 is that the function $\tilde{c}_u^\pi$ as a function of $\pi$ is continuous, which is expected to be satisfied if there is no abrupt change in the associated distributions.

**Theorem 3.** *Under deterministic dynamics $s_{t+1} = h(s_t, a_t)$ and Assumptions 1, 2, and 3, the $u$-quantile $q_u^\pi(s_0)$ is expressed as the expectation of the sum of actual cost and a $\pi$-independent additional cost $\tilde{c}_u^{\pi'}(s,a)$ for $\pi'$ satisfying $\max_s KL(\pi'(\cdot|s) \,\|\, \pi(\cdot|s)) \leq \delta$:*

$$\left| q_u^\pi(s_0) - \mathbb{E}_\pi \left[ \sum_{t=0}^\infty \gamma^t \left\{ c(s_t, a_t) + \tilde{c}_u^{\pi'}(s_t, a_t) \right\} \right] \right| \leq \frac{\epsilon R}{(1-\epsilon)(1-\gamma)} + \frac{C_u}{1-\gamma} \delta. \tag{110}$$

*Proof.* From Assumption 3, the additional cost $\tilde{c}_u^\pi(s,a)$ is bounded as follows

$$\tilde{c}_u^\pi(s,a) \leq \tilde{c}_u^{\pi'}(s,a) + C_u \cdot \max_s KL(\pi'(\cdot|s) \,\|\, \pi(\cdot|s)) \tag{111}$$

$$\leq \tilde{c}_u^{\pi'}(s,a) + C_u \cdot \delta \tag{112}$$

$$\tilde{c}_u^\pi(s,a) \geq \tilde{c}_u^{\pi'}(s,a) - C_u \cdot \max_s KL(\pi'(\cdot|s) \,\|\, \pi(\cdot|s)) \tag{113}$$

$$\geq \tilde{c}_u^{\pi'}(s,a) - C_u \cdot \delta \tag{114}$$

for $\pi'$ satisfying $\max_s KL(\pi'(\cdot|s) \,\|\, \pi(\cdot|s)) \leq \delta$. Thus, from (88), we can obtain the following bounds

$$q_u^\pi(s_0) \leq \mathbb{E}_\pi \left[ \sum_{t=0}^\infty \gamma^t \left\{ c(s_t, a_t) + \tilde{c}_u^\pi(s_t, a_t) \right\} \right] + \frac{\epsilon R}{(1-\epsilon)(1-\gamma)} \tag{115}$$

$$\leq \mathbb{E}_\pi \left[ \sum_{t=0}^\infty \gamma^t \left\{ c(s_t, a_t) + \tilde{c}_u^{\pi'}(s_t, a_t) + C_u \cdot \delta \right\} \right] + \frac{\epsilon R}{(1-\epsilon)(1-\gamma)} \tag{116}$$

$$= \mathbb{E}_\pi \left[ \sum_{t=0}^\infty \gamma^t \left\{ c(s_t, a_t) + \tilde{c}_u^{\pi'}(s_t, a_t) \right\} \right] + \frac{\epsilon R}{(1-\epsilon)(1-\gamma)} + \frac{C_u}{1-\gamma} \delta \tag{117}$$

$$q_u^\pi(s_0) \geq \mathbb{E}_\pi \left[ \sum_{t=0}^\infty \gamma^t \left\{ c(s_t, a_t) + \tilde{c}_u^\pi(s_t, a_t) \right\} \right] - \frac{\epsilon R}{(1-\epsilon)(1-\gamma)} \tag{118}$$

$$\geq \mathbb{E}_\pi \left[ \sum_{t=0}^\infty \gamma^t \left\{ c(s_t, a_t) + \tilde{c}_u^{\pi'}(s_t, a_t) - C_u \cdot \delta \right\} \right] - \frac{\epsilon R}{(1-\epsilon)(1-\gamma)} \tag{119}$$

$$= \mathbb{E}_\pi \left[ \sum_{t=0}^\infty \gamma^t \left\{ c(s_t, a_t) + \tilde{c}_u^{\pi'}(s_t, a_t) \right\} \right] - \frac{\epsilon R}{(1-\epsilon)(1-\gamma)} - \frac{C_u}{1-\gamma} \delta \tag{120}$$

$\square$

## B.5 Proof of Policy Improvement Condition

**Lemma 2** (Telescoping Lemma for $u$-quantile). *Under deterministic dynamics $s_{t+1} = h(s_t, a_t)$ and Assumption 1 and 2, the following holds for any two policies $\pi$ and $\pi'$:*

$$\left| q_u^\pi(s_0) - \left\{ q_u^{\pi'}(s_0) + \mathbb{E}_\pi \left[ \sum_{t=0}^\infty \gamma^t \left( c(s_t, a_t) + \tilde{c}_u^\pi(s_t, a_t) + \gamma q_u^{\pi'}(s_{t+1}) - q_u^{\pi'}(s_t) \right) \right] \right\} \right| \tag{121}$$

$$\leq \frac{\epsilon R}{(1-\epsilon)(1-\gamma)} \tag{122}$$

*Proof.* With Assumption 1 and 2, we have the following inequality by Theorem 2:

$$q_u^\pi(s_0) \le \mathbb{E}_\pi \left[ \sum_{t=0}^\infty \gamma^t \{c(s_t, a_t) + \tilde{c}_u^\pi(s_t, a_t)\} \right] + \frac{\epsilon R}{(1-\epsilon)(1-\gamma)} \tag{123}$$

$$q_u^\pi(s_0) \ge \mathbb{E}_\pi \left[ \sum_{t=0}^\infty \gamma^t \{c(s_t, a_t) + \tilde{c}_u^\pi(s_t, a_t)\} \right] - \frac{\epsilon R}{(1-\epsilon)(1-\gamma)} \tag{124}$$

Then note that

$$q_u^{\pi'}(s_0) = -\mathbb{E}_\pi \left[ \sum_{t=0}^\infty \gamma^t \left\{ \gamma q_u^{\pi'}(s_{t+1}) - q_u^{\pi'}(s_t) \right\} \right]. \tag{125}$$

Therefore,

$$q_u^\pi(s_0) - q_u^{\pi'}(s_0) \tag{126}$$

$$\le \mathbb{E}_\pi \left[ \sum_{t=0}^\infty \gamma^t \{c(s_t, a_t) + \tilde{c}_u^\pi(s_t, a_t)\} \right] + \mathbb{E}_\pi \left[ \sum_{t=0}^\infty \gamma^t \left\{ \gamma q_u^{\pi'}(s_{t+1}) - q_u^{\pi'}(s_t) \right\} \right] \tag{127}$$

$$+ \frac{\epsilon R}{(1-\epsilon)(1-\gamma)} \tag{128}$$

$$= \mathbb{E}_\pi \left[ \sum_{t=0}^\infty \gamma^t \left\{ c(s_t, a_t) + \tilde{c}_u^\pi(s_t, a_t) + \gamma q_u^{\pi'}(s_{t+1}) - q_u^{\pi'}(s_t) \right\} \right] + \frac{\epsilon R}{(1-\epsilon)(1-\gamma)} \tag{129}$$

$$q_u^\pi(s_0) - q_u^{\pi'}(s_0) \tag{130}$$

$$\ge \mathbb{E}_\pi \left[ \sum_{t=0}^\infty \gamma^t \{c(s_t, a_t) + \tilde{c}_u^\pi(s_t, a_t)\} \right] + \mathbb{E}_\pi \left[ \sum_{t=0}^\infty \gamma^t \left\{ \gamma q_u^{\pi'}(s_{t+1}) - q_u^{\pi'}(s_t) \right\} \right] \tag{131}$$

$$- \frac{\epsilon R}{(1-\epsilon)(1-\gamma)} \tag{132}$$

$$= \mathbb{E}_\pi \left[ \sum_{t=0}^\infty \gamma^t \left\{ c(s_t, a_t) + \tilde{c}_u^\pi(s_t, a_t) + \gamma q_u^{\pi'}(s_{t+1}) - q_u^{\pi'}(s_t) \right\} \right] - \frac{\epsilon R}{(1-\epsilon)(1-\gamma)} \tag{133}$$

$$\square$$

Next, we can obtain the following corollary.

**Corollary 2.** *Under deterministic dynamics $s_{t+1} = h(s_t, a_t)$ and Assumptions 1, 2, and 3, the following holds for any two policies $\pi$ and $\pi'$ :*

$$\left| q_u^\pi(s_0) - \left\{ q_u^{\pi'}(s_0) + \mathbb{E}_\pi \left[ \sum_{t=0}^\infty \gamma^t \left( c(s_t, a_t) + \tilde{c}_u^{\pi'}(s_t, a_t) + \gamma q_u^{\pi'}(s_{t+1}) - q_u^{\pi'}(s_t) \right) \right] \right\} \right| \tag{134}$$

$$\le \frac{\epsilon R}{(1-\epsilon)(1-\gamma)} + \frac{C_u}{1-\gamma} \max_s KL(\pi'(\cdot|s) \parallel \pi(\cdot|s)) \tag{135}$$

*Proof.* Let denote $\delta = \max_s \mathrm{KL}(\pi'(\cdot|s) \parallel \pi(\cdot|s))$ for simplicity in this proof. If Assumption 3 holds, this can be rewritten as

$$q_u^\pi(s_0) - q_u^{\pi'}(s_0) \tag{136}$$

$$\le \mathbb{E}_\pi \left[ \sum_{t=0}^\infty \gamma^t \left\{ c(s_t, a_t) + \tilde{c}_u^\pi(s_t, a_t) + \gamma q_u^{\pi'}(s_{t+1}) - q_u^{\pi'}(s_t) \right\} \right] + \frac{\epsilon R}{(1-\epsilon)(1-\gamma)} \tag{137}$$

$$\le \mathbb{E}_\pi \left[ \sum_{t=0}^\infty \gamma^t \left\{ c(s_t, a_t) + \tilde{c}_u^{\pi'}(s_t, a_t) + C_u \cdot \delta + \gamma q_u^{\pi'}(s_{t+1}) - q_u^{\pi'}(s_t) \right\} \right] + \frac{\epsilon R}{(1-\epsilon)(1-\gamma)} \tag{138}$$

$$= \mathbb{E}_\pi \left[ \sum_{t=0}^\infty \gamma^t \left\{ c(s_t, a_t) + \tilde{c}_u^{\pi'}(s_t, a_t) + \gamma q_u^{\pi'}(s_{t+1}) - q_u^{\pi'}(s_t) \right\} \right] + \frac{\epsilon R}{(1-\epsilon)(1-\gamma)} + \frac{C_u}{1-\gamma} \delta \tag{139}$$

$$q_u^\pi(s_0) - q_u^{\pi'}(s_0) \tag{140}$$

$$\geq \mathbb{E}_\pi \left[ \sum_{t=0}^\infty \gamma^t \left\{ c(s_t, a_t) + \tilde{c}_u^\pi(s_t, a_t) + \gamma q_u^{\pi'}(s_{t+1}) - q_u^{\pi'}(s_t) \right\} \right] - \frac{\epsilon R}{(1-\epsilon)(1-\gamma)} \tag{141}$$

$$\geq \mathbb{E}_\pi \left[ \sum_{t=0}^\infty \gamma^t \left\{ c(s_t, a_t) + \tilde{c}_u^{\pi'}(s_t, a_t) - C_u \cdot \delta + \gamma q_u^{\pi'}(s_{t+1}) - q_u^{\pi'}(s_t) \right\} \right] - \frac{\epsilon R}{(1-\epsilon)(1-\gamma)} \tag{142}$$

$$= \mathbb{E}_\pi \left[ \sum_{t=0}^\infty \gamma^t \left\{ c(s_t, a_t) + \tilde{c}_u^{\pi'}(s_t, a_t) + \gamma q_u^{\pi'}(s_{t+1}) - q_u^{\pi'}(s_t) \right\} \right] - \frac{\epsilon R}{(1-\epsilon)(1-\gamma)} - \frac{C_u}{1-\gamma} \delta \tag{143}$$

$\square$

To prove improvement theorem, we need a definition of $\alpha$-coupled policy and several lemmas similar to [19].

**Definition 1** (From [19]). *The two policies $\pi$ and $\pi'$ are $\alpha$-coupled if $Pr\,(a \neq a') \leq \alpha$, $(a, a') \sim (\pi(a|s), \pi'(a'|s))$ for all $s$.*

For the $u$-quantile, we define an advantage function $A_u^\pi(s, a)$ using the additional cost function $\tilde{c}_u^\pi(s, a)$ as

$$A_u^\pi(s, a) := c(s, a) + \tilde{c}_u^\pi(s, a) + \gamma \mathbb{E}_{s'} \left[ q_u^\pi(s') \right] - q_u^\pi(s) \tag{144}$$

**Lemma 3** (Similar to Lemma 2 in [19]). *Under deterministic dynamics $s_{t+1} = h(s_t, a_t)$ and Assumptions 1 and 2, $\alpha$-coupled policies $\pi$ and $\pi'$ satisfy the following inequality*

$$\left| \mathbb{E}_\pi \left[ A_u^{\pi'}(s, a) \right] \right| \leq 2\alpha \max_{s,a} \left| A_u^{\pi'}(s, a) \right| + \frac{\epsilon R}{(1-\epsilon)} \tag{145}$$

*for all $s$.*

*Proof.* (Similar to the proof of Lemma 2 in [19]) First we note that the following holds by (93) and Theorem 1:

$$\left| \mathbb{E}_{a \sim \pi'} \left[ A_u^{\pi'}(s, a) \right] \right| = \left| \mathbb{E}_{a \sim \pi'} \left[ c(s, a) + \tilde{c}_u^{\pi'}(s, a) + \gamma q_u^{\pi'}(s') - q_u^{\pi'}(s) \right] \right| \tag{146}$$

$$= \left| \mathbb{E}_{a \sim \pi'} \left[ \frac{p_{X^{\pi'}(s')} \left( \frac{q_u^{\pi'}(s) - c(s,a)}{\gamma} \right)}{\gamma p_{X^{\pi'}(s)} \left( q_u^{\pi'}(s) \right)} \left\{ c(s, a) + \gamma q_u^{\pi'}(s') - q_u^{\pi'}(s) \right\} \right] \right| \tag{147}$$

$$\leq \frac{\epsilon R}{(1-\epsilon)} \tag{148}$$

Therefore,

$$\left| \mathbb{E}_\pi \left[ A_u^{\pi'}(s, a) \right] \right| \overset{(a)}{\leq} \left| \mathbb{E}_{a \sim \pi} \left[ A_u^{\pi'}(s, a) \right] - \mathbb{E}_{a' \sim \pi'} \left[ A_u^{\pi'}(s, a') \right] \right| + \left| \mathbb{E}_{\pi'} \left[ A_u^{\pi'}(s, a) \right] \right| \tag{149}$$

$$\overset{(b)}{\leq} \left| \mathbb{E}_{a \sim \pi} \left[ A_u^{\pi'}(s, a) \right] - \mathbb{E}_{a' \sim \pi'} \left[ A_u^{\pi'}(s, a') \right] \right| + \frac{\epsilon R}{(1-\epsilon)} \tag{150}$$

$$= \left| \mathbb{E}_{(a,a') \sim (\pi, \pi')} \left[ A_u^{\pi'}(s, a) - A_u^{\pi'}(s, a') \right] \right| + \frac{\epsilon R}{(1-\epsilon)} \tag{151}$$

$$= \left| Pr\,(a = a')\, \mathbb{E}_{(a,a') \sim (\pi, \pi')|_{a=a'}} \left[ A_u^{\pi'}(s, a) - A_u^{\pi'}(s, a') \right] \right| \tag{152}$$

$$+ \Pr\left(a \neq a'\right) \mathbb{E}_{(a,a') \sim (\pi,\pi')|_{a \neq a'}} \left[ A_u^{\pi'}(s,a) - A_u^{\pi'}(s,a') \right] \bigg| + \frac{\epsilon R}{(1 - \epsilon)} \quad (153)$$

$$= \Pr\left(a \neq a'\right) \left| \mathbb{E}_{(a,a') \sim (\pi,\pi')|_{a \neq a'}} \left[ A_u^{\pi'}(s,a) - A_u^{\pi'}(s,a') \right] \right| + \frac{\epsilon R}{(1 - \epsilon)} \quad (154)$$

$$\overset{(c)}{\leq} 2\alpha \max_{s,a} \left| A_u^{\pi'}(s,a) \right| + \frac{\epsilon R}{(1 - \epsilon)} \quad (155)$$

where (a) holds by the triangular inequality, (b) holds by (148), and (c) holds since $\pi$ and $\pi'$ are $\alpha$-coupled policies. $\qquad \square$

**Lemma 4** (Similar to Lemma 3 in [19]). *Under deterministic dynamics $s_{t+1} = h(s_t, a_t)$ and Assumptions 1 and 2, the following holds for $\alpha$-coupled policies $\pi$ and $\pi'$*

$$\left| \mathbb{E}_{s_t \sim \pi} \left[ \mathbb{E}_{a \sim \pi} \left[ A_u^{\pi'}(s_t, a) \right] \right] - \mathbb{E}_{s_t \sim \pi'} \left[ \mathbb{E}_{a \sim \pi} \left[ A_u^{\pi'}(s_t, a) \right] \right] \right| \quad (156)$$

$$\leq \left( 1 - (1 - \alpha)^t \right) \left\{ 4\alpha \max_{s,a} \left| A_u^{\pi'}(s,a) \right| + \frac{2\epsilon R}{(1 - \epsilon)} \right\} \quad (157)$$

*Proof.* (Similar to the proof of Lemma 3 in [19]) For $\alpha$-coupled policies $\pi$ and $\pi'$, first we consider trajectories drawn from each policy, i.e., $\tau = (s_0, a_0, s_1, a_1, \ldots) \sim \pi$ and $\tau' = (s_0, a_0', s_1', a_1', \ldots) \sim \pi'$. We consider the timestep $t$ and observe the advantage of $\pi'$ over $\pi$. Let define $n_t$ as the number of times that mismatched actions occurs, $a_i \neq a_i'$ for $i < t$. Then

$$\mathbb{E}_{s_t \sim \pi} \left[ \mathbb{E}_{a \sim \pi(\cdot|s_t)} \left[ A_u^{\pi'}(s_t, a) \right] \right] \quad (158)$$

$$= P(n_t = 0) \cdot \mathbb{E}_{s_t \sim \pi|n_t=0} \left[ \mathbb{E}_{a \sim \pi(\cdot|s_t)} \left[ A_u^{\pi'}(s_t, a) \right] \right]$$

$$+ P(n_t > 0) \cdot \mathbb{E}_{s_t \sim \pi|n_t>0} \left[ \mathbb{E}_{a \sim \pi(\cdot|s_t)} \left[ A_u^{\pi'}(s_t, a) \right] \right] \quad (159)$$

$$\mathbb{E}_{s_t \sim \pi'} \left[ \mathbb{E}_{a \sim \pi(\cdot|s_t)} \left[ A_u^{\pi'}(s_t, a) \right] \right] \quad (160)$$

$$= P(n_t = 0) \cdot \mathbb{E}_{s_t \sim \pi'|n_t=0} \left[ \mathbb{E}_{a \sim \pi(\cdot|s_t)} \left[ A_u^{\pi'}(s_t, a) \right] \right]$$

$$+ P(n_t > 0) \cdot \mathbb{E}_{s_t \sim \pi'|n_t>0} \left[ \mathbb{E}_{a \sim \pi(\cdot|s_t)} \left[ A_u^{\pi'}(s_t, a) \right] \right] \quad (161)$$

For the case $n_t = 0$,

$$\mathbb{E}_{s_t \sim \pi|n_t=0} \left[ \mathbb{E}_{a \sim \pi(\cdot|s_t)} \left[ A_u^{\pi'}(s_t, a) \right] \right] = \mathbb{E}_{s_t \sim \pi'|n_t=0} \left[ \mathbb{E}_{a \sim \pi(\cdot|s_t)} \left[ A_u^{\pi'}(s_t, a) \right] \right] \quad (162)$$

Thus by subtracting (161) and (159), we can obtain

$$\mathbb{E}_{s_t \sim \pi} \left[ \mathbb{E}_{a \sim \pi(\cdot|s_t)} \left[ A_u^{\pi'}(s_t, a) \right] \right] - \mathbb{E}_{s_t \sim \pi'} \left[ \mathbb{E}_{a \sim \pi(\cdot|s_t)} \left[ A_u^{\pi'}(s_t, a) \right] \right] \quad (163)$$

$$= P(n_t > 0) \cdot \left( \mathbb{E}_{s_t \sim \pi|n_t>0} \left[ \mathbb{E}_{a \sim \pi(\cdot|s_t)} \left[ A_u^{\pi'}(s_t, a) \right] \right] - \mathbb{E}_{s_t \sim \pi'|n_t>0} \left[ \mathbb{E}_{a \sim \pi(\cdot|s_t)} \left[ A_u^{\pi'}(s_t, a) \right] \right] \right) \quad (164)$$

From the definition of $\alpha$-coupled policy, we get

$$P(n_t = 0) \geq (1 - \alpha)^t, \qquad P(n_t > 0) \leq 1 - (1 - \alpha)^t \quad (165)$$

Then note that

$$\left| \mathbb{E}_{s_t \sim \pi|n_t>0} \left[ \mathbb{E}_{a \sim \pi(\cdot|s_t)} \left[ A_u^{\pi'}(s_t, a) \right] \right] - \mathbb{E}_{s_t \sim \pi'|n_t>0} \left[ \mathbb{E}_{a \sim \pi(\cdot|s_t)} \left[ A_u^{\pi'}(s_t, a) \right] \right] \right| \quad (166)$$

$$\overset{(a)}{\leq} \left| \mathbb{E}_{s_t \sim \pi|n_t>0} \left[ \mathbb{E}_{a \sim \pi} \left[ A_u^{\pi'}(s_t, a) \right] \right] \right| + \left| \mathbb{E}_{s_t \sim \pi'|n_t>0} \left[ \mathbb{E}_{a \sim \pi} \left[ A_u^{\pi'}(s_t, a) \right] \right] \right| \quad (167)$$

$$\leq 2 \max_s \left| \mathbb{E}_{a \sim \pi} \left[ A_u^{\pi'}(s, a) \right] \right| \quad (168)$$

$$\overset{(b)}{\leq} 4\alpha \max_{s,a} \left| A_u^{\pi'}(s,a) \right| + \frac{2\epsilon R}{(1 - \epsilon)} \quad (169)$$

where (a) holds by the triangular inequality, and (b) holds by Lemma 3. Therefore using (164), (165), and (169), we can conclude

$$\mathbb{E}_{s_t \sim \pi} \left[ \mathbb{E}_{a \sim \pi(\cdot|s_t)} \left[ A_u^{\pi'}(s_t, a) \right] \right] - \mathbb{E}_{s_t \sim \pi'} \left[ \mathbb{E}_{a \sim \pi(\cdot|s_t)} \left[ A_u^{\pi'}(s_t, a) \right] \right] \tag{170}$$

$$\leq \left(1 - (1-\alpha)^t\right) \left\{ 4\alpha \max_{s,a} \left| A_u^{\pi'}(s, a) \right| + \frac{2\epsilon R}{(1-\epsilon)} \right\} \tag{171}$$

$$\square$$

Now we define $L_u^{\pi'}(\pi)$ as

$$L_u^{\pi'}(\pi) := q_u^{\pi'}(s_0) + \mathbb{E}_{\pi'} \left[ \sum_{t=0}^{\infty} \gamma^t \mathbb{E}_{a \sim \pi} \left[ A_u^{\pi'}(s_t, a) \right] \right] \tag{172}$$

$$= q_u^{\pi'}(s_0) + \mathbb{E}_{\pi'} \left[ \sum_{t=0}^{\infty} \gamma^t \mathbb{E}_{a \sim \pi} \left[ c(s_t, a) + \tilde{c}_u^{\pi'}(s_t, a) + \gamma q_u^{\pi'}(s_{t+1}) - q_u^{\pi'}(s_t) \right] \right] \tag{173}$$

Then note that

$$L_u^{\pi'}(\pi') = q_u^{\pi'}(s_0) + \mathbb{E}_{\pi'} \left[ \sum_{t=0}^{\infty} \gamma^t \left\{ c(s_t, a_t) + \tilde{c}_u^{\pi'}(s_t, a_t) + \gamma q_u^{\pi'}(s_{t+1}) - q_u^{\pi'}(s_t) \right\} \right] \tag{174}$$

$$= \mathbb{E}_{\pi'} \left[ \sum_{t=0}^{\infty} \gamma^t \left\{ c(s_t, a_t) + \tilde{c}_u^{\pi'}(s_t, a_t) \right\} \right] \tag{175}$$

Then from Theorem 2,

$$\left| q_u^{\pi'}(s_0) - L_u^{\pi'}(\pi') \right| = \left| q_u^{\pi'}(s_0) - \mathbb{E}_{\pi'} \left[ \sum_{t=0}^{\infty} \gamma^t \left\{ c(s_t, a_t) + \tilde{c}_u^{\pi'}(s_t, a_t) \right\} \right] \right| \tag{176}$$

$$\leq \frac{\epsilon R}{(1-\epsilon)(1-\gamma)}. \tag{177}$$

Therefore, we get

$$q_u^{\pi'}(s_0) \geq L_u^{\pi'}(\pi') - \frac{\epsilon R}{(1-\epsilon)(1-\gamma)} \tag{178}$$

**Proposition 1.** *Under deterministic dynamics $s_{t+1} = h(s_t, a_t)$ and Assumptions 1, 2, and 3, the following holds*

$$q_u^{\pi}(s_0) \leq L_u^{\pi'}(\pi) + C_1 \max_s KL(\pi'(\cdot|s) \parallel \pi(\cdot|s)) + C_2 \frac{\epsilon}{1-\epsilon} \tag{179}$$

*where*

$$C_1 = \left( \frac{4\gamma \max_{s,a} \left| A_u^{\pi'}(s, a) \right| + \gamma R}{(1-\gamma)^2} + \frac{C_u}{1-\gamma} \right), \qquad C_2 = \frac{R}{(1-\gamma)^2} \tag{180}$$

*Proof.* Let define $B = \max_{s,a} \left| A_u^{\pi'}(s, a) \right|$. Remind that the definition of the advantage for the $u$-quantile (144) $A_u^{\pi'}(s, a) := c(s, a) + \tilde{c}_u^{\pi'}(s, a) + \gamma \mathbb{E}_{s'} \left[ q_u^{\pi'}(s') \right] - q_u^{\pi'}(s)$, Corollary 2

$$\left| q_u^{\pi}(s_0) - \left\{ q_u^{\pi'}(s_0) + \mathbb{E}_{\pi} \left[ \sum_{t=0}^{\infty} \gamma^t \underbrace{\left( c(s_t, a_t) + \tilde{c}_u^{\pi'}(s_t, a_t) + \gamma q_u^{\pi'}(s_{t+1}) - q_u^{\pi'}(s_t) \right)}_{=A_u^{\pi'}(s_t, a_t)} \right] \right\} \right|$$

$$\leq \frac{\epsilon R}{(1-\epsilon)(1-\gamma)} + \frac{C_u}{1-\gamma} \max_s KL(\pi'(\cdot|s) \parallel \pi(\cdot|s)) \tag{181}$$

and the definition of $L_u^{\pi'}(\pi)$ in (172)

$$L_u^{\pi'}(\pi) := q_u^{\pi'}(s_0) + \mathbb{E}_{\pi'}\left[\sum_{t=0}^{\infty} \gamma^t \mathbb{E}_{a\sim\pi}\left[A_u^{\pi'}(s_t, a)\right]\right] \tag{182}$$

Then we can obtain

$$\left|q_u^{\pi}(s_0) - L_u^{\pi'}(\pi)\right| \tag{183}$$

$$\overset{(a)}{\leq} \left|q_u^{\pi}(s_0) - \left\{q_u^{\pi'}(s_0) + \mathbb{E}_{\pi}\left[\sum_{t=0}^{\infty} \gamma^t A_u^{\pi'}(s_t, a_t)\right]\right\}\right| \tag{184}$$

$$+ \left|\left\{q_u^{\pi'}(s_0) + \mathbb{E}_{\pi}\left[\sum_{t=0}^{\infty} \gamma^t A_u^{\pi'}(s_t, a_t)\right]\right\} - L_u^{\pi'}(\pi)\right| \tag{185}$$

$$\overset{(b)}{\leq} \frac{\epsilon R}{(1-\epsilon)(1-\gamma)} + \frac{C_u}{1-\gamma}\max_s \mathrm{KL}(\pi'(\cdot|s) \,\|\, \pi(\cdot|s)) \tag{186}$$

$$+ \left|\left\{q_u^{\pi'}(s_0) + \mathbb{E}_{\pi}\left[\sum_{t=0}^{\infty} \gamma^t A_u^{\pi'}(s_t, a_t)\right]\right\} - L_u^{\pi'}(\pi)\right| \tag{187}$$

$$= \left|\left\{q_u^{\pi'}(s_0) + \mathbb{E}_{\pi}\left[\sum_{t=0}^{\infty} \gamma^t A_u^{\pi'}(s_t, a_t)\right]\right\} - \left\{q_u^{\pi'}(s_0) + \mathbb{E}_{\pi'}\left[\sum_{t=0}^{\infty} \gamma^t \mathbb{E}_{a\sim\pi}\left[A_u^{\pi'}(s_t, a)\right]\right]\right\}\right| \tag{188}$$

$$+ \frac{\epsilon R}{(1-\epsilon)(1-\gamma)} + \frac{C_u}{1-\gamma}\max_s \mathrm{KL}(\pi'(\cdot|s) \,\|\, \pi(\cdot|s)) \tag{189}$$

$$= \underbrace{\left|\mathbb{E}_{\pi}\left[\sum_{t=0}^{\infty} \gamma^t \mathbb{E}_{a\sim\pi}\left[A_u^{\pi'}(s_t, a)\right]\right] - \mathbb{E}_{\pi'}\left[\sum_{t=0}^{\infty} \gamma^t \mathbb{E}_{a\sim\pi}\left[A_u^{\pi'}(s_t, a)\right]\right]\right|}_{(c)} \tag{190}$$

$$+ \frac{\epsilon R}{(1-\epsilon)(1-\gamma)} + \frac{C_u}{1-\gamma}\max_s \mathrm{KL}(\pi'(\cdot|s) \,\|\, \pi(\cdot|s)) \tag{191}$$

where (a) holds by the triangular inequality, (b) holds from (181). The term (c) can be written as

$$\left|\mathbb{E}_{\pi}\left[\sum_{t=0}^{\infty} \gamma^t \mathbb{E}_{a\sim\pi}\left[A_u^{\pi'}(s_t, a)\right]\right] - \mathbb{E}_{\pi'}\left[\sum_{t=0}^{\infty} \gamma^t \mathbb{E}_{a\sim\pi}\left[A_u^{\pi'}(s_t, a)\right]\right]\right| \tag{192}$$

$$\overset{(a)}{\leq} \sum_{t=0}^{\infty} \gamma^t \left|\mathbb{E}_{s_t\sim\pi}\left[\mathbb{E}_{a\sim\pi}\left[A_u^{\pi'}(s, a)\right]\right] - \mathbb{E}_{s_t\sim\pi'}\left[\mathbb{E}_{a\sim\pi}\left[A_u^{\pi'}(s, a)\right]\right]\right| \tag{193}$$

$$\overset{(b)}{\leq} \sum_{t=0}^{\infty} \gamma^t \left(1 - (1-\alpha)^t\right)\left\{4\alpha\max_{s,a}\left|A_u^{\pi'}(s, a)\right| + \frac{2\epsilon R}{(1-\epsilon)}\right\} \tag{194}$$

$$= \left(4\alpha\max_{s,a}\left|A_u^{\pi'}(s, a)\right| + \frac{2\epsilon R}{(1-\epsilon)}\right)\left(\frac{1}{1-\gamma} - \frac{1}{1-\gamma(1-\alpha)}\right) \tag{195}$$

$$= \left(4\alpha\max_{s,a}\left|A_u^{\pi'}(s, a)\right| + \frac{2\epsilon R}{(1-\epsilon)}\right)\frac{\alpha\gamma}{(1-\gamma)(1-\gamma(1-\alpha))} \tag{196}$$

$$\overset{(c)}{\leq} \left(4\alpha\max_{s,a}\left|A_u^{\pi'}(s, a)\right| + \frac{2\epsilon R}{(1-\epsilon)}\right)\frac{\alpha\gamma}{(1-\gamma)^2} \tag{197}$$

where (a) holds by the triangular inequality, (b) holds by Lemma 4, and (c) holds by $\alpha < 1$ ($\alpha$ is for $\alpha$-coupled policy). Therefore by putting (197) into term (c) in (191), we can obtain

$$\left|q_u^{\pi}(s_0) - L_u^{\pi'}(\pi)\right| \tag{198}$$

$$= \underbrace{\left|\mathbb{E}_{\pi}\left[\sum_{t=0}^{\infty} \gamma^t \mathbb{E}_{a\sim\pi}\left[A_u^{\pi'}(s_t, a)\right]\right] - \mathbb{E}_{\pi'}\left[\sum_{t=0}^{\infty} \gamma^t \mathbb{E}_{a\sim\pi}\left[A_u^{\pi'}(s_t, a)\right]\right]\right|}_{(c)} \tag{199}$$

$$+ \frac{\epsilon R}{(1-\epsilon)(1-\gamma)} + \frac{C_u}{1-\gamma} \max_s \mathrm{KL}(\pi'(\cdot|s) \parallel \pi(\cdot|s)) \tag{200}$$

$$\leq \left( 4\alpha \max_{s,a} \left| A_u^{\pi'}(s,a) \right| + \frac{2\epsilon R}{(1-\epsilon)} \right) \frac{\alpha\gamma}{(1-\gamma)^2} + \frac{\epsilon R}{(1-\epsilon)(1-\gamma)} + \frac{C_u}{1-\gamma} \max_s \mathrm{KL}(\pi'(\cdot|s) \parallel \pi(\cdot|s)) \tag{201}$$

$$= \left( \frac{4\gamma \max_{s,a} \left| A_u^{\pi'}(s,a) \right|}{(1-\gamma)^2} \right) \alpha^2 + \left( 2\alpha \frac{\epsilon}{1-\epsilon} \right) \cdot \frac{\gamma R}{(1-\gamma)^2} \tag{202}$$

$$+ \left( \frac{(1-\gamma)R}{(1-\epsilon)(1-\gamma)^2} \right) \epsilon + \frac{C_u}{1-\gamma} \max_s \mathrm{KL}(\pi'(\cdot|s) \parallel \pi(\cdot|s)) \tag{203}$$

$$\overset{(a)}{\leq} \left( \frac{4\gamma \max_{s,a} \left| A_u^{\pi'}(s,a) \right|}{(1-\gamma)^2} \right) \alpha^2 + \left\{ \alpha^2 + \left( \frac{\epsilon}{1-\epsilon} \right)^2 \right\} \cdot \frac{\gamma R}{(1-\gamma)^2} \tag{204}$$

$$+ \left( \frac{(1-\gamma)R}{(1-\epsilon)(1-\gamma)^2} \right) \epsilon + \frac{C_u}{1-\gamma} \max_s \mathrm{KL}(\pi'(\cdot|s) \parallel \pi(\cdot|s)) \tag{205}$$

$$= \left( \frac{4\gamma \max_{s,a} \left| A_u^{\pi'}(s,a) \right| + \gamma R}{(1-\gamma)^2} \right) \alpha^2 + \frac{C_u}{1-\gamma} \max_s \mathrm{KL}(\pi'(\cdot|s) \parallel \pi(\cdot|s)) \tag{206}$$

$$+ \left( \frac{(1-\gamma)R}{(1-\gamma)^2} \right) \left( \frac{\epsilon}{1-\epsilon} \right) + \frac{\gamma R}{(1-\gamma)^2} \left( \frac{\epsilon}{1-\epsilon} \right)^2 \tag{207}$$

$$\overset{(b)}{\leq} \left( \frac{4\gamma \max_{s,a} \left| A_u^{\pi'}(s,a) \right| + \gamma R}{(1-\gamma)^2} \right) \alpha^2 + \frac{C_u}{1-\gamma} \max_s \mathrm{KL}(\pi'(\cdot|s) \parallel \pi(\cdot|s)) \tag{208}$$

$$+ \left( \frac{(1-\gamma)R}{(1-\gamma)^2} \right) \left( \frac{\epsilon}{1-\epsilon} \right) + \frac{\gamma R}{(1-\gamma)^2} \left( \frac{\epsilon}{1-\epsilon} \right) \tag{209}$$

$$= \frac{R}{(1-\gamma)^2} \left( \frac{\epsilon}{1-\epsilon} \right) + \left( \frac{4\gamma \max_{s,a} \left| A_u^{\pi'}(s,a) \right| + \gamma R}{(1-\gamma)^2} \right) \alpha^2 + \frac{C_u}{1-\gamma} \max_s \mathrm{KL}(\pi'(\cdot|s) \parallel \pi(\cdot|s)) \tag{210}$$

where (a) holds by the inequality of arithmetic and geometric means, and (b) holds from the definition of $0 < \epsilon < \frac{1}{2}$ in Assumption 2. Like [19], if we take $\alpha$ as the maximum of the total variation of two policies $\pi$ and $\pi'$, i.e., $\alpha = \max_s D_{TV}(\pi'(\cdot|s)||\pi(\cdot|s))$, then these policies are $\alpha$-coupled. Since $D_{TV}(\pi'(\cdot|s)||\pi(\cdot|s))^2 \leq \mathrm{KL}(\pi'(\cdot|s)||\pi(\cdot|s))$, eq. (210) becomes

$$\left| q_u^\pi(s_0) - L_u^{\pi'}(\pi) \right| \leq C_1 \max_s \mathrm{KL}(\pi'(\cdot|s) \parallel \pi(\cdot|s)) + C_2 \frac{\epsilon}{1-\epsilon} \tag{211}$$

where

$$C_1 = \left( \frac{4\gamma \max_{s,a} \left| A_u^{\pi'}(s,a) \right| + \gamma R}{(1-\gamma)^2} + \frac{C_u}{1-\gamma} \right), \qquad C_2 = \frac{R}{(1-\gamma)^2} \tag{212}$$

$\square$

Together with Proposition 1 above, and Theorem 1 in [19], we can obtain the Theorem 4 for policy improvement condition.

**Theorem 4.** *Let $\pi_{new} := \pi_{\theta_{new}}$ be the solution of the problem of maximizing*

$$L^{\pi_{old}}(\pi_\theta) - \tilde{C}_1 \max_s KL(\pi_{old}(\cdot|s) \parallel \pi_\theta(\cdot|s)), \tag{213}$$

*where*

$$L^{\pi_{old}}(\pi_\theta) = L_r^{\pi_{old}}(\pi_\theta) - \lambda L_{1-\epsilon_0}^{\pi_{old}}(\pi_\theta) \tag{214}$$

$$= \left(V^{\pi_{old}}(s_0) - \lambda q_{1-\epsilon_0}^{\pi_{old}}(s_0)\right) + \mathbb{E}_{\pi_{old}}\left[\sum_{t=0}^{\infty} \gamma^t \mathbb{E}_{a \sim \pi_\theta}\left[A_r^{\pi_{old}}(s_t, a) - \lambda A_{1-\epsilon_0}^{\pi_{old}}(s_t, a)\right]\right], \tag{215}$$

*and some constant $\tilde{C}_1 > 0$. Then, under deterministic dynamics $s_{t+1} = h(s_t, a_t)$ and Assumptions 1, 2, and 3, the following inequality holds:*

$$L_{quant}(\pi_{new}, \lambda) - L_{quant}(\pi_{old}, \lambda) \tag{216}$$

$$\geq L^{\pi_{old}}(\pi_{new}) - L^{\pi_{old}}(\pi_{old}) - \tilde{C}_1 KL_{max}(\pi_{old}||\pi_{new}) - \underbrace{\tilde{C}_2 \frac{\epsilon}{1-\epsilon}}_{\text{approximation loss}} \tag{217}$$

*for a given Lagrange multiplier $\lambda > 0$, some constant $\tilde{C}_2$ and small $\epsilon > 0$.*

Remind that Theorem 1 of [19] with our notation:

**Theorem 5** (Theorem 1 of [19])**.**

$$V^\pi(s_0) \geq \underbrace{V^{\pi'}(s_0) + \mathbb{E}_{\pi'}\left[\sum_{t=0}^{\infty} \gamma^t \mathbb{E}_{a \sim \pi}\left[A_r^{\pi'}(s_t, a)\right]\right]}_{=:L_r^{\pi'}(\pi)} - C_3 \max_s KL(\pi'(\cdot|s) \| \pi(\cdot|s)) \tag{218}$$

$$= L_r^{\pi'}(\pi) - C_3 \max_s KL(\pi'(\cdot|s) \| \pi(\cdot|s)) \tag{219}$$

*where*

$$C_3 = \frac{4\gamma \max_{s,a}\left|A_r^{\pi'}(s, a)\right|}{(1-\gamma)^2} \tag{220}$$

$$A_r^{\pi'}(s, a) := r(s, a) + \gamma \mathbb{E}_{s'}\left[V^{\pi'}(s')\right] - V^{\pi'}(s) \tag{221}$$

We omit the proof of Theorem 5. Please see [19] for the proof. Note that Theorem 5 holds for any two policies $\pi$ and $\pi'$ as we can see in the appendix of the original paper [19].

Finally now we prove Theorem 4.

*Proof of Theorem 4.* From Proposition 1, we have

$$q_u^\pi(s_0) \leq \underbrace{q_u^{\pi'}(s_0) + \mathbb{E}_{\pi'}\left[\sum_{t=0}^{\infty} \gamma^t \mathbb{E}_{a \sim \pi}\left[A_u^{\pi'}(s_t, a)\right]\right]}_{L_u^{\pi'}(\pi)} + C_1 \max_s KL(\pi'(\cdot|s) \| \pi(\cdot|s)) + C_2 \frac{\epsilon}{1-\epsilon}$$

$$\tag{222}$$

for

$$A_u^{\pi'}(s, a) := c(s, a) + \tilde{c}_u^{\pi'}(s, a) + \gamma \mathbb{E}_{s'}\left[q_u^{\pi'}(s')\right] - q_u^{\pi'}(s) \tag{223}$$

$$C_1 = \left(\frac{4\gamma \max_{s,a}\left|A_u^{\pi'}(s, a)\right| + \gamma R}{(1-\gamma)^2} + \frac{C_u}{1-\gamma}\right) \tag{224}$$

$$C_2 = \frac{R}{(1-\gamma)^2}, \tag{225}$$

and from Theorem 1 in [19] (or Theorem 5 in this appendix), we have

$$V^\pi(s_0) \geq \underbrace{V^{\pi'}(s_0) + \mathbb{E}_{\pi'}\left[\sum_{t=0}^{\infty} \gamma^t \mathbb{E}_{a \sim \pi}\left[A_r^{\pi'}(s_t, a)\right]\right]}_{=:L_r^{\pi'}(\pi)} - C_3 \max_s KL(\pi'(\cdot|s) \| \pi(\cdot|s)) \tag{226}$$

where

$$A_r^{\pi'}(s, a) := r(s, a) + \gamma \mathbb{E}_{s'}\left[V^{\pi'}(s')\right] - V^{\pi'}(s) \tag{227}$$

$$C_3 = \frac{4\gamma \max_{s,a}\left|A_r^{\pi'}(s, a)\right|}{(1 - \gamma)^2}. \tag{228}$$

For a given $\lambda > 0$, by subtracting $\lambda \times$ (222) from (226), then we have

$$V^\pi(s_0) - \lambda q_u^\pi(s_0)$$

$$\geq L_r^{\pi'}(\pi) - \lambda L_u^{\pi'}(\pi) - (\lambda C_1 + C_3) \max_s \mathrm{KL}(\pi'(\cdot|s) \parallel \pi(\cdot|s)) - \lambda C_2 \frac{\epsilon}{1 - \epsilon} \tag{229}$$

$$= V^{\pi'}(s_0) + \mathbb{E}_{\pi'}\left[\sum_{t=0}^{\infty} \gamma^t \mathbb{E}_{a\sim\pi}\left[A_r^{\pi'}(s_t, a)\right]\right] - \lambda\left\{q_u^{\pi'}(s_0) + \mathbb{E}_{\pi'}\left[\sum_{t=0}^{\infty} \gamma^t \mathbb{E}_{a\sim\pi}\left[A_u^{\pi'}(s_t, a)\right]\right]\right\}$$

$$- (\lambda C_1 + C_3) \max_s \mathrm{KL}(\pi'(\cdot|s) \parallel \pi(\cdot|s)) - \lambda C_2 \frac{\epsilon}{1 - \epsilon} \tag{230}$$

$$= \left(V^{\pi'}(s_0) - \lambda q_u^{\pi'}(s_0)\right) + \underbrace{\mathbb{E}_{\pi'}\left[\sum_{t=0}^{\infty} \gamma^t \mathbb{E}_{a\sim\pi}\left[A_r^{\pi'}(s_t, a) - \lambda A_u^{\pi'}(s_t, a)\right]\right]}_{=:L^{\pi'}(\pi)} \tag{231}$$

$$- (\lambda C_1 + C_3) \max_s \mathrm{KL}(\pi'(\cdot|s) \parallel \pi(\cdot|s)) - \lambda C_2 \frac{\epsilon}{1 - \epsilon} \tag{232}$$

$$= L^{\pi'}(\pi) - (\lambda C_1 + C_3) \max_s \mathrm{KL}(\pi'(\cdot|s) \parallel \pi(\cdot|s)) - \lambda C_2 \frac{\epsilon}{1 - \epsilon} \tag{233}$$

Therefore now we have

$$L_{quant}(\pi, \lambda) = V^\pi(s_0) - \lambda\left(q_u^\pi(s_0) - d_{th}\right) \tag{234}$$

$$\geq L^{\pi'}(\pi) + \lambda \cdot d_{th} - (\lambda C_1 + C_3) \max_s \mathrm{KL}(\pi'(\cdot|s) \parallel \pi(\cdot|s)) - \lambda C_2 \frac{\epsilon}{1 - \epsilon} \tag{235}$$

Note that

$$L_r^{\pi'}(\pi') = V^{\pi'}(s_0) + \mathbb{E}_{\pi'}\left[\sum_{t=0}^{\infty} \gamma^t \mathbb{E}_{a'\sim\pi'}\left[A_r^{\pi'}(s_t, a')\right]\right] \tag{236}$$

$$= V^{\pi'}(s_0) \tag{237}$$

$$L_u^{\pi'}(\pi') = q_u^{\pi'}(s_0) + \mathbb{E}_{\pi'}\left[\sum_{t=0}^{\infty} \gamma^t \mathbb{E}_{a'\sim\pi'}\left[A_u^{\pi'}(s_t, a')\right]\right] \tag{238}$$

$$\overset{(a)}{\leq} q_u^{\pi'}(s_0) + \frac{\epsilon R}{(1 - \epsilon)(1 - \gamma)} \tag{239}$$

where (a) holds by (178). Therefore,

$$L^{\pi'}(\pi') = L_r^{\pi'}(\pi') - \lambda L_u^{\pi'}(\pi') \tag{240}$$

$$\geq V^{\pi'}(s_0) - \lambda\left(q_u^{\pi'}(s_0) + \frac{\epsilon R}{(1 - \epsilon)(1 - \gamma)}\right) \tag{241}$$

$$= V^{\pi'}(s_0) - \lambda\left(q_u^{\pi'}(s_0) - d_{th}\right) - \lambda\left(d_{th} + \frac{\epsilon R}{(1 - \epsilon)(1 - \gamma)}\right) \tag{242}$$

$$= L_{quant}(\pi', \lambda) - \lambda \cdot d_{th} - \lambda \frac{\epsilon R}{(1 - \epsilon)(1 - \gamma)}. \tag{243}$$

By rearranging this, we get

$$-L_{quant}(\pi', \lambda) \geq -L^{\pi'}(\pi') - \lambda \cdot d_{th} - \lambda \frac{\epsilon R}{(1 - \epsilon)(1 - \gamma)}. \tag{244}$$

Therefore by adding (244) and (235), we can conclude

$$L_{quant}(\pi, \lambda) - L_{quant}(\pi', \lambda) \geq L^{\pi'}(\pi) - L^{\pi'}(\pi') - (\lambda C_1 + C_3) \max_s \mathrm{KL}(\pi'(\cdot|s) \parallel \pi(\cdot|s)) \tag{245}$$

$$- \lambda C_2 \frac{\epsilon}{1 - \epsilon} - \lambda \frac{\epsilon R}{(1 - \epsilon)(1 - \gamma)} \tag{246}$$

$$= L^{\pi'}(\pi) - L^{\pi'}(\pi') - \tilde{C}_1 \max_s \mathrm{KL}(\pi'(\cdot|s) \parallel \pi(\cdot|s)) - \tilde{C}_2 \frac{\epsilon}{1 - \epsilon} \tag{247}$$

where

$$\tilde{C}_1 = \lambda C_1 + C_3 \tag{248}$$

$$= \lambda \left( \frac{4\gamma \max_{s,a} \left| A_u^{\pi'}(s, a) \right| + \gamma R}{(1 - \gamma)^2} + \frac{C_u}{1 - \gamma} \right) + \frac{4\gamma \max_{s,a} \left| A_r^{\pi'}(s, a) \right|}{(1 - \gamma)^2} \tag{249}$$

$$= \frac{4\gamma \left( \max_{s,a} \left| A_r^{\pi'}(s, a) \right| + \lambda \max_{s,a} \left| A_u^{\pi'}(s, a) \right| \right)}{(1 - \gamma)^2} + \lambda \frac{\gamma R}{(1 - \gamma)^2} + \lambda \frac{C_u}{1 - \gamma} \tag{250}$$

$$\tilde{C}_2 = \lambda C_2 + \lambda \frac{R}{1 - \gamma} \tag{251}$$

$$= \lambda \left( \frac{R}{(1 - \gamma)^2} + \frac{R}{1 - \gamma} \right) \tag{252}$$

$$= \lambda \frac{(2 - \gamma) R}{(1 - \gamma)^2} \tag{253}$$

$$\square$$

# C  Detailed Explanation of The Environments

The considered environments are SimpleButtonEnv, DynamicEnv [27], GremlinEnv, and DynamicButtonEnv, which are based on Safety Gym [17], MuJoCo [23], and OpenAI Gym [4]. The experiments are performed on a server with Intel(R) Xeon(R) Gold 6240R CPU @2.40GHz, and each experiment takes $8 \sim 10$ hours. The environments are illustrated in Fig. 7. The goal of these environments is for a robot (red sphere) to reach a goal (the orange sphere wrapped by a grey translucent pillar for SimpleButtonEnv and DynamicButtonEnv, and the green pillar for DynamicEnv and GremlinEnv), while avoiding hazards (blue circles) or the non-goal button (the orange sphere). Once the robot reaches the current goal, the environments generate the next goal deterministically (SimpleButtonEnv) or randomly (DynamicEnv, GremlinEnv, DynamicButtonEnv). When the robot performs an action at time step $t$, it receives a reward $\left\{\|p_{t+1} - p_{\text{goal}}\|_2 - \|p_t - p_{\text{goal}}\|_2\right\} + 1_{\text{goal reached}}$, where $p_t$ is the position $(x, y)$ of the robot at time step $t$ and $p_{\text{goal}}$ is the current goal position at time step $t$. It also receives a cost $+1$ if the robot touches non-goal objects (a hazard or the non-goal button), and 0 otherwise. Hence, for the robot, it receives a higher return when the robot touches more goals in a maximum timesteps $T = 1000$, and causes a higher sum of costs when the robot touches the other objects more often.

**SimpleButtonEnv:** This environment consists of a robot (the red sphere), three hazards (blue pillars), a goal button (the orange sphere wrapped by a grey translucent pillar), and a non-goal button (the orange sphere). When it starts a new episode, it locates the robot randomly in in a restricted region $[x_{min}, x_{max}, y_{min}, y_{max}] = [-1.5, 1.5, -1.5, 1.5]$ and the other objects in a fixed position. When the robot reaches the current goal, it sets the next goal as the non-goal button. Thus, the objective of this environment is to touch two buttons many times iteratively in a fixed maximum timestep.

**DynamicEnv:** This environment consists of a robot (the red sphere), three hazards (blue pillars), and a goal (the green pillar). When it starts a new episode, it locates these objects randomly in a restricted region $[x_{min}, x_{max}, y_{min}, y_{max}] = [-1.5, 1.5, -1.5, 1.5]$. When the robot reaches the current goal, the next goal is generated at a random position.

**GremlinEnv:** This environment consists of a robot (the red sphere), five hazards (blue pillars), three gremlins (purple moving cubes), and a goal (the green pillar). This is similar to DynamicEnv except the gremlins and higher complexity of the task. Each gremlin goes around in a circle, and when the agent touches a gremiln, it receives a cost. When it starts a new episode, it locates these objects randomly in a restricted region $[x_{min}, x_{max}, y_{min}, y_{max}] = [-2, 2, -2, 2]$. When the robot reaches the current goal, the next goal is generated at a random position.

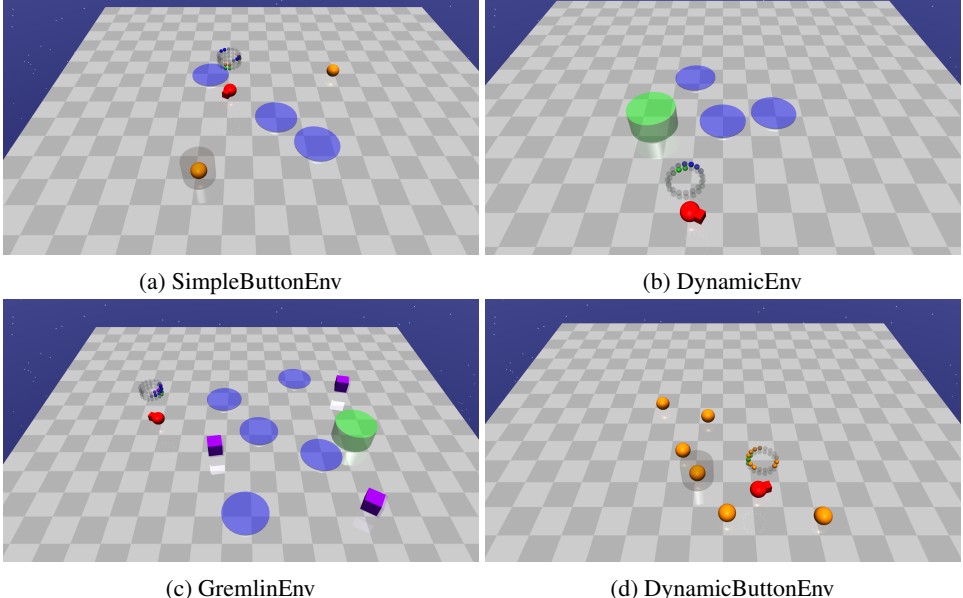

(a) SimpleButtonEnv  (b) DynamicEnv

(c) GremlinEnv  (d) DynamicButtonEnv

Figure 7: The considered environments

**DynamicButtonEnv:** This environment consists of a robot (the red sphere), and goal button (the orange sphere wrapped by a grey translucent pillar), and five non-goal buttons (the orange sphere). When it starts a new episode, it locates these objects randomly in a restricted region $[x_{min}, x_{max}, y_{min}, y_{max}] = [-1.5, 1.5, -1.5, 1.5]$. When the robot reaches the current goal, it sets the next goal randomly among non-goal buttons. This environment is similar to DynamicEnv but the hazards are the non-goal buttons.

**Observation Space:** The observation in these environments is sensor values (accelerometer, velocimeter, gyro, and magnetometer) plus lidar values which measure the distance between the robot and the other objects. There are 16 lidar sensors for each object (a goal, hazards, buttons, gremlins) and these are located around the robot. Each lidar sensor for an object measures the distance between the robot and the object located in its corresponding direction. Gathering all these sensor values, the environment gives these values to the agent as an observation at the current time. The dimensions of the observation spaces are 44 (DynamicEnv, DynamicButtonEnv) and 60 (SimpleButtonEnv and GremlinEnv).

# D    More Results

## D.1    QCPO with Various Target Outage Probability $\epsilon_0$

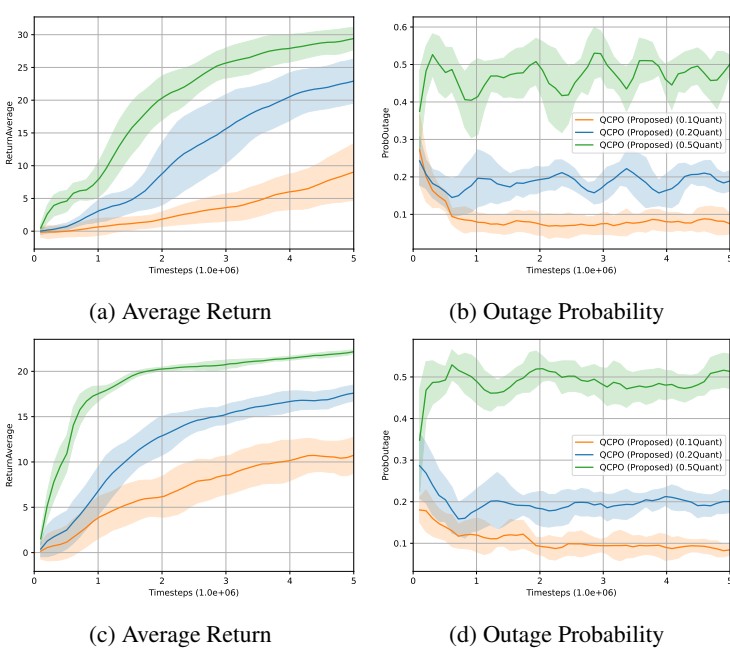

(a) Average Return

(b) Outage Probability

(c) Average Return

(d) Outage Probability

Figure 8: Results of QCPO with $\epsilon_0 = 0.5$ (green), $0.2$ (blue) and $0.1$ (orange) on SimpleButtonEnv (1st row), DynamicEnv (2nd row): (left) average return of the most current 100 episodes and (right) outage probability of the most current 100 episodes.

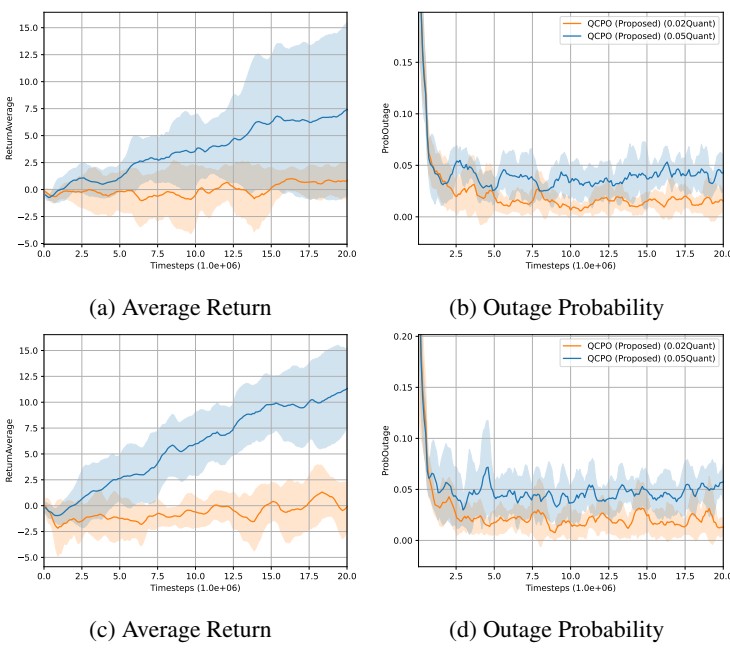

(a) Average Return

(b) Outage Probability

(c) Average Return

(d) Outage Probability

Figure 9: Results of QCPO with $\epsilon_0 = 0.05$ (blue) and $0.02$ (orange) on SimpleButtonEnv (1st row) and DynamicEnv (2nd row): (left) average return of the most current 100 episodes and (right) outage probability of the most current 100 episodes.

In this subsection, we provide results of QCPO with various target outage probabilities $\epsilon_0 = 0.5, 0.2, 0.1, 0.05$ and $0.02$. Fig. 8 shows the average return and the outage probability of QCPO with $\epsilon_0 = 0.5, 0.2$ and $0.1$. It is seen that QCPO satisfies the outage probability constraint after some initial time and then tries to increase the return while satisfying the outage probability constraint. Fig. 9 shows the average return and the outage probability of QCPO with $\epsilon_0 = 0.05$ and $0.02$. In Fig. 9, it is again seen that QCPO satisfies the outage probability constraint after some initial time and then tries to increase the return while satisfying the outage probability constraint. However, it seems that more initial time steps are required than in the case of $\epsilon_0 = 0.5, 0.2$ and $0.1$ to satisfy the target outage probability.

## D.2 WCSAC with Weibull distribution approximation

In this subsection, we provide results of QCPO, WCSAC[27], and WCSAC with Weibull distribution approximation. In Fig. 10, it is seen that WCSAC with Weibull distribution approximation satisfies the outage probability constraint, while the original WCSAC with Gaussian distribution approximation does not. These results can imply that Weibull distribution approximation can estimate the true underlying distribution of the cumulative sum cost better than Gaussian distribution, and this is due to the limited capability of Gaussian distribution to capture the decay rate of the tail probability.

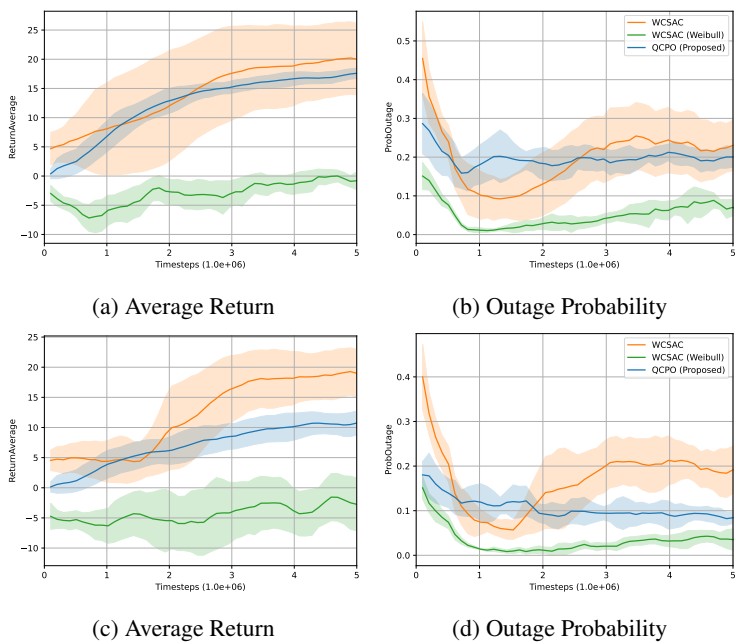

(a) Average Return         (b) Outage Probability

(c) Average Return         (d) Outage Probability

Figure 10: Results of QCPO (blue), WCSAC[27] (orange) and WCSAC with Weibull distribution approximation (green) on DynamicEnv: (1st row) $\epsilon_0 = 0.2$, (2nd row) $\epsilon_0 = 0.1$, (left) average return of the most current 100 episodes and (right) outage probability of the most current 100 episodes.

## D.3 Performance Comparison

Fig. 11 shows the results of the considered algorithms on SimpleButtonEnv, DynamicEnv, GremlinEnv, and DynamicButtonEnv explained in Appendix C. All experiments were done with 10 different random seeds, and the real line and the shaded area represent the average and average $\pm$ standard deviation, respectively. PPO with the Lagrangian multiplier method for (ExpCP) (green) keeps the average of the sum cost around the threshold $d_{th} = 15$ well (see Fig. 11c, 11f, 11i, and 11l), and its outage probability is around $0.35$ on SimpleButtonEnv and DynamicEnv (Fig. 11b and 11e), and $0.3$ on GremlinEnv and DynamicButtonEnv (Fig. 11h and 11k). Note that the CVaR approach (WCSAC) should satisfy a sufficient condition for satisfying the outage probability constraint in (ProbCP). It is seen that WCSAC ($\epsilon_0 = 0.2$ (purple), $\epsilon_0 = 0.1$ (red)) achieves a lower or similar outage probability to the threshold $\epsilon_0$ in Fig. 11b, but the algorithm does not satisfy the outage probability constraint exactly in Fig. 11e, 11h, and 11k. This means that the Gaussian distribution approximation of the

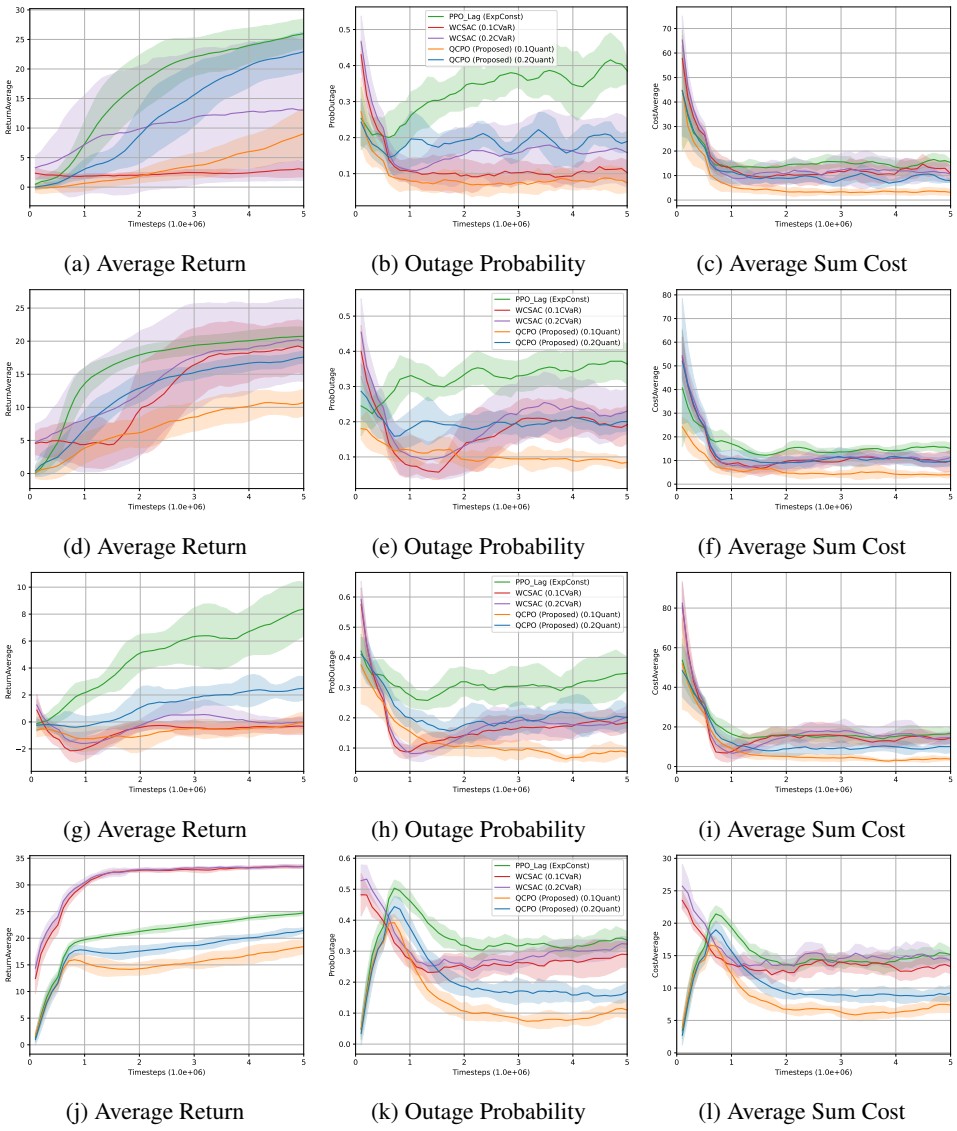

Figure 11: Results on SimpleButtonEnv (1st row), DynamicEnv (2nd row), GremlinEnv (3rd row), and DynamicButtonEnv (4th row): (1st column) average return of the most current 100 episodes, (2nd column) outage probability of the most current 100 episodes, and (3rd column) average sum of costs of the most current 100 episodes.

distribution of $X^\pi(s)$ has limited capability to capture the decay rate of the tail probability. On the other hand, the proposed QCPO ($\epsilon_0 = 0.2$ (blue), $\epsilon_0 = 0.1$ (orange)) maintains the outage probability around the desired target outage probability very well, as shown in Fig. 11b, 11e, 11h, and 11k.

Now consider the average return of these algorithms. In constrained RL, in general, if an algorithm is allowed to have a higher sum of costs, then it has a higher return. Thus, as seen in Fig. 11b, 11e, 11h, and 11k, PPO_Lag induces the highest outage probability, so it has the highest average return, as shown in Fig. 11a, 11d, and 11g. (For DynamicButtonEnv, WCSAC outperforms PPO_Lag, and this is because that SAC, the base algorithm of WCSAC, is a better algorithm than PPO, the base algorithm of PPO_Lag, on most unconstrained environments.) The direct comparison between WCSAC and QCPO is less meaningful in DynamicEnv, GremlinEnv, and DynamicButtonEnv since WCSAC does not satisfy the outage probability constraint, but it is fair in SimpleButtonEnv because both algorithms satisfy the outage probability constraint. As seen in Fig. 11a, QCPO achieves a higher average return than WCSAC for the same target probability constraint $\epsilon_0 = 0.1, 0.2$. This is

because QCPO satisfies the target outage probability exactly, i.e., uses the given cost budget fully for a higher return.

## E Implementation Details

The implementation of the proposed algorithm[5] is based on the implementation of [21][6]

### E.1 Network structures

Since the proposed algorithm is based on PPO [20], the network structure is similar to the network structure of PPO. The networks for the value function, the quantile function, the policy, and the Weibull distribution parameters have a common shared network to extract a feature of its observation. The common network has two MLP layers of size $512$ with the tanh activation function, and an LSTM layer of size $512$ with tanh activation function. The current observation changes to its feature through the two MLP layers, then concatenates this feature of the current observation, the previous action, the previous reward, and the previous cost to input the LSTM network. Thus, the common network outputs a feature of all previous information in the current trajectory. The output of the LSTM layer is then used as the input of the uncommon parts of the functions. The value function for reward has a linear MLP layer of size 1, and the quantile function for cost has a MLP layer of size $n_q$ (number of quantile estimates) with exponential activation $\exp(x)$. Thus, the feature computed by the common feature network goes through these MLP networks to compute its value $V^\pi(s)$ and its quantile $q_u^\pi(s)$ for $u \in \{u_1, u_2, \ldots, u_{n_q}\}$. The policy network has a linear MLP layer of size 1, which outputs the mean parameter of Gaussian distribution, and a variable which indicates state-independent log standard deviation for Gaussian distribution. For the Weibull distribution parameters, there are two networks, one for $\alpha(s)$ and the other for $\beta(s)$, having a MLP layer of size 1. For $\alpha(s)$, the network has 4 * sigmoid activation function, and for $\beta(s)$, the network has the exponential activation function.

### E.2 Loss Functions

The parameters are updated by minimizing their own loss functions. The loss function of the value parameter $\phi$ is

$$L(\phi) = \frac{1}{2}\hat{\mathbb{E}}_{s\sim\rho^\pi}\left[\|V_\phi(s) - R\|^2\right], \tag{254}$$

where $R$ is a sampled return at $s$, and $\hat{\mathbb{E}}$ is the sample mean for $s$ drawn from $\rho^\pi$. This loss function is the same as that of PPO. For the quantile function, the loss function is composed of two losses. The first one is the value parameter loss for cost, defined as

$$L_{value}(\psi) = \frac{1}{2}\hat{\mathbb{E}}_{s\sim\rho^\pi}\left[\left\|\frac{1}{n_q}\sum_{i=1}^{n_q} q_{\psi,u_i}(s) - C\right\|^2\right], \tag{255}$$

where $C$ is the sampled cumulative sum cost at $s$. Note that value function for cost is computed as $C^\pi(s) := \mathbb{E}_\pi\left[\sum_{t=0}^\infty \gamma^t c(s_t, a_t)\right] = \int_0^1 q_u^\pi(s)\, du \approx \frac{1}{n_q}\sum_{i=1}^{n_q} q_{\psi,u_i}(s)$. The second loss for the quantile function is the quantile loss $l_{Huber,u_i}(x)$ (for definition, please see Appendix A.1) with the Huber loss $L_\kappa(x)$, defined as

$$L_{quant}(\psi) = \frac{1}{n_q^2}\sum_{i,j=1}^{n_q} \hat{\mathbb{E}}_{(s,a,s')\sim\pi}\left[l_{Huber,u_i}(\delta_{ij}(s,a,s'))\right]$$

$$\delta_{ij}(s,a,s') = c(s,a) + \gamma q_{\psi_{old},u_j}(s') - q_{\psi,u_i}(s),$$

where $(s,a,s') \sim \pi$ means that $s \sim \rho^\pi(\cdot)$, $a \sim \pi(\cdot|s)$, and $s' \sim M(\cdot|s,a)$, $\psi_{old}$ is a copied parameter of $\psi$ which does not update when $\psi$ updates. Thus, the loss function for the quantile function parameter $\psi$ is given by

$$L(\psi) = L_{value}(\psi) + L_{quant}(\psi). \tag{256}$$

The parameters $\xi$ and $\zeta$ for estimating the Weibull distribution parameters $\alpha_\xi(s)$ and $\beta_\zeta(s)$ at state $s$ are updated by minimizing the following loss function:

$$L(\xi,\zeta) = \hat{\mathbb{E}}_{s\sim\rho^\pi}\left[\frac{1}{k}\sum_{i=n_q-k+1}^{n_q}\frac{1}{2}\left\|\log\beta_\zeta(s) + \frac{\log c_{u_i}}{\alpha_\xi(s)} - \log q_{\psi,u_i}(s)\right\|^2\right], \tag{257}$$

---

[5] https://github.com/wyjung0625/QCPO, (MIT License)
[6] https://github.com/astooke/rlpyt/tree/master/rlpyt/projects/safe, (MIT License)

where $c_u = -\log(1-u)$. Note that the $u$-quantile of Weibull distribution with parameters $\alpha$ and $\beta$ is $\beta \cdot (c_u)^{1/\alpha}$. Thus, (257) is the mean square error of log-scale of the $u$-quantile for $u \in \{u_{n_q-k+1}, \ldots, u_{n_q}\}$.

## E.3 Policy Loss Function

As aforementioned in Section 4, the basic policy loss function of QCPO for a given Lagrange multiplier $\lambda$ is

$$L^{\pi_{old}}(\pi_\theta) - \tilde{C}_1 \max_s \mathrm{KL}(\pi_{old}(\cdot|s) \| \pi_\theta(\cdot|s)) \tag{258}$$

where

$$L^{\pi_{old}}(\pi_\theta) = \left(V^{\pi_{old}}(s_0) - \lambda q_{1-\epsilon_0}^{\pi_{old}}(s_0)\right) + \mathbb{E}_{\pi_{old}}\left[\sum_{t=0}^\infty \gamma^t \mathbb{E}_{a \sim \pi_\theta}\left[A_r^{\pi_{old}}(s_t, a) - \lambda A_{1-\epsilon_0}^{\pi_{old}}(s_t, a)\right]\right]$$

$$= \left(V^{\pi_{old}}(s_0) - \lambda q_{1-\epsilon_0}^{\pi_{old}}(s_0)\right) + \mathbb{E}_{s \sim \rho^{\pi_{old}}, a \sim \pi_\theta}\left[A_r^{\pi_{old}}(s_t, a) - \lambda A_{1-\epsilon_0}^{\pi_{old}}(s_t, a)\right] \tag{259}$$

$$A_r^{\pi_{old}}(s, a) = r(s, a) + \gamma \mathbb{E}_{s' \sim M(\cdot|s,a)}\left[V^{\pi_{old}}(s')\right] - V^{\pi_{old}}(s) \tag{260}$$

$$A_{1-\epsilon_0}^{\pi_{old}}(s, a) = c(s, a) + \tilde{c}_{1-\epsilon_0}^{\pi_{old}}(s, a) + \gamma \mathbb{E}_{s' \sim M(\cdot|s,a)}\left[q_{1-\epsilon_0}^{\pi_{old}}(s')\right] - q_{1-\epsilon_0}^{\pi_{old}}(s), \tag{261}$$

Note that

$$\mathbb{E}_{\pi_{old}}\left[A_{1-\epsilon_0}^{\pi_{old}}(s, a)\right] = \mathbb{E}_{\pi_{old}}\left[c(s, a) + \tilde{c}_{1-\epsilon_0}^{\pi_{old}}(s, a) + \gamma q_{1-\epsilon_0}^{\pi_{old}}(s') - q_{1-\epsilon_0}^{\pi_{old}}(s)\right] \tag{262}$$

$$= \mathbb{E}_{\pi_{old}}\left[\frac{p_{X^{\pi_{old}}(s')}\left(\frac{q_{1-\epsilon_0}^{\pi_{old}}(s)-c(s,a)}{\gamma}\right)}{\gamma \cdot p_{X^{\pi_{old}}(s)}\left(q_{1-\epsilon_0}^{\pi_{old}}(s)\right)}\left\{c(s, a) + \gamma q_{1-\epsilon_0}^{\pi_{old}}(s') - q_{1-\epsilon_0}^{\pi_{old}}(s)\right\}\right] \tag{263}$$

since

$$\tilde{c}_u^{\pi_{old}}(s, a) = \left(\frac{p_{X^{\pi_{old}}(s')}\left(\frac{q_{1-\epsilon_0}^{\pi_{old}}(s)-c(s,a)}{\gamma}\right)}{\gamma \cdot p_{X^{\pi_{old}}(s)}\left(q_{1-\epsilon_0}^{\pi_{old}}(s)\right)} - 1\right)\left\{c(s, a) + \gamma q_{1-\epsilon_0}^{\pi}(s')\right\} \tag{264}$$

$$1 = \mathbb{E}_{\pi_{old}}\left[\frac{p_{X^{\pi_{old}}(s')}\left(\frac{q_{1-\epsilon_0}^{\pi_{old}}(s)-c(s,a)}{\gamma}\right)}{\gamma \cdot p_{X^{\pi_{old}}(s)}\left(q_{1-\epsilon_0}^{\pi_{old}}(s)\right)}\right] \tag{265}$$

Therefore we use $\dfrac{p_{X^{\pi_{old}}(s')}\left(\frac{q_{1-\epsilon_0}^{\pi_{old}}(s)-c(s,a)}{\gamma}\right)}{\gamma \cdot p_{X^{\pi_{old}}(s)}\left(q_{1-\epsilon_0}^{\pi_{old}}(s)\right)}\left\{c(s, a) + \gamma q_{1-\epsilon_0}^{\pi_{old}}(s') - q_{1-\epsilon_0}^{\pi_{old}}(s)\right\}$ as the advantage for the $(1-\epsilon_0)$-quantile.

$$A_{1-\epsilon_0}^{\pi_{old}}(s, a) = \frac{p_{X^{\pi_{old}}(s')}\left(\frac{q_{1-\epsilon_0}^{\pi_{old}}(s)-c(s,a)}{\gamma}\right)}{\gamma \cdot p_{X^{\pi_{old}}(s)}\left(q_{1-\epsilon_0}^{\pi_{old}}(s)\right)}\left\{c(s, a) + \gamma q_{1-\epsilon_0}^{\pi_{old}}(s') - q_{1-\epsilon_0}^{\pi_{old}}(s)\right\} \tag{266}$$

Finally QCPO is based on PPO[20], the actual policy loss function is as follows:

$$L(\theta) = -\hat{\mathbb{E}}_{\pi_{\theta_{old}}}\left[\min\left\{\hat{A}_1(s, a, s'), \hat{A}_2(s, a, s')\right\}\right] \tag{267}$$

where

$$\hat{A}_1(s, a, s') = \mathrm{clip}\left(\frac{\pi_\theta(a|s)}{\pi_{\theta_{old}}(a|s)}, 1 - r_{clip}, 1 + r_{clip}\right) \times \hat{A}(s, a, s') \tag{268}$$

$$\hat{A}_2(s, a, s') = \frac{\pi_\theta(a|s)}{\pi_{\theta_{old}}(a|s)}\hat{A}(s, a, s') \tag{269}$$

$$\hat{A}(s,a,s') := \hat{A}_r(s,a,s') - \lambda \hat{A}_{q,1-\epsilon}(s,a,s') \tag{270}$$

$$\hat{A}_r(s,a,s') := r(s,a) + \gamma V_{\phi_{old}}(s') - V_{\phi_{old}}(s) \tag{271}$$

$$\hat{A}_{1-\epsilon_0}(s,a,s') = \frac{p_{\alpha_{old}(s'),\beta_{old}(s')}\left(\frac{q_{\psi_{old},1-\epsilon_0}(s)-c(s,a)}{\gamma}\right)}{\gamma \cdot p_{\alpha_{old}(s'),\beta_{old}(s')}\left(q_{\psi_{old},1-\epsilon_0}(s)\right)} \left\{c(s,a) + \gamma q_{\psi_{old},1-\epsilon_0}(s') - q_{\psi_{old},1-\epsilon_0}(s)\right\} \tag{272}$$

Here,

$$p_{\alpha_{old}(s'),\beta_{old}(s')}(x) = \frac{\alpha_{\xi_{old}}(s')}{\beta_{\zeta_{old}}(s')}\left(\frac{x}{\beta_{\zeta_{old}}(s')}\right)^{\alpha_{\xi_{old}}(s')-1} \exp\left(-\left(\frac{x}{\beta_{\zeta_{old}}(s')}\right)^{\alpha_{\xi_{old}}(s')}\right) \tag{273}$$

is the probability density function (PDF) of the approximated weibull distribution with parameter $\alpha_{\xi_{old}}(s')$ and $\beta_{\zeta_{old}}(s')$.

However, the variance of the ratio $\frac{p_{\alpha_{old}(s'),\beta_{old}(s')}\left(\frac{q_{\psi_{old},1-\epsilon_0}(s)-c(s,a)}{\gamma}\right)}{\gamma \cdot p_{\alpha_{old}(s'),\beta_{old}(s')}\left(q_{\psi_{old},1-\epsilon_0}(s)\right)}$ is large with actual samples.
Hence, for implementation, using the Taylor series $\log x = (x-1) + \frac{1}{2}(x-1)^2 + \cdots$ around $x = 1$, we smooth the weight as

$$\frac{p_{\alpha_{old}(s'),\beta_{old}(s')}\left(\frac{q_{\psi_{old},1-\epsilon_0}(s)-c(s,a)}{\gamma}\right)}{\gamma \cdot p_{\alpha_{old}(s'),\beta_{old}(s')}\left(q_{\psi_{old},1-\epsilon_0}(s)\right)}$$

$$\approx \left(1 + \text{clip}\left(\log \frac{p_{\alpha_{old}(s'),\beta_{old}(s')}\left(\frac{q_{\psi_{old},1-\epsilon_0}(s)-c(s,a)}{\gamma}\right)}{\gamma \cdot p_{\alpha_{old}(s'),\beta_{old}(s')}\left(q_{\psi_{old},1-\epsilon_0}(s)\right)}, -c_{\text{clip}}, c_{\text{clip}}\right)\right) \tag{274}$$

and apply (274) into (272) so the actual advantage estimate we used is

$$\hat{A}_{1-\epsilon_0}(s,a,s')$$
$$= \left(1 + \text{clip}\left(\log \frac{p_{\alpha_{old}(s'),\beta_{old}(s')}\left(\frac{q_{\psi_{old},1-\epsilon_0}(s)-c(s,a)}{\gamma}\right)}{\gamma \cdot p_{\alpha_{old}(s'),\beta_{old}(s')}\left(q_{\psi_{old},1-\epsilon_0}(s)\right)}, -c_{\text{clip}}, c_{\text{clip}}\right)\right)$$
$$\times \left\{c(s,a) + \gamma q_{\psi_{old},1-\epsilon_0}(s') - q_{\psi_{old},1-\epsilon_0}(s)\right\} \tag{275}$$

### E.4 Lagrange Multiplier for Quantile Constraint

We also need the Lagrange multiplier $\lambda$ in (270) for the policy loss function to satisfy the quantile constraint. The Lagrange multiplier is updated to minimize the Lagrange form of (QuantCP) $L_{quant}(\pi, \lambda) := V^\pi(s_0) - \lambda\left(q_{1-\epsilon}^\pi(s_0) - d_{th}\right)$ to satisfy the quantile constraint $q_{1-\epsilon}^\pi(s_0) \le d_{th}$. Thus, the update rule of the Lagrange multiplier $\lambda$ is $\lambda \leftarrow \max\{\lambda + \eta(q_{1-\epsilon}^\pi(s_0) - d_{th}), 0\}$, where $\eta$ is a learning rate. To constrain the outage probability of the sum of costs in a trajectory, we collect 100 trajectories, and compute the $(1-\epsilon)$-quantile of them to replace $q_{1-\epsilon}^\pi(s_0)$ in the Lagrange update rule.

### E.5 Hyper-parameters

For the quantile network, we used $n_q = 25$, and $u_i = \frac{2i-1}{2n_q} = \frac{2i-1}{50}$, $i = 1, \ldots, n_q(= 25)$ for $\{u_1, u_2, \ldots, u_{n_q}\}$. For training the Weibull network, we used the rightmost $k$-quantiles among $n_q$, and the $k$ is 8 ($\approx$ 30% of $n_q$ quantiles). The discount factor $\gamma$ is 0.99, and all learning rates for Adam optimizers for all parameters are $10^{-4}$. The $\eta$ for updating the Lagrange multiplier is 0.1. The $r_{clip}$ in (268) for updating the policy parameter is 0.1, and the $c_{clip}$ in (275) for computing $\tilde{c}_u^{\pi'}(s, a)$ is 0.5. Since PPO is an on-policy algorithm, it first collects 12000 samples by interaction with its environment. Then, it reshapes these samples by 120 sub-trajectories of length 100, and uses all sub-trajectories to update its parameters (this is because we use LSTM for the feature extraction network). This update is performed 8 times for the same collected sub-trajectories, then we remove them and collect new 12000 samples by interaction with the environment. This procedure is performed until the maximum training timesteps $5 \times 10^6$.