# OpenReview forum: "Quantile Constrained Reinforcement Learning: A Reinforcement Learning Framework Constraining Outage Probability"
_NeurIPS.cc/2022/Conference — NeurIPS 2022 Accept_

### Official Review · Reviewer_BZhD · 2022-07-08

**Rating:** 5
**Confidence:** 3
**Soundness:** 3 good
**Presentation:** 2 fair
**Contribution:** 2 fair

**Summary:**

This paper proposes a framework, named Quantile Constrained RL (QCRL), to constrain the quantile of the distribution of cumulative
sum cost that is a necessary and sufficient condition to satisfy the outage constraint. The paper converts the outage probability constraint in (ProbCP) into a quantile constraint which is equivalent to the original probabilistic constraint and solve the equivalent optimization using  Lagrange multiplier.

**Questions:**

1. How to know that the underlying distribution of the sum of costs is right-tailed in practice if we do not have much information about the true environment. How does this method perform in practice if the model assumption is mis-specified. For example, what if the real distribution is left-tailed or multi-modal?
2. Similar to Q1, how to choose proper values of $\epsilon_0$ and $d_{th}$ if there is no much information about the environment.
3. The experiments are done with 10 different seeds while the variance seems to be large. Is the proposed method sampling efficient and how the estimations especially the quantiles are affected by the data sample?
4. There are a lot of studies about Distributional RL which uses the Huber loss to learn the quantiles. Can this approach be applied here?

**Limitations:**

This paper considers a more general more general outage probability constraint for Constrained Reinforcement Learning. However, this paper fails to provide enough insight about this setup and the experimental results do not provide a strong support.

**Strengths And Weaknesses:**

Strength:

1. This paper considers the more general outage probability constraint in (ProbCP) than the classic expectation constraint (ExpCP) and modifies the optimization problem using quantile constraint.
2. The paper applies policy gradient theorem to the quantile and provides theoretical results for approximating the gradient of the quantile.

Weakness:

1. This paper does not fully clarify why it is necessary to employ the outage probability constraint in practice and what may be the potential impact of this setting to the real-world examples.
2. The empirical part is not strong enough to show the advantage of the proposed method, and some more realistic and interesting examples such as the autonomous driving or the telecommunication system that are mentioned by the authors may be helpful.

---

> ### Author Response · Authors · 2022-08-02
> **Response to Reviewer BZhD (Part 1)**
>
> [Comment] Simulation on realistic environments that need the proposed method.
>
> [Response]   Please note that the test environments  DynamicEnv,  SimpleButtonEnv and GremlinEnv in the paper are actually simplified versions of a real environment of autonomous serving robot.  The objective of the serving robot is to carry dishes to the customer as fast as possible while avoiding collision with  fixed objects and moving people. We can consider that a cost occurs whenever a collision happens either with fixed objects like pillars or tables or moving people in the restaurant, and we want to limit the probability of collisions.
> We demonstrated  that the proposed method can learn a well-performing policy in these simplified environments while successfully constraining the outage probability.
>
>
>
>
>
>
>
>
>
> [Comment] Setting $\epsilon_0$ and $d_{th}$
>
> [Response] We think that we cannot set these parameters without information about the environment.
> However, in many real-world RL problems, to accomplish the task, we need to have some information about the task and sometimes we even need to design the reward function to properly accomplish the task through RL.
>
> Let us consider the above automatic serving robot example. In order to actually run the robot, we need to design the reward function and the cost function to deliver dishes to the customer while avoiding collision which can damage the robot, other objects and people in the restaurant.  The setting of these functions is crucial to the learned behavior of the robot.
>
>
> In this example, we can design the reward function as the difference of distance to the customer minus a time penalty, and the cost function as an indicator of whether an object or person is in a caution range from the robot.
> Once we design the reward and cost functions, we can set  the sum cost threshold $d_{th}$  and the outage probability $\epsilon_0$. These are our design choice depending on restaurant management.
>
>
> [Comment] Variance of the experimental results
>
> [Response] As the reviewer mentioned, the results show  high variance in both the proposed method (QCPO) and the previous method (WCSAC). This suggests that the estimation of quantile or distribution is more difficult than estimation of  expectation. Thus, both QCPO and WCSAC have lower sample efficiency than algorithms with expectation constraint, and this is a common problem of constrained RL with distributional constraints. However, it is observed that QCPO has lower variance than WCSAC, showing that QCPO learns  policy more stably than WCSAC.
>
>
>
> [Comment] On the right-tail of the underlying distribution of cumulative sum costs.
>
>
> [Response] Note that the cost is for undesired events and is typically set as non-negative. In this case, the sum cost is bounded below by zero and is right-sided (i.e., has right-tail).
>
> As mentioned by the reviewer, the distribution of the sum cost can be multimodal.  However, large deviations theory says as follows [3]. If we set a threshold $\tau = \mu + \delta$ with sufficiently large $\delta >0$ (corresponding to small tail probability) for a right-tailed distribution satisfying large deviations principle, the probability of the sum cost above the threshold is exponentially decaying. Here, $\mu$ is the mean of the distribution of the sum cost.   Moreover, [3] says that the sum of a function of Markov chain satisfies large deviations principle. So, we can apply the tail probability approximation when $\tau$ is away from the mean (yielding small outage probability)  with large deviations principle.
>
> Please see the reply to the last comment, too.
>
>
>
> [3] A. Dembo and O. Zeitouni. Large Deviations Techniques and Applications. Springer, 1998.

---

> > ### Author Response · Authors · 2022-08-02
> > **Response to Reviewer BZhD (Part 2)**
> >
> > [Comment] Applicant of the proposed policy gradient for quantile to distributional RL.
> >
> > [Response] Thanks for this insightful comment. We  realized that the proposed method could be applied to on-policy distributional RL. Most distributional RL papers consider only the average return of its policy as their objective function,
> > but  can be used to  maximize  the $u$-quantile of the return distribution, say $u=0.2$
> >
> >
> > Note that in this case, we are interested in the left tail of the distribution, not the right tail of the distribution, to optimize a policy. The theory we derived in the paper can be applied directly to this case, but it needs additional steps to apply our implementation to this case. The only difficulty in the direct application of our method is an approximation of the probability density function (pdf) $p_{Z^\pi(s)}(\cdot)$ of the return random variable $Z^\pi$ to compute the additional cost $\tilde{c}^\pi_u(s, a) = \left( \frac{p_{Z^\pi(s')} \left( \frac{q_u^\pi(s) - r(s,a)}{\gamma} \right)}{\gamma p_{Z^\pi(s)}\left( q^\pi_u(s) \right)} - 1 \right) \cdot \left[ c(s,a) + \gamma q^\pi_u(s') \right]$. We omit $\pi$ and $s$ from the random variable $Z^\pi(s)$ and its quantile $q^\pi_u(s)$ for simplicity from now on. To approximate the left tail of the distribution using a Weibull distribution, we need to change the left tail to a right tail. Consider the left side of the distribution $p_{Z}(\cdot) 1_{Z \leq q_{0.5}}$ from the 0.5 quantile (median). We can make a pdf of some non-negative random variable $Y$ whose shape is the same as the flipped shape of the cropped distribution $p_{Z}(\cdot) 1_{Z \leq q_{0.5}}$ by shifting left to $q_{0.5}$ and then flipping the cropped distribution: $p_{Z}(\cdot) 1_{Z \leq q_{0.5}} \rightarrow p_{Z - q_{0.5}}(\cdot) 1_{Z - q_{0.5} \leq 0} \rightarrow p_Y(\cdot) = 2 \cdot p_{q_{0.5} - Z}(\cdot) 1_{q_{0.5} - Z \geq 0}$. Then, we can approximate the pdf of $Y$ using a weibull distribution, and use it to compute the additional cost. Therefore, we can apply the implementation of the proposed method to distributional RL in this way. We thank the reviewer for the suggestion.
> >
> > Please that the considered method can be an answer to the left-tail case in the above comment.

---

### Official Review · Reviewer_mHDz · 2022-07-12

**Rating:** 7
**Confidence:** 5
**Soundness:** 3 good
**Presentation:** 3 good
**Contribution:** 3 good

**Summary:**

The paper "Quantile Constrained Reinforcement Learning" studies the problem of reinforcement learning for constraining outage probability. The paper converts this constraint into an equivalent quantile constraint. The policy gradient theorem can not be directly applied, due to the lack of a closed form gradient under this constraint. The paper uses large deviations principle and derives an "approximate" gradient for the quantile. Using this, a policy gradient based reinforcement learning algorithm "Quantile Constrained Policy Optimization" (QCPO) is proposed. For theoretical result, an improvement theorem is provided. The experimental results show the improvement over existing methods.



**Questions:**

1. Please state the assumptions used in the theorems clearly.

2. The improvements shown in the approximations degrade as the complexity of environments increase. For example, the algorithm seems to perform better in simple button environment compared to other algorithms, but in Dynamic Env, its performance benefits are not much.

**Limitations:**

The limitations are not discussed in the paper. The authors are suggested to include a discussion in the paper about the possible limitations. The problem has a probability constraint, hence, with a small probability, the catastrophic events happen.

**Strengths And Weaknesses:**

So far the literature is mainly focused on expected cumulative sum cost constraint, very few policy gradient results exist for VAR and CVAR.  It is not a straightforward derivation of policy gradient for risk constrained problems as in expected cost constraint. This paper tries to overcome this issue by computing approximate gradients. The technical contribution is sound, and the paper is readable. These results are significant in the risk constrained RL literature. That said, in the main paper, the theoretical results have many assumptions. The repeated usage of phase "under the mild assumptions" is misleading (Theorem 1, Corollary 1, Lemma 1, Theorem 2, etc). Authors should clearly state the assumptions. In line 204, transition dynamics of the CMDP are assumed deterministic for the theorems to work. This has to be stated as an assumption. The experimental evaluation part is too simple and weak.

---

> ### Author Response · Authors · 2022-08-02
> **Response to Reviewer mHDz**
>
> [Comment] Assumptions for Theorems
>
> [Response] We agree on this point. Basically, the assumptions are about boundedness and smoothness. We will put the assumptions for Theorem 1 currently in Appendix B.1 into the main context if we have one extra page in the case that the paper is accepted.
>
>
>
>
> [Comments] Performance improvement
>
> [Response] As the reviewer mentioned, it is seen that there is less improvement in DynamicEnv, which is a harder task than SimpleButtonEnv. However, the result on GremlinEnv, which is the hardest task among the three and needs to avoid both fixed hazards and moving gremlins, shows that QCPO learns a better policy than WCSAC for $\epsilon_0= 0.2$. Thus, we cannot conclude that the performance improvement is less on the task with high complexity.

---

> > ### Comment · Reviewer_mHDz · 2022-08-09
> > **Comment on the Rebuttal Response**
> >
> > I thank the authors for taking time to answer my questions.

---

> > > ### Author Response · Authors · 2022-08-10
> > > **Thanks**
> > >
> > > Thanks again for accepting a reviewer of this paper, and for your valuable comments.

---

### Official Review · Reviewer_doKY · 2022-07-13

**Rating:** 6
**Confidence:** 4
**Soundness:** 3 good
**Presentation:** 3 good
**Contribution:** 3 good

**Summary:**

This paper considers the problem of solving a constrained MDP under quantile constrained, i.e., that the tail probability of cost is below a certain desired value.  The problem falls under the general framework of risk constrained reinforcement learning and is relevant to systems where a stranger metric than expected cost constraints are desired.  The approach is to derive a proxy for the policy gradient of the quantile probability, and utilize it in a policy gradient approach to ensure the constraint is met.  The derivation of the appropriate policy gradient term is the major contribution.

**Questions:**

Please see weaknesses.  Addressing the gaps in the logic, namely deterministic transition, nearness of base policy to current policy and hence finding delta and using the bound in in the implementation should be addressed.  I would increase my score if this is satisfactorily addressed.

**Limitations:**

Yes, it appears to be sufficient.

**Strengths And Weaknesses:**

Strengths

+ The paper makes a novel observation in obtaining the policy gradient of the Lagrange dual of the risk constrained problem. The policy gradient in this case depends not he tall probability itself, as well as a const that is upper bounded by what appears to be a policy independent term.

+ The authors use a standard PPO codebase to implement their approach and verify its performance in relevant scenarios.

Weaknesses


- The imprecations of Theorem 3 are unclear.  It appears that the result depends on deterministic transitions, and the availability of a base policy $\pi'$  whose KL divergence from $\pi$ should be less than $\delta.$  How doe we obtain $\pi'$?  Furthermore, the actual problem does not have deterministic transitions.  How does this result apply to the problem?

- Related to the above question, in section 4, step 2 is on estimating $X^\pi(s)$ to compute the cost $\tilde{c}^{\pi'}_u(s,a).$  How is this done?  The last paragraph of the section in step 3 says that we must compute the additional cost, again without showing how to do it.  Appendix E.2 only presents loss functions, not how this step is accomplished.

- The above seem to be fundamental to the analytical claims of the approach as well as its implementation.  Otherwise, the implementation would become some kind of constraint on the rate function of the occupancy measure generated by the policy.  I am not sure if the implantation is consistent with the theory developed.

---

> ### Author Response · Authors · 2022-08-02
> **Response to Reviewer doKY (Part 1)**
>
> [Comment] How to choose  $\pi'$
>
> [Response] We noticed that there is no explanation for choosing $\pi'$. We are sorry about the inconvenience of understanding the proposed method.
> Most on-policy algorithms such as TRPO[1] and PPO[2] use an old policy to constrain the current policy to be near the old policy,
> so a natural way of choosing a nearby base policy $\pi'$ for  $\pi$ update is to choose the old policy $\pi_{old}$ as $\pi'$.  Please note that in TRPO, the KLD between the old base policy $\pi_{old}$ and the updating policy $\pi$ is constrained within some distance $\delta$.
> So, if we use TRPO, then the issue of choosing $\pi'$ and distance $\delta$ is resolved.
>
> For other algorithms such as PPO,
> Theorem 3 can be interpreted in the other  way around. If we first simply select $\pi_{old}$ of PPO as $\pi'$, then the updating $\pi$ is near from the base policy $\pi_{old}=\pi'$, and we can compute the corresponding KL distance $\delta$ between $\pi_{old}$ (= $\pi'$) and $\pi$. Then, still
>  the inequality (13) holds for $\delta = \max_s KL\left( \pi_{old}(\cdot | s) || \pi(\cdot | s) \right)$. Please note that eq. (13) means that for small $\delta$, we have small approximation error.
>
>
>
>
> [1] J. Schulman, et al. Trust region policy optimization. ICML 2015.
>
> [2] J. Schulman, et al. Proximal policy optimization algorithms. arXiv:1707.06347
>
>
> [Comment] Computing the additional cost $\tilde{c}^{\pi'}\_u(s,a)$ and applying the proposed method when the dynamics are not deterministic.
>
> [Response] As noted by the above reply, we set $\pi'$ as $\pi\_{old}$ of PPO, when PPO is used as the baseline.  Once the base policy $\pi'$ is set by the old policy $\pi_{old}$, we can compute the additional cost $\tilde{c}^{\pi\_{old}}\_u(s,a)$ for the old policy $\pi_{old}$ by using estimated quantile $q^{\pi\_{old}}\_u(s,a)$ and estimated probability density function (pdf) $p\_{X^{\pi\_{old}}(s)}(\cdot)$ as follows:
>
> $\tilde{c}^{\pi\_{old}}(s,a) = \left( \mu^{\pi\_{old}}\_u(s,a) - 1 \right) \cdot \left[ c(s,a) + \gamma q^{\pi\_{old}}\_u(s') \right]$, where $\mu^{\pi\_{old}}\_u(s,a) := \frac{ p\_{X^{\pi\_{old}}(s')}\left( \frac{q^{\pi\_{old}}\_u(s) - c(s,a)}{\gamma} \right) }{\gamma \cdot p\_{X^{\pi\_{old}}(s)}\left( q^{\pi\_{old}}\_u(s) \right)}$, and $s' = h(s,a)$ (deterministic dynamics).
>
>
>
> In the case of stochastic dynamics, we can apply our theoretical results with some modifications. We know that Corollary 1 holds even if the dynamics are not deterministic, and we start from this. Since $\mu^{\pi\_{old}}\_u(s, a, s') := \frac{ p\_{X^{\pi\_{old}}(s')}\left( \frac{q^{\pi\_{old}}\_u(s) - c(s,a)}{\gamma} \right) }{\gamma \cdot p\_{X^{\pi\_{old}}(s)}\left( q^{\pi\_{old}}\_u(s) \right)}$ is a function of $s$, $a$ and $s'$, we can make the additional cost $\tilde{c}^{\pi\_{old}}\_u(s, a, s')$ as a function of $s$, $a$, $s'$: $\tilde{c}^{\pi\_{old}}\_u(s, a, s') = \left( \mu^{\pi\_{old}}\_u(s, a, s') - 1 \right) \cdot \left[ c(s,a) + \gamma q^\pi\_u(s') \right]$.
>
>
> Note that the form of this additional cost is the same as that of the deterministic dynamics case, but the only difference is the dependence on the next state $s'$. If we use an average of this additional cost over the next state, $\bar{c}^\pi\_u(s,a) = \mathbb{E}\_{s' \sim M(\cdot | s, a)}\left[ \tilde{c}^\pi\_u(s,a,s') \right]$, all remaining procedure  is the same as before.  One limitation of this approach is that we need multiple samples for the same pair $(s,a)$. Practically, this may not be feasible with environment interaction. Hence, practically an algorithm may be constructed without averaging over $s'$.

---

> > ### Author Response · Authors · 2022-08-02
> > **Response to Reviewer doKY (Part 2)**
> >
> > [Comment] The gap between theoretical results and its implementation
> >
> > [Response] The theoretical results state that we can approximate the policy gradient for the $u$-quantile by using the additional cost. However, in its implementation, we need more approximations as mentioned in Appendix E.3 for better performance.
> >
> > Let us consider the policy gradient of the $u$-quantile only. Then, the policy gradient of QCPO which is based on the PPO, is the gradient of $\mathbb{E}\_{s \sim \rho^{\pi\_{old}}(\cdot)} \left[ \mathbb{E}\_{a \sim \pi(\cdot | s)}\left[  A^{\pi\_{old}}\_{1-\epsilon\_0}(s,a) \right] \right] + C KL(\pi\_{old}(\cdot | s) || \pi(\cdot | s))$, where $\rho^\pi(s)$ is the stationary state distribution defined as $\frac{1}{1-\gamma} \sum\_t \gamma^t Pr(s\_t = s | s\_0, \pi)$. We omit the KL divergence term from now on for simplicity. When $\pi = \pi\_{old}$, the first term in the quantile policy loss function $\mathbb{E}\_{s \sim \rho^{\pi\_{old}}(\cdot)}\left[ \mathbb{E}\_{a \sim \pi(\cdot | s)}\left[  A^{\pi\_{old}}\_{1-\epsilon\_0}(s,a) \right] \right]$ can be rewritten as $\mathbb{E}\_{s \sim \rho^{\pi\_{old}}(\cdot)}\left[ \mathbb{E}\_{a \sim \pi\_{old}(\cdot | s)}\left[  c(s,a) + \tilde{c}^{\pi\_{old}}\_{1-\epsilon\_0}(s,a) + \gamma q^{\pi\_{old}}\_{1-\epsilon\_0}(s') - q^{\pi\_{old}}\_{1-\epsilon\_0}(s) \right] \right]$ and this becomes $\mathbb{E}\_{s \sim \rho^{\pi\_{old}}(\cdot)}\left[ \mathbb{E}\_{a \sim \pi\_{old}(\cdot | s)}\left[  \mu^{\pi\_{old}}\_{1-\epsilon\_0}(s,a) \cdot  \left( c(s,a) + \gamma q^{\pi\_{old}}\_{1-\epsilon\_0}(s') - q^{\pi\_{old}}\_{1-\epsilon\_0}(s) \right) \right] \right]$ using the following two facts: 1) the definition of $\tilde{c}^{\pi\_{old}}(s,a) = \left( \mu^{\pi\_{old}}\_{1-\epsilon\_0}(s,a) - 1 \right) \cdot \left[ c(s,a) + \gamma q^{\pi\_{old}}\_{1-\epsilon\_0}(s') \right]$, 2) the average of the probability ratio is always 1, $\mathbb{E}\_{a \sim \pi\_{old}(\cdot |s)} \left[ \mu^{\pi\_{old}}\_{1-\epsilon\_0}(s,a) \right] = 1$. Therefore, the policy gradient of the quantile policy loss function at $\pi = \pi\_{old}$ is the same as that of $\mathbb{E}\_{s \sim \rho^{\pi\_{old}}(\cdot)}\left[ \mathbb{E}\_{a \sim \pi(\cdot | s)}\left[  \mu^{\pi\_{old}}\_{1-\epsilon\_0}(s,a) \cdot \left( c(s,a) + \gamma q^{\pi\_{old}}\_{1-\epsilon\_0}(s') - q^{\pi\_{old}}\_{1-\epsilon\_0}(s) \right) \right] \right]$ at $\pi = \pi\_{old}$. Therefore, we approximate the policy gradient of the quantile policy loss function as the gradient of $\mathbb{E}\_{s \sim \rho^{\pi\_{old}}(\cdot)}\left[ \mathbb{E}\_{a \sim \pi(\cdot | s)}\left[  \frac{ p\_{X^{\pi\_{old}}(s')}\left( \frac{q^{\pi\_{old}}\_{1-\epsilon\_0}(s) - c(s,a)}{\gamma} \right) }{\gamma \cdot p\_{X^{\pi\_{old}}(s)}\left( q^{\pi\_{old}}\_{1-\epsilon\_0}(s) \right)} \left( c(s,a) + \gamma q^{\pi\_{old}}\_{1-\epsilon\_0}(s') - q^{\pi\_{old}}\_{1-\epsilon\_0}(s) \right) \right] \right]$.
> >
> >
> > This is explained in Appendix E.3.

---

> > > ### Comment · Reviewer_doKY · 2022-08-07
> > > **Response to rebuttal**
> > >
> > > Thanks you for the responses.  I went over the paper again viewing the results from the trust region perspective and it makes sense how you are approaching the problem.   There appears to be a little bit of a gap when addressing the stochastic transitions part, in that you can address it empirically but getting the equivalent of Theorem 3 does not seem to be completely straightforward.    I am increasing my score to 6.

---

> > > > ### Author Response · Authors · 2022-08-10
> > > > **Thanks**
> > > >
> > > > We are glad that our response helps you to understand this paper. Thanks again for your valuable comments, and we will revise this paper for better readability in the case that the paper is accepted.

---

### Official Review · Reviewer_uaQ3 · 2022-07-14

**Rating:** 7
**Confidence:** 3
**Soundness:** 3 good
**Presentation:** 3 good
**Contribution:** 3 good

**Summary:**

This paper proposes a framework QCRL to constrain the quantile of the distribution of cumulative sum cost, which is equal to guarantee a target probability of outage event that the cumulative sum cost exceeds a given threshold. This work is a good complement work for the constrained reinforcement learning area by considering a "new" quantile constraint. By constraining the quantile (instead of the expectation) of the cumulative sum cost,  we can have "precise" control over the target outage probability, which is very critical in practice , especially when high-cost event is unsafe.

This paper can be seen as a great work following WCSAC, the main contributions of this paper are: (1) better estimation for the quantile of the sum cost (2) focusing directly on the constraint on quantile.

**Questions:**


In this paper,  three different constrained RL problems and related algorithms are discussed:  (1) ExpCp (2) ProbCP ( equivalent to QuantCP) (3) CVaR-CP.  The manuscript would benefit from adding a formal formulation for CVaR constraints problems with the similar type as the other two problems.  Besides, the WCSAC is not for ProbCP,  but for CVaR-CP.

The second question is about the experiments. The authors compare three different methods, QCPO, PPO_Lag, and WCSAC. However, these three methods are for three different problems (ExpCP, ProCP, CVaR-CP) respectively. It is not surprising that QCPO is the best in terms of controlling outage probability. Is that fair to compare the performance for different methods when given different constraints? What are these experiments designed for?

The third issue is about the results on DynamicEnv-v0.  WCSAC does get a bad estimation of the tail probability such that we can not compare the return of WCSAC with QCPO later.  How about adding another experiment of WCSAC by replacing the quantile and tail estimation part with the proposed method in section 4.1.

Another question is about the range of outage probability $\epsilon_0$? If the $\epsilon$ is smaller (say 0.05, 0.02), could the proposed method still work well? What if the $\epsilon$ is very large (say 0.5)?

The last question is about the conclusion "the implementation algorithm learns an optimal policy while it keeps satisfying the outage probability constraint during its learning process".   I am confused about whether the constraint is always satisfied when you use a Lagrange-based optimization and also keep updating the Lagrange multiplier $\lambda$ during the learning procedure. It is also obvious from the experimental results that the constraint is not always satisfied, especially for the early stage.

#################
update:  Thanks for authors' response, which has addressed most of my concerns.  I would like to see the results of WCSAC with a Weibull approximation included in the final version.  Therefore, I would like to keep my score as 7.

**Limitations:**

yes

**Strengths And Weaknesses:**

I find it is easy to understand the ideas, motivation, and methods of this paper, results are also well-understandable. Overall, I think this paper gives a very critical and interesting setting for constrained RL theoretically and practically.

---

> ### Author Response · Authors · 2022-08-02
> **Response to Reviewer uaQ3**
>
> [Comment] Formulation of (CVaR-CP)
>
> [Response] We changed the sentence regarding WCSAC and mentioned that  WCSAC is proposed to solve (CVaR-CP) not (ProbCP) (line 103, line 331). However, WCSAC can be used for solving (ProbCP) because the CVaR constraint is a sufficient condition for outage probability constraint.
> We added the formal definition of (CVaR-CP) in appendix A.3 because of the limitation of space. We will move the definition of (CVaR-CP) to the main text if this paper is accepted, by using one extra page for an accepted paper.
>
>
>
> [Comment] The design of experiments and fairness of comparison
>
> [Response] As mentioned by the reviewer, the comparison can be unfair when we compare algorithms developed for (ExpCP), (CVaR-CP) and (ProbCP) in the measure of satisfying outage probability constraint.
> However, we found no algorithm  solving (ProbCP), so we cannot compare the proposed method developed for (ProbCP). Most previous methods were proposed to solve (ExpCP), and the WCSAC algorithm  solving (CVaR-CP) was proposed recently.
> Note that the method for (CVaR-CP) can be used to solve (ProbCP) in theory because the CVaR constraint is a sufficient condition for the outage probability constraint. Thus, we chose WCSAC for performance comparison.
>
>
>
>
>
>
> [Comment] Result of WCSAC with a Weibull approximation
>
> [Response] Thanks for this insightful comment. We are running the experiment of WCSAC with a Weibull approximation, and hope that we can show this result at least in the final version.
>
>
> [Comment] Performance of QCPO with various target outage probability ($\epsilon_0 = 0.5, 0.05, 0.02$)
>
> [Response] As suggested by the reviewer, we ran QCPO for $\epsilon_0 = 0.5, 0.05, 0.02$, and  added the results in Appendix D.1. We observed that QCPO successfully learns a policy satisfying the target outage probabilities even in small outage probability (0.05, 0.02) and high outage probability 0.5.
>
>
> [Comment] The sentence about the conclusion
>
> [Response] We admit that the sentence that the reviewer mentioned can mislead readers. We changed the sentence simply to "the implemented algorithm satisfies the outage probability constraint after the training period".

---

### Author Response · Authors · 2022-08-02
**Common Response**

We thank all reviewers for their valuable comments. During the rebuttal period, we changed some sentences in the paper as the reviewers commented, placed the result of a more complex environment, GremlinEnv, in the main text, and ran simulations on QCPO with various target outage probabilities (large or small) and added the results in Appendix D.1. We colored the changed parts in blue both in the paper and the appendix for the reviewers to find these changes easily. Some experiments required by the reviewers  are still running, and we hope that we can add these results before the discussion period ends.

[Comment] About the tested environments

[Response] As mentioned in the introduction and background of this paper, controlling the outage probability is meaningful for real-world environments such as autonomous driving cars and telecommunication. The proposed method can have benefit in such safety-critical environments that need to constrain outage probability. We measured the performance of the proposed method in the tested environments of  DynamicEnv, SimpleButtonEnv, and GremlinEnv, and these environments are simplified versions of a safety-critical real environment of  autonomous serving robot. SimpleButtonEnv is the simplest case in which the robot moves two fixed  destinations iteratively while avoiding pillars at fixed locations, DynamicEnv is the environment where the robot moves a random destination while avoiding pillars at  random locations, and finally GremlinEnv is the hardest environment in which the robot moves to a random destination while avoiding both pillars at  random locations and customers moving in some random areas.
Therefore, the tested environments for  the proposed method are already simple versions of a real autonomous  moving robot.



[Comment] About the experimental results

[Response] Note that this is the first work that directly solves (ProbCP) through equivalent (QuantCP), so we cannot compare the proposed method directly with other algorithms proposed to solve (ProbCP).   Most of the previous methods are designed to solve (ExpCP) and the previous method, called WCSAC, was proposed to solve (CVaR-CP) which is a  stricter problem than (ProbCP). Therefore, our experiments are performed to demonstrate the issue of applying the previous algorithms for (ExpCP) and (CVaR-CP) to solve (ProbCP)
and to show that the derived theoretical results can be implemented and the implemented algorithm can learn a policy satisfying  the outage probability constraint.

---

### Meta-Review · Area_Chair_HGP5 · 2022-08-25

**Recommendation:** Accept
**Confidence:** Certain

**Metareview:**

This paper made novel contributions in developing policy gradient methods in constrained RL. Unlike previous approaches that focused on expected cost constraint this work studied a more chance-constrained setting and proposed a Quantile Constrained RL (QCRL) framework that is based on distributional RL and Large Deviation Principle (LDP). Experiments show that this newly developed approach outperforms SOTA both in learning performance and safety guarantees. Therefore, without major controversies the review committee has the consensus of accepting this paper to be published at NeurIPS22.


**Award:**

No

---

### Decision · Program_Chairs · 2022-09-14

Accept